



# Climate change impacts on regional fire weather in heterogeneous landscapes of Central Europe

Julia Miller[1,2,3], Andrea Böhnisch[4], Ralf Ludwig[4], and Manuela I. Brunner[1,2,3]

[1]Institute for Atmospheric and Climate Science, ETH Zurich, Zurich, Switzerland
[2]WSL Institute for Snow and Avalanche Research SLF, Davos Dorf, Switzerland
[3]Climate Change, Extremes and Natural Hazards in Alpine Regions Research Center CERC, Davos Dorf, Switzerland
[4]Department of Geography, Ludwig-Maximilians-University Munich, Munich, Germany

**Correspondence:** Julia Miller (julia.miller@slf.ch)

**Abstract.** Wildfires have reached an unprecedented scale in the Northern Hemisphere. The summers of 2021 and 2022 demonstrated the destructive power of wildfires especially in Northern America and Southern Europe. Global warming indicates that fire seasons will become more extreme and will extend to more temperate regions in northern latitudes in the future. Multiple studies claim that natural variability hides the trend of increasing fire danger in climate model simulations for future potentially fire-prone areas. Single Model Initial-Condition Large Ensembles (SMILEs) help scientists to distinguish climate trends from natural variability. So far, the SMILE framework has only been applied for fire danger estimation on a global scale. In this study, we use a regional SMILE of the Canadian regional climate model version 5 (CRCM5-LE) over Central Europe under the RCP 8.5 scenario from 1980 to 2099, to analyze fire danger trends in a currently not fire-prone area. We use the meteorological Canadian Fire Weather Index (FWI) as a fire danger indicator. The study area covers four heterogeneous landscapes, namely the Alps, the Alpine Foreland, the lowlands of the Southern German Escarpment and the Eastern Mountain Ranges of the Bavarian Forest. We demonstrate that the CRCM5-LE is a suitable dataset to disentangle climate trends from natural variability in a multivariate fire danger metric. Results show the strongest increases in the median (50th) and extreme (90th) percentile of the FWI in the northern parts of the study area in the summer months July and August, where high fire danger becomes the median condition and extremes occur earlier in the fire season. The southern parts of the study region are affected less strongly, but due to weaker variability in these regions, time of emergence (TOE) is reached there in the early 2040's. In the northern parts, the climate change trend exceeds natural variability in the late 2040's. We find that today's threshold for a 100 year FWI event, will occur every 30 years by 2050 and every 10 years by 2099. Our results highlight Central Europe's potential for severe fire events from a meteorological perspective and the need for fire management in the near future even in temperate regions.

## 1 Introduction

The wildfire seasons of 2021 and 2022 affected the northern hemisphere at an unprecedented scale. Especially South-Eastern Europe and British Columbia experienced multiple extreme fire events in terms of intensity, severity and damage (Giannaros et al., 2022; Gillett et al., 2022). In Greece, five wildfires at the beginning of August 2021 burned an area of 94 000 ha, which





corresponds to three times the average annual burned area over the period 2008 to 2019 (Giannaros et al., 2022). 90 % of

the village Lytton in British Columbia (Canada) was destroyed by fires in the summer of 2021 during a heatwave caused by a blocking synoptic weather condition (Hoffman et al., 2022). While the Mediterranean region and the Western US are historically fire prone areas, Central Europe showed exposure to wildfires only in the recent years, e.g. in Treuenbritzen 2022, Brandenburg, Germany (Spiegel, 2022), and Küps 2022, Bavaria, Germany (BR, 2022).

In many cases, these fire events occurred under fire-favouring conditions of hot, dry and windy weather during the summer

months. Summer heatwaves and drought events decrease soil moisture and increase the flammability of the vegetation prior to the fire event (Ruffault et al., 2020). Blocking synoptic conditions trap hot air over distinct areas and lead to extreme temperatures, which contribute to very high fire danger (Hoffman et al., 2022). Due to climate change, fire weather and hence the likelihood of fire events is projected to increase in several regions of the world in the future (IPCC, 2021). From a meteorological perspective, the risk of igniting a fire increases with higher temperatures and wind speed and with lower relative humidity.

Alterations in these variables will more than double the risk of extreme fire weather until the end of the 21st century (Touma et al., 2021) and increase the duration, severity and spatial extent of fires (De Rigo et al., 2017; Ruffault et al., 2020; Fargeon et al., 2020; Bowman et al., 2020).

Fire indices, such as the Canadian Fire Weather Index (FWI) (Van Wagner and Pickett, 1985), the National Fire Danger Rating Sytem (NFDRS) of the U. S. Forest Service (Bradshaw et al., 1984), and the Australian McArthur Rating Sytem (Mark

5) (McArthur, 1966) represent the statistical correlation between fire events and meteorological conditions. They have been proven to produce reliable ratings of fire danger in short and long term weather predictions on a global scale. However, these indices only describe the probability of a fire occurrence and do not guarantee an actual fire ignition (Di Giuseppe et al., 2016). For assessing long-term fire risk with climate projections, the FWI is the most commonly used index, because it solely relies on meteorological inputs and does not propagate ambiguity from land use change (Touma et al., 2021).

Trends in fire risk show robust increases for Southern Europe and the Mediterranean region (IPCC, 2021), but also in the Boreal zone, fire season length and fire frequency are projected to increase under climate change (Bakke et al., 2023). Ruffault et al. (2020) have shown that under the RCP 8.5 emission scenario, the frequency of heat induced wildfires will increase by 30% in the Mediterranean region by the end of the century. Fargeon et al. (2020) found that under RCP 8.5, today's 10 year FWI maxima are reached every second year in the future in France. In Central Europe, trends related to fire danger are uncertain

and not clearly distinguishable from natural variability. Arnell et al. (2021) and Fargeon et al. (2020) have shown for England and France, respectively, that this uncertainty originates from an under-representation of natural variability in climate multi-model ensembles. In France, future fire danger exceedance of inter-annual variability decreases from South to North (Fargeon et al., 2020). Arnell et al. (2021) assessed the effects of climate change on fire danger indicators for the UK and found that the magnitude of fire danger change is hidden by the large natural variability of the input variables of the fire danger index and

the differences between the different climate multi-model ensembles. Both studies highlight the importance of quantifying the natural variability of changes in future fire weather and its relevance for decision making with respect to fire risk mitigation and planning. However, it is challenging to properly quantify natural variability with multi-model ensembles for temperate climate regions (Arnell et al., 2021; Fargeon et al., 2020).



This limitation, i. e. the under-representation of natural variability in fire danger estimates in regions with currently temperate climate, can be overcome by evaluating climate model simulations derived from a single model initial-condition large ensemble (SMILE). SMILEs represent an ensemble of simulations derived using one single climate model started at different initial conditions. This allows SMILEs to account for the internal variability of the climate system (Maher et al., 2021; Kay et al., 2015; Deser et al., 2012). Touma et al. (2021) successfully attributed changes in fire danger to anthropogenic greenhouse gas increases by analysing results from the global Community Earth System Model Large Ensemble. Most of the available SMILEs rely on global circulation or global earth system models with a coarse spatial resolution and are unsuitable to assess changes in fire weather over regions with complex terrrain such as Central Europe including the Alps.

In this study, we use the CRCM5-LE, a regionally downscaled, high-resolution SMILE, to disentangle climate change induced fire danger trends from natural variability over heterogeneous landscapes in Central Europe. First, we assess the suitability of the dataset, consisting of 50 climate model members, to reproduce typical FWI characteristics which are similar to the ERA-5-based FWI benchmark provided by Vitolo et al. (2020) for the present time period (1980–2009). The unique setup of the CRCM5-LE allows us to further evaluate how fire danger increases in the future under the RCP 8.5 greenhouse gas emissions scenario by taking internal variability into account. Second, we test the following hypotheses on future fire weather trends in the study area: (H1) The spatio-temporal development of the FWI between 1980 and 2099 in Central Europe increases significantly in four hydro-climatologically diverse subregions; (H2) the time of emergence (TOE) is reached latest by 2099; and (H3) today's 100-year FWI occurs at least every fifty years by the end of the century.

## 2 Data and Methods

### 2.1 CRCM5-LE

To quantify changes and natural variability in fire danger trends for Central Europe, we use the Canadian Regional Climate Model version 5 Large Ensemble (CRCM5-LE) of Leduc et al. (2019). The dataset consists of 50 members at a spatial resolution of 12 km and was generated within the ClimEx project (https://www.climex-project.org/) to assess the hydrological impacts of climate change in Bavaria and Québec. It includes continuous simulations of climate variables from 1950 to 2099 under the RCP8.5 emission scenario over two domains in Europe and Northeast North America (Leduc et al., 2019).

The CRCM5-LE is derived from the CanESM2-LE (Fyfe et al., 2017), which was created by applying small random perturbations at two different points in time (i. e. 1850 and 1950) to a 1000-year equilibrium climate simulation under pre-industrial conditions (Leduc et al., 2019). In a first step, small random atmospheric perturbations were added to the equilibrium run to obtain five historical simulation families starting in 1850. In a second step, ten random perturbations were added to each family, resulting in a 50 member ensemble. After a 5-year spin-up phase, the modeled climate of the initialized 50 members can be regarded as independent. This global SMILE was dynamically downscaled using the CRCM5 (Martynov et al., 2013; Šeparović et al., 2013) to obtain the regional SMILE CRCM5-LE (Leduc et al., 2019). For more details on the ensemble setup, the reader is referred to Leduc et al. (2019) (CRCM5-LE) and Fyfe et al. (2017) (CanESM2-LE).





Our analysis considers the period 1980 to 2099. At this time, all members share the same climatology and span a range of possible climate realizations, which give insights into the internal climate variability of the model (Leduc et al., 2019). In this study, we interpret internal variability as natural variability (Böhnisch et al., 2021; Von Trentini et al., 2019; Kay et al., 2015). A comparison between the CRCM5-LE and a multi-model ensemble (i. e. EURO-CORDEX) was conducted by Von Trentini

et al. (2019). Their results have shown, that the CRCM5-LE shows smaller member spread for temperature and equal member spread for precipitation than EURO-CORDEX (Von Trentini et al., 2019). The CRCM5-LE was bias corrected over the study area for the FWI input variables at a three-hourly resolution using the quantile mapping approach of Mpelasoka and Chiew (2009) (Poschlod et al., 2020). Bias corrected data are commonly used for projections of fire weather indicators like the FWI (e. g. Yang et al. (2015), Kirchmeier-Young et al. (2017), Ruffault et al. (2020), Fargeon et al. (2020)), because frequencies of

FWI extremes are significantly better represented than in non-bias-corrected climate data (Yang et al., 2015).

## 2.2 Study Area

Our study assesses changes in fire danger over a hydro-climatologically diverse region in Central Europe with temperate climate. The boundaries of the study area are set by the river catchments of the Danube, Main and Elbe, which intersect with the German federal state of Bavaria. The study area exceeds the boundaries of political Bavaria in terms of these catchments and

is therefore further referred to as "Hydrological Bavaria", short HydBav. HydBav has an overall size of approximately 103,200 km$^2$. We divide HydBav into four subregions according to their geography and climatology: (1) The Alps in the South, (2) the Alpine Foreland north of the Alps bounded by the course of the Danube, (3) the Southgerman Escarpment north of the course of the Danube and (4) the Eastern Mountain Ranges of the Bavarian Forest in the East of the study area (s. figure 1). This subdivision into complex landscapes is adopted from the ClimEx-Project and the study of Willkofer et al. (2020). Since

fire is closely related to the availability, or rather the absence of water, we assume that the water availability, climatology and landscape of different river systems reflect the fire regime of an area.

Figure 1 gives a brief overview of HydBav and its four subregions. According to the present climate period between 1980 and 2009 (present), the mean precipitation over the study areas increases from north to south, with annual precipitation sums between 500 and 1100 mm for the South German Escarpment, 1000 mm for the Eastern Mountain Ranges, 1500 and 2500

mm in the Alpine Foreland and 1000 and 2000 mm in the Alps. However, the valleys of the Inn catchment represent a more arid region with precipitation sums lower than 1000 mm (Poschlod et al., 2020). The annual mean temperatures are also higher in the North than in the South. The annual mean temperature in the Main catchment, which mainly covers the South German Escarpment subregion is around 10 °C, whereas in the Alps, the annual mean temperature is around 5 °C. For the regions of the Alpine Foreland and Eastern Mountain Ranges, temperatures vary between 6 and 9 °C, depending on the elevation (Willkofer

et al., 2020). The climatology in the study area is influenced by orography (Poschlod et al., 2020). The influence of orography on local conditions is relevant for wildfire propagation. For example, steep slopes can favour fire spread due to local thermal winds and southern facing slopes show hotter and drier conditions, which increases the risk of fire ignition and propagation (San-Miguel-Ayanz et al., 2018).

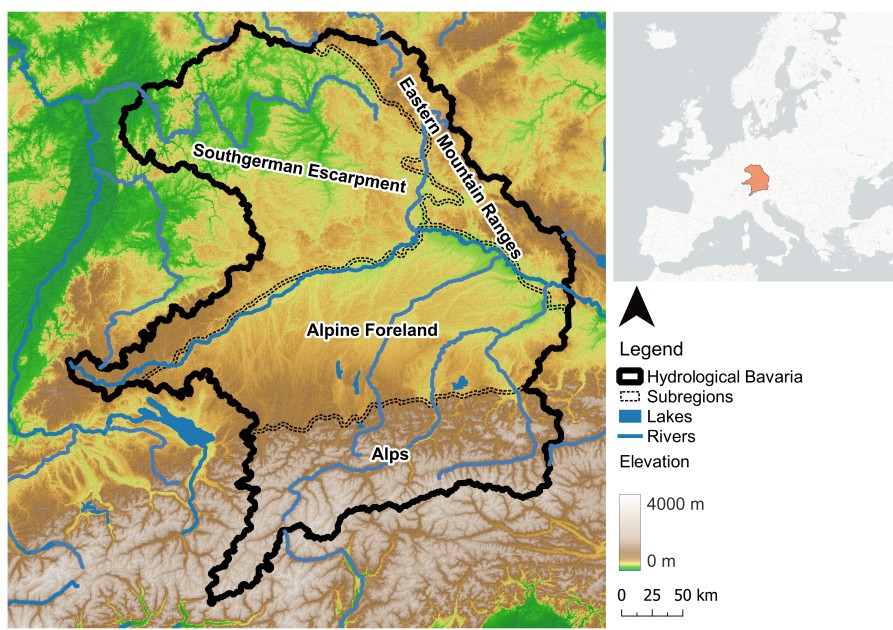

**Figure 1.** Subregions by landscapes and land cover of Hydrological Bavaria (modified, CLMS (2021))

### 2.3 The Canadian Fire Weather Index

In this study, we use the Canadian Fire Weather Index (FWI) of Van Wagner and Pickett (1985) to asses fire risk, because fire occurrences are strongly related to the FWI (Barbero et al., 2015) and its global applicability has been demonstrated by several studies (Di Giuseppe et al., 2016; Touma et al., 2021). The Canadian Forest Fire Weather Index System (CFFWIS) constitutes of five sub-indices, which together built the sixth index, i.e. the final FWI (s. figure 2). The CFFWIS uses meteorological conditions of the atmosphere on the day of interest (temperature, relative humidity, wind speed at noon and 24-h accumulated 130 precipitation) and antecedent weather conditions up to 52 days to estimate fire behaviour and fuel moisture (Van Wagner, 1987).

The first three sub-indices represent the fuel moisture codes and can be understood as bookkeeping systems, which increase moisture after rain and reduce moisture for each day of drying. Fine Fuel Moisture Code (FFMC), Duff Moisture Code (DMC) and Drought Code (DC) model daily changes in the moisture content of three different fuel layers with respect to different 135 time lags (De Rigo et al., 2017): The FFMC rates the moisture content of the surface litter (up to 1.2 cm), the DMC accounts for moisture of loosely-compacted organic matter in up to 7 cm depth and the DC estimates the moisture content of compact, organic layers up to 15 cm of ground depth. According to the increasing layer depth of the specific fuel moisture codes, the response to immediate atmospheric effects decelerates (De Rigo et al., 2017). The fuel moisture codes are considered to dry exponentially over time, so that their immediate drying rate is proportional to the free moisture content. The time lag accounts



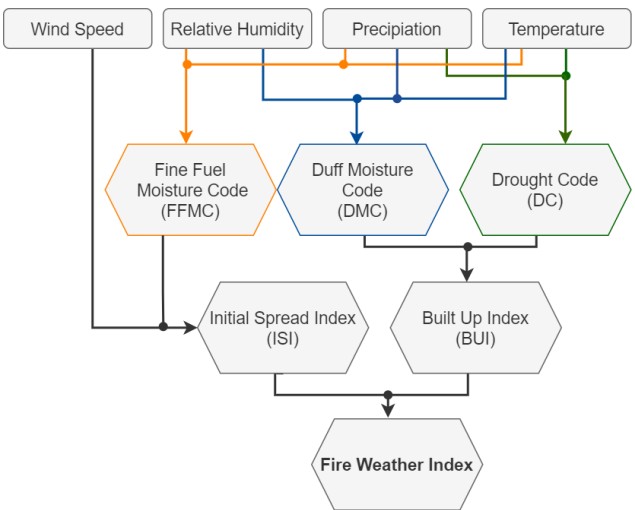

**Figure 2.** The Canadian Forest Fire Weather Index System, its input variables, and its intermediate indices. Fuel moisture codes (FFMC, DMC and DC) capture the antecedent moisture conditions of the vegetation. Fire behaviour codes (ISI, BUI and FWI) describe the potential spread and intensity of the fire (modified, Van Wagner (1987))

for the drying speed. DMC and DC respond to changing day length as the season progresses, since less time is available for drying when day length decreases (Van Wagner and Pickett, 1985).

The other two sub-indices, i.e. Build Up Index (BUI) and Initial Spread Index (ISI), together with the resulting FWI, describe the fire behaviour in case of an ignition. They are stateless and without memory of past conditions. The ISI combines wind speed and the FFMC to represent the rate of spread without the influence of fuel variability. The BUI combines DMC and

DC to represent the available fuel of the spreading fire. Finally, a combination of ISI and BUI leads to a representation of the potential intensity of the spreading fire in terms of the energy output rate per unit length of fire front, known as the FWI (De Rigo et al., 2017).

Originally, the index was calibrated to pine forest. Pine forest is found almost continuously across Canada, where the index was developed. However, the main goal of the CFFWIS is to create a fire danger rating solely based on weather and to provide

uniform results throughout Canada. Therefore, the calibration to a specific fuel type can be neglected (Van Wagner and Pickett, 1985). The applicability to other fuel types in different regions of the world has been demonstrated by various studies (e.g., Di Giuseppe et al., 2016; Barbero et al., 2020; De Rigo et al., 2017; Touma et al., 2021). The full formulas of the CFFWIS and a detailed description of all sub-indices is provided by Van Wagner (1987).

### 2.4 Estimating Fire Danger using the CRCM5-LE

We calculated daily FWIs for each year (January to December) and climate model ensemble member between 1980 and 2099 using the CFFDRS R package (Wang et al., 2017). The generated dataset is later cropped to the fire season (April 1st to September 30th) of the northern hemisphere (Vitolo et al., 2019). If not stated otherwise, the results shown refer to this fire





season. Subsetting the dataset to the fire season of annually calculated FWI values crops out the spin-up phase of 52-days in the DC. To facilitate the interpretation of the FWI, we use the seven fire danger classes proposed by the European Forest Fire Information System (EFFIS, 2021) and assign the FWI to particular fire danger levels. These FWI danger levels and their corresponding color mapping are shown in table 1.

**Table 1.** Fire danger levels of the FWI according to EFFIS (2021)

| FWI range | FWI danger level | Color |
|-----------|------------------|-------|
| < 5.2     | No Danger        |       |
| 5.2 - 11.2 | Low             |       |
| 11.2 - 21.3 | Moderate       |       |
| 21.3 - 38 | High             |       |
| 38 - 50   | Very High        |       |
| 50-70     | Extreme          |       |
| > 70      | Very Extreme     |       |

To ensure that the CRCM5-LE samples the FWI in a meaningful way, we compare the CRCM5-LE FWI for the reference period (1980–2009) to the ERA-5 based FWI dataset of Vitolo et al. (2020), further referred to as "reference dataset" (REF). A majority of the reference data points is located in the blue shaded area, which represents the ensemble's standard deviation for the FWI median of each member (s. figure 3). The remaining data points are located between the 25th and 75th percentile of the ensemble (blue lines). Overall, the ensemble slightly overestimates the FWI by an average deviation of +0.76, but includes the reference dataset values within its 25th and 75th percentiles. The spatial differences between the dataset are fairly low for the

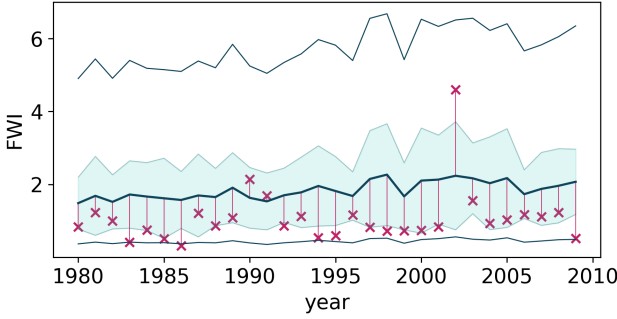

**Figure 3.** Median FWI for the CRCM5-LE mean (thick blue line) and standard deviation (light blue shading) in comparison to the reference dataset of Vitolo et al. (2020) marked pink (X for values, lines for deviation from the CRCM5-LE mean). Top and bottom blue lines mark the 25th and 75th percentile of the CRCM5-LE.

Alps and Alpine Foreland in the South (s. figure 4). In the northern parts and especially northwestern parts (i. e. Southgerman Escarpment) of the study area, the CRCM5-LE overestimates FWI values in comparison to the REF dataset on a magnitude





between two and four. However, this does not affect the climate change impact assessment of our study, because we compare
       FWI values solely derived from the CRCM5-LE.

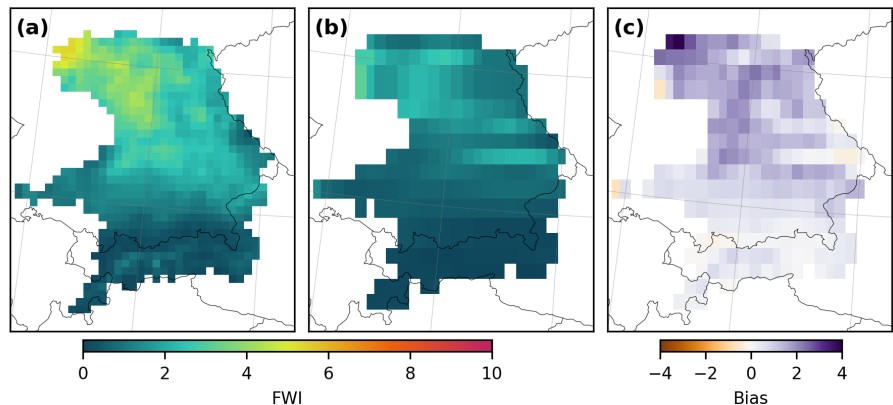

**Figure 4.** Median FWI of (a) the CRCM5-LE, (b) reference dataset of Vitolo et al. (2020) and (c) difference (CRCM5-LE - reference
dataset) for the present time period (1980–2009). The dataset difference is calculated from resampling (a) to the spatial resolution of (b)
using a nearest neighbour approach.

## 2.5    Changes in Fire Danger

### 2.5.1    Trends

We evaluate the fire danger trends derived from the CRCM5-LE over the time period 1980 to 2099 in the study area with
statistical metrics: Median conditions are examined using the $50^{th}$ percentile (median) of the FWI. Extreme conditions are
       evaluated via the $90^{th}$ percentile (extreme). The percentiles are calculated for different aggregation levels, either temporally,
       summarizing FWI values of a fire season on daily, monthly or annual basis, or spatially for the previously defined subregions.
       Increasing fire danger is either analyzed continuously from 1980 to 2099 or compared between two climate periods. For the
       climate period comparison, the dataset is split into two 30-year periods: 1980–2009 as present and 2070–2099 as future.
To highlight areas which show particularly robust and strong changes in the FWI between the climate periods, signal maps
       are created using the approach of Böhnisch et al. (2021). Signal maps consider the robustness and magnitude of fire danger
       increases. If the average fire danger level in the future period is higher in comparison to the fire danger level in the present
       period for more than 90 % of the climate model ensemble members (45 of 50), the signal is assumed to be robust. This method
       is used instead of a statistical test (Böhnisch et al., 2021). We create signal maps for the fire season months representing the
median and extreme FWI (s. figure 5) using the FWI danger levels of EFFIS (2021) (s. table 1). Further, we provide fire-rings in
       the style of the warming stripes of Hawkins (2018) to show how the FWI changes over the years on a monthly and subregional
       scale.



### 2.5.2 Time of Emergence

The second part of the climate change impacts analysis focuses on the time of emergence (TOE), which is calculated following
the approach of Fargeon et al. (2020): the TOE is reached when a projected metric (e. g. the median of the FWI) crosses the upper bound of the confidence interval. The confidence interval is defined as one standard deviation across the climate model ensemble members of the present climate period. The TOE is defined as the time when the 30-year running average trend of the ensemble mean exceeds the confidence interval. To account for the heterogeneous climate conditions in the study area, the TOE is calculated for each subregion separately. We use the median and extreme FWI percentiles of each fire season and
member between 1980 and 2099 for each subregion as the basis for calculating the time of emergence.

### 2.5.3 Frequency Changes

We assess changes in the fire danger levels of the median and extreme data samples of our dataset by comparing the frequency of daily fire danger classes for each subregion between the present and future climate periods (median) and calculating the return periods of different FWI extreme thresholds (extreme). To illustrate the median changes of fire danger, we classify the
daily values of each member of the CRCM5-LE according to the EFFIS classification in table 1 (EFFIS, 2021). We show the relative frequency of each fire danger level across the ensemble members for the present and future climate period to highlight fire danger level changes in the four subregions.

We determine return periods for different fire danger levels to put the extreme values of the CRCM5-LE in a more tangible context. We calculate the return periods on the basis of the 90th, 95th, 98th and 99th percentiles of the present climate period
(1980–2009) to account for 10, 20, 50 and 100 year FWI events in the four subregions. From 2010 to 2099, we create centered 30-year windows of our data sample and determine the FWI percentiles corresponding to the different return periods of the present climate period. The last full 30-year window is 2084. For the following years the window size decreases by one element until 2099, with a window size of 15 years. We then compute the non-exceedance probability of the present percentiles given the future cumulative distribution. From the future non-exceedance probability, we estimate the future return periods using the
function

$$T = 1/(1-p) \tag{1}$$

where $T$ is the return period and $p$ is the non-exceedance probability (Brunner et al., 2021; Coles, 2001).

## 3 Results

### 3.1 Trends

Fire danger will increase up to a high level for the FWI median (50th percentile) and up to an extreme level for the FWI extreme (90th percentile) in the study area by the end of the century (2070–2099) (s. Figure 5). Significant changes for an increase of one fire danger level (thin dot in figure 5) first occur in June and remain present until September for the median and extreme





FWI. For increases of two fire danger levels, significant changes occur in the months July and August for the median FWI and for the months July, August and September for the extreme FWI (thick dot in figure 5). We distinguish between weaker (one level, thin dots) and stronger (two levels, thick dots) fire danger level rises, because in July and August almost the entire study area shows a robust fire danger level rise of one level for both median and extreme conditions. This helps us to identify regional hotspots. The Southgerman Escarpment in the northwest is most affected by changes in the median FWI while the Alps and the Eastern Mountain Ranges experience the strongest fire danger level rises in the extreme FWI in the months July to September (s. figure 5).

Increases in fire danger are particularly pronounced from July to September, but are also visible throughout Mai to September (s. figure 6). The median case points out that high fire danger becomes the average condition in the Alpine Foreland by 2080, in the Southgerman Escarpment by 2060 and in the Eastern Mountain Ranges by 2070 (s. figure 6 [1]). The Alps are exposed to high fire danger only in the extreme case (s. figure 6 [2]) from 2070 onwards. The other subregions are much more strongly affected in the extreme case: Very high and high fire danger occur frequently in July and August in the second half of the 21$^{st}$ century in the Alpine Foreland and Eastern Mountain Ranges. In the Southgerman Escarpment, this is the case in June and September. However, for July and August, very high fire and high danger occur frequently from 2030 onwards (s. figure 6). The ensemble mean shows hardly any fire danger changes over the 21$^{st}$ century in the median and extreme case for April (s. figure 6). High fire danger becomes the mean condition (median) in the summer months for large parts of the study region (figures 5 and 6).

## 3.2  Time of Emergence

The results for the TOE show that the climate change signal exceeds the natural variability before the middle of the 21$^{st}$ century in all subregions for both median and extreme FWI (s. figure 7). For all subregions, except the Alpine Foreland, the TOE is reached in the same year for the median and the extreme FWI. The earliest TOE is reached in the Alps in 2032, followed by the Alpine Foreland in 2039 for the median and in 2041 for the extreme FWI. In the Southgerman Escarpment, the TOE is reached in 2044 and in the Eastern Mountain Ranges in 2047 for both the median and the extreme FWI. This finding indicates that the distribution of the FWI extremes resembles the distribution of the FWI median.

FWI changes in the Alps are weaker than in the other subregions. Still, the TOE is reached quite early in this region because the FWI and its variability are very low in the present climate period. Throughout the 21$^{st}$ century, the median and extreme FWI increase strongly. While the extreme FWI is projected to shift from low to moderate fire danger, the median FWI shows hardly any changes and remains in the no danger level (below five) according to EFFIS (2021).

For the other subregions, the median of the fire season is currently low but increases towards a moderate danger level in the future. In the extreme case, the average fire danger is moderate (11.2 < moderate > 21.3, s. table 1) in the present, but increases until the end of the century up to a high level (21.3 < high > 38, s. table 1) with values greater than 30 for the Southgerman Escarpment, slightly smaller than 30 for the Eastern Mountain Ranges and approximately 25 in the Alpine Foreland (figure 7). In general, increases in fire danger in the extreme FWI are of such a magnitude that the lower bounds of the ensemble standard

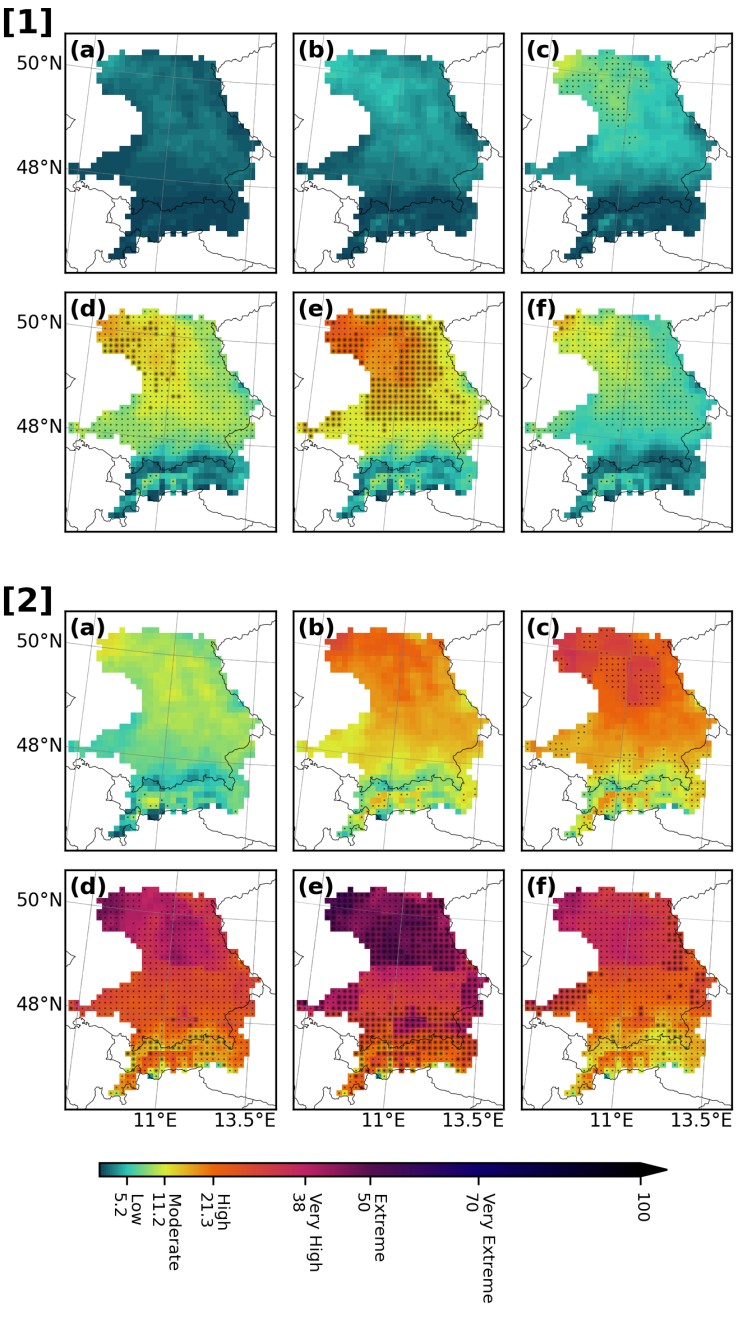

**Figure 5.** Ensemble mean of the median ([1], 50[th] percentile) and extreme FWI ([2], 90[th] percentile) by fire season month (April (a) - September (f)) for the future time period 2070–2099. Dots indicate that 90% of the CRCM5-LE members agree on a fire danger level increase of one (thin black dots) or two (thick black dots) levels compared to the present period (1980–2009)


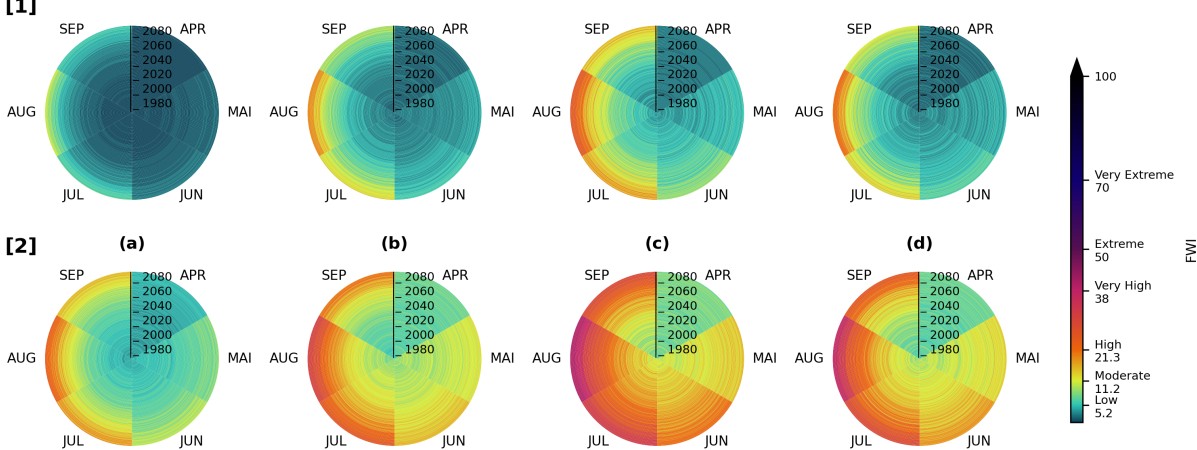

**Figure 6.** Fire rings show the ensemble mean of the monthly median ([1], 50[th] percentile) and extreme ([2], 90[th] percentile) FWI of each subregion (Alps, Alpine Foreland, Southgerman Escarpment and Eastern Mountain Ranges (a-d)) during the fire season (April - September) between 1980 and 2099

deviation exceeds the upper bounds of the standard deviation of the present climate period for all subregions by the end of the 21[st] century.

### 3.3 Increasing Frequency of Extreme Events

In the future (2070–2099), days with fire danger (>= low) shift from one out of ten to one out of three days in the Alps, one third
to half in the Alpine Foreland and Eastern Mountain Ranges, and one third to less than half in the Southgerman Escarpment
(s. figure 8). In the Alps, no and low fire danger days represent 182 out of 183 days of the fire season in the present. In the
future, this will drop to approximately 160 days of the 183 day long fire season. In the Alpine Foreland, moderate, high and
very high fire danger occur during one third of the days in the future fire season. The high and very high class are not observed
in the present climate period, but occur in the future. In the Eastern Mountain Ranges, similar results are observed: While the
higher danger levels already occur in the present, very high danger levels additionally occur in the future. In comparison to
the Alpine Foreland, moderate fire danger days are less frequent and high fire danger days are more frequent in the Eastern
Mountain Ranges. For the Southgerman Escarpment, changes from present to future resemble those in the Eastern Mountain
Ranges with the difference that the high and very high class are represented more frequently.

In all subregions, the frequency of distinct FWI extremes increases towards the end of the 21[st] century (figure 9). The
frequency of 100-, 50-, 20- and 10-year FWI extremes at least doubles until the end of the 21[st] century in all subregions (s.
figure (9). Generally, the results for the four subregions are quite similar and vary only slightly in details. In all subregions, the
return period of the present 100-year event will become the 50-year event in the early 2030ies, the 20-year event in the 2060ies

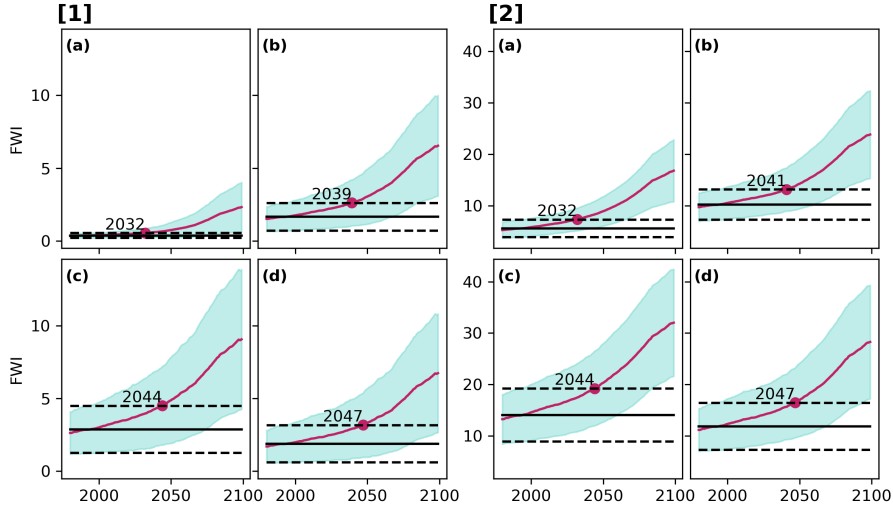

**Figure 7.** Trends of the median ([1], 50[th] percentile) and extreme ([2], 90[th] percentile) FWI between 1980 and 2099 differentiated by subregion: (a) Alps, (b) Alpine Foreland, (c) Southgerman Escarpment, (d) Eastern Mountain Ranges. The ensemble mean trend is derived on a fire season basis and represented by solid pink lines smoothed over a 30-year window. The ensemble's standard deviation is represented by shaded blue areas. Black solid and dashed lines represent the ensemble mean and spread of the present climate period (1980–2009). The TOE, marked with a pink dot and year annotation, is reached when the ensemble mean (pink line) crosses the upper boundary of the ensemble standard deviation in the present climate period (black dashed line).

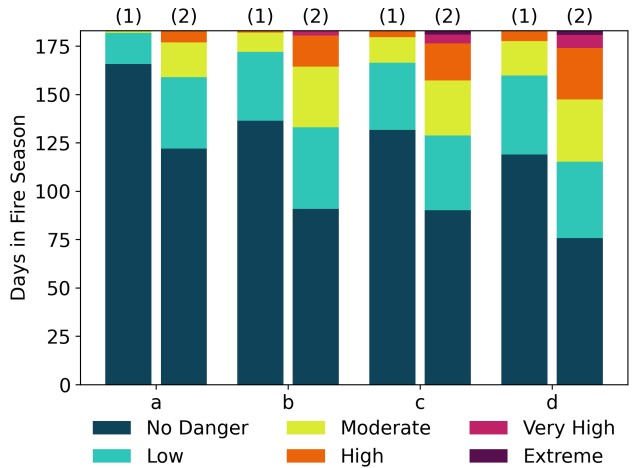

**Figure 8.** Frequency of fire danger levels in the fire season (April – September, 183 days) for the present (1; 1980–2009) and future (2; 2070–2099) climate period. FWI danger classes are derived for the subregions (a) Alps, (b) Alpine Foreland, (c) Southgerman Escarpment and (d) Eastern Mountain Ranges.


and the ten year event by 2090. The 10-year events of the present will occur every five years by 2060 and every three years by 2090.

Surprisingly, the spread of the return periods decreases in the future, indicating a stronger increase of very extreme events (i.e. 100- and 50- year events) than for mid-range extreme events (i.e. 20- and 10-year events). This finding is depicted by the increasing density and overlapping ensemble realisations for different return periods towards the end of the century. For example, the 100-year event of the Eastern Mountain Ranges becomes a 10-year event by 2090. If the increase of extreme events was proportional, we would expect the 10-year event to become a 1-year event at approximately the same time. This is

not the case. Instead, the 10-year event becomes a 3-year event by 2090 in the Eastern Mountain Ranges, implying changes in the FWI value distribution over time (s. figure 9).

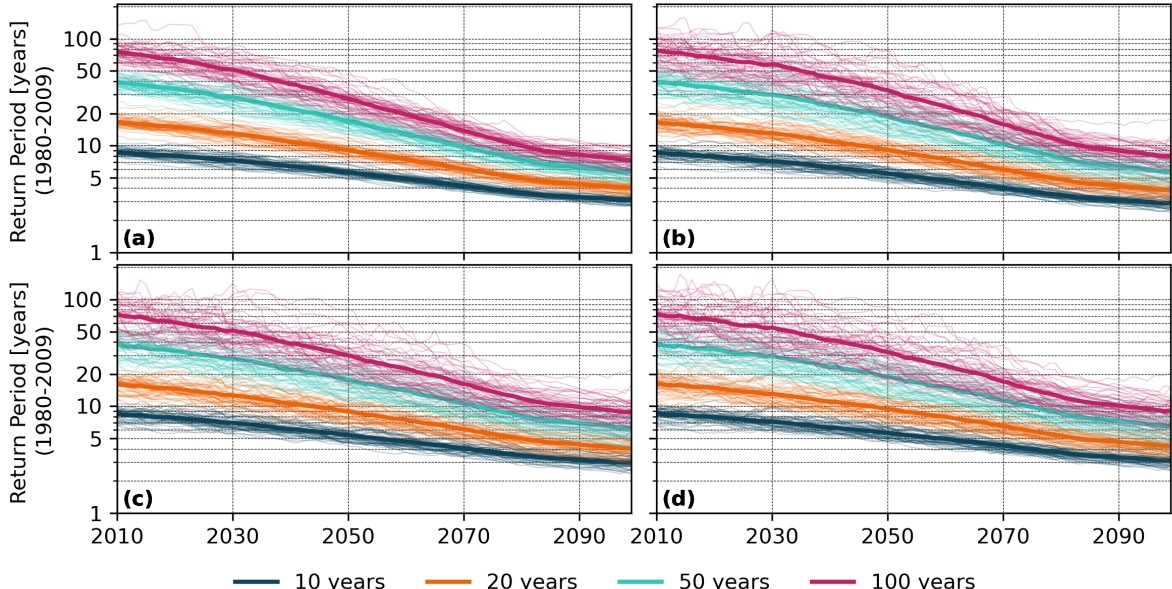

**Figure 9.** Changes in current return periods (1980–2009) of the 90th, 95th, 98th and 99th FWI percentile throughout the 21st century (2010–2099), distinguished by subregion: (a) Alps, (b) Alpine Foreland, (c) Southgerman Escarpment, (d) Eastern Mountain Ranges. The thick solid line represents the CRCM5-LE mean, while thin lines represent the ensemble members.

## 4    Discussion

Our results demonstrate that fire danger increases dramatically over the next few decades in Central Europe. The trend towards hazardous fire danger conditions in the future emerges for all presented metrics in this study, i.e. different temporal, spatial

and ensemble aggregation levels. First, the discussion highlights the relevant characteristics of the dataset used in this study. Second, we examine the limitations of the FWI and the fire danger levels. Third, we discuss the results in a spatio-temporal





context with a special focus on fire danger variability. Last, we elaborate on the societal and ecologic implications of our findings.

## 4.1 Data Basis

Before starting our analysis, we compared the results from the CRCM5-LE to the dataset of Vitolo et al. (2020) for the present climate period (1980–2009). The CRCM5-LE slightly exceeds the FWI in comparison to the reference dataset (s. figure 3), especially in the northwestern parts of the study area (s. figure 4). The runs with the CRCM5-LE are generated using the CFFDRS R package, whereas the reference dataset from Vitolo et al. (2020) uses the Global ECMWF Fire Forecast (GEFF) model (Di Giuseppe et al., 2016). These models differ slightly in their results, since the GEFF model applies the original

FWI formulas from Van Wagner and Pickett (1985), while Wang et al. (2017) adjusted the formulas for DC and DMC in the CFFDRS R-package (Vitolo et al., 2019). Nevertheless, the validation set-up demonstrates that the implemented algorithm generates meaningful results.

Another aspect, which has to be discussed, is the strong tiling pattern visible in figure 5 [2] in the months June and August. This tiling pattern is already visible in the extreme values of the input variables. We provide a sensitivity analysis of the

FWI in the Appendix (s. figure C1), where the tiling occurs for temperature and relative humidity in the 95[th] percentile as well. The pattern correlates with invariate fields from the geophysical baseline parameterisation of the CanESM2, e.g. bedrock depth. Over the areas where the strong tiling occurs, bedrock depth is about 5m. The water storage potential of the ground is especially high in this area compared to its surrounding areas with an average bedrock depth between 1 or 2 meters. Such high storage potential can affect evaporation and leads to a higher cooling in areas with high bedrock depths which results in lower

temperatures and higher relative humidity.

Lastly, the SMILE used in this study assesses climate change signals against internal climate variability but does not consider uncertainties related to emission scenarios and the chosen climate model. However, the choice of emissions scenarios also introduces uncertainty. Fire danger increase is projected and analyzed only for the RCP8.5 scenario, which represents the strongest temperature increase scenario. It remains open and subject to policy making if this scenario becomes reality. Arnell

et al. (2021) find that reducing emissions to a level consistent with an increase of a global mean temperature of 2°C, i.e. RCP 2.6, reduces fire danger substantially compared to RCP8.5. This finding implies that our change estimates represent an upper boundary of changes in fire danger expected in the future.

## 4.2 FWI and Fire Danger Levels

In this study, the statements on extremes are solely based on the percentile thresholds (e. g. 90[th] percentile for extremes) rather

than on individual extreme events. The generalization of the daily calculated FWI to long-term ensemble averages of single months cancels out single extreme events on a sub-monthly scale. Because the FWI is an indicator which identifies and rates dangerous conditions for fire events rather than predicts fire occurrences (Di Giuseppe et al., 2016), aggregation does not limit our findings but eliminates outliers of locally small or temporarily short but very extreme FWI values (FWI > 70). Fires start only in case of an ignition and the FWI as a danger rating index quantifies the ease of ignition, rate of spread and difficulty





to control a potential fire (De Rigo et al., 2017). The FWI cannot be analysed in terms of events, similar to other indices like the Percent of Normal Index for drought events (Böhnisch et al., 2021). It rather describes the potential for fire weather development and is therefore suitable to assess future changes in fire danger.

Further, the classification into danger levels proposed by EFFIS (2021) needs to be critically reflected upon: The FWI consists of four input variables, which implies multivariate inter-dependencies that lead to a high internal variability of the 320 index itself. Parts of our results are based on the EFFIS fire danger levels for the FWI (s. figure 8). First, class memberships are not linearly distributed over the index values within this classification scheme. For example, a FWI increase of 15 between a FWI of 10 and 25 rises the class membership from low to high and covers two classes, whereas the same difference between a FWI of 25 and 40 increases the fire danger level from high to very high. Second, the FWI itself is a complex scheme with exponential, rather than linear relationships (s. Drought Codes of the FWI; Van Wagner, 1987). Therefore, using classified 325 values to identify a robust signal normalizes the scale and offers a more interpretable approach to assess increases in fire danger. The CRCM5-LE helped us to show that the increase in fire danger is significant thanks to the ensemble agreement on the observed increases in fire danger classes.

While the FWI addresses fire danger in a meteorological context, it does not account for the flammability of the surface. For large-scale FWI analyses, non-burnable areas such as deserts and bare soil are masked out (Vitolo et al., 2020; Touma 330 et al., 2021). However, in the context of the study area HydBav and the 11-km resolution of the CRCM5-LE, land use is highly variable on a sub-pixel scale and non-burnable areas are therefore not masked out in this study (s. figure A1).

### 4.3 Spatio-Temporal Trends and Variability

We find that the region affected most strongly by FWI increases is the northwest, i. e. the Southgerman Escarpment (s. figures 5, 6 and 8). Noteworthy is, that average changes (median) are smaller in the Alps, but increases in the extreme FWI are 335 strongest in the Alps. The trends of the median are similar for the Alpine Foreland and the Eastern Mountain Ranges, but FWI extremes in the Eastern Mountain Ranges increase more strongly than in the Alpine Foreland. We summarize that increases in fire danger extremes are more pronounced than increases in median conditions and therefore variability increases in regions with heterogeneous terrain (Alps and Eastern Mountain Ranges). For less complex terrain (Alpine Foreland and Southgerman Escarpment), the increases in fire danger extremes are less variable. These findings corroborate findings by Wastl et al. (2012), 340 who explained the higher fire danger variability in mountain regions by the higher terrain variability, i.e. rain-shadow effects and katabatic dry winds (foehn).

Comparing the median and extreme conditions, derived from the $50^{th}$ and $90^{th}$ percentile gives insights into the dataset's variability and can differ by the chosen aggregation level, e. g. monthly or daily values. The differences between the percentiles are smaller for April, May and June, than for July, August and September (figure 5). This finding indicates that the seasonal 345 variability is higher for the last three months of the fire season and implies that the probability for extreme FWI conditions is elevated during these late summer months. The ring plots in figure 6 confirm this assumption for the defined subregions. This subregional analysis confirms the Southgerman Escarpment as the hotspot for dangerous FWI conditions within Hydrological Bavaria. Nevertheless, the other subregions are subject to tremendous changes as well. Especially the months August and July





can be identified as seasonal hotspots throughout the study area. On average (median), the fire danger will be high in the Alpine
Foreland, Southgerman Escarpment and Eastern Mountain Ranges and moderate in the Alps by the end of the century. For the
extreme FWI events, such high levels can already be observed by the middle of the century (figure 6).

The question arises whether the fire season considered in this study is too short, when looking at the differences between
median and extreme FWI results in the signal maps (figure 5), and the strong increase of the FWI in September in the ring
plots (figure 6). According to the results demonstrated in the ring plots (figure 6), the fire season in HydBav starts in May,
when the first dangerous FWI conditions (moderate fire risk) are reached for the extreme FWI sample. In the future, the fire
danger levels are still elevated and no longer on a no-danger level as in April for the median. This finding suggests that the fire
season length increases to at least October towards the end of the century. For the Southern Alps, Wastl et al. (2012) identified
the main fire season between December and April because of low precipitation and missing vegetation cover in the winter
half year. Therefore, future studies assessing changes in fire danger in the Alps should focus on the whole year instead of the
summer season only.

Additionally, we want to highlight the special characteristics of the Alps which are characterized by very complex terrain.
Due to their elevation-dependent colder climate, the mean FWI does not reach as dramatic values as in the other subregions.
Nevertheless, this region is very sensitive to climate change induced fire weather changes as demonstrated by its early TOE (s.
figure 7) and its significant danger level changes in the months July and August (s. figure 5). In the Alps, TOE is reached strik-
ingly earlier than in the other subregions, mainly because of small natural variability in the present climate period. This small
variability occurs because there exists currently no fire danger on this high data aggregation level in this specific subregion.

Besides the low variability of the FWI in the Alps, resulting in a very early TOE, we want to point out that our results
for the TOE and the projected FWI in the other subregions are similar to findings for France (Fargeon et al., 2020). Fargeon
et al. (2020) found FWI increases between two and twelve index values for the median (50[th] percentile) and from 15 to 22
index values for the extreme FWI (90[th] percentile) using a multi-model ensemble under the RCP8.5 scenario over France.
TOE is reached in both percentiles around 2060, which is about 20 years later than observed in the results of this study for
HydBav. Reasons for this delay could be due to the later and shorter reference period (1995–2015), the overestimation of natural
variability in the multi-model ensemble (Fargeon et al., 2020) or the slight overestimation of the CRCM5-LE (s. chapter 2.4).
The CRCM5-LE used in our study embodies a substantially larger database than the database used by Fargeon et al. (2020)
thanks to its SMILE-setup, which helps to better represent natural variability. While Fargeon et al. (2020) point out that fire
danger increases are hard to distinguish from natural variability in northern France in multi-model ensembles, we demonstrate
using a SMILE that increases in fire danger are robust for Central Europe.

### 4.4 Societal and Ecologic Impacts

Our results highlight the increasing frequency of currently anomalously extreme fire weather that will affect the study regions'
fire regime as well. Prolonged droughts and exacerbating heat events may limit fuel availability and therefore fire activity in
more arid regions, such as the Mediterranean (Bowman et al., 2020; Pausas and Paula, 2012). However, for wetter, more pro-
ductive regions and seasons, i.e. our study area in Central Europe, aridity does not limit fuel availability, which implies higher





sensitivity to flammable conditions (e.g., after hot and dry seasons) and points out the importance of considering vegetation and fuel structure changes in further studies (Pausas and Paula, 2012; Turco et al., 2018). Further, Bowman et al. (2020) suggest
that declining snow cover in spring and drier fuels in summer will increase burned area in mountain forests, as present in the Alps and Eastern Mountain Ranges in our study area.

For the Mediterranean, Turco et al. (2018) expect changes in meteorological fire weather of such a magnitude, that current fire suppression measures are not sufficient anymore. The guidelines for forest fire defence in the federal state of Bavaria currently only ask the public for cautious behaviour when fire danger is elevated. In case of high or very high fire danger,
surveillance flights are carried out in the respective areas (StMLF, 2013). Studies in other regions, i.e. the UK (Arnell et al., 2021) and France (Fargeon et al., 2020), suggest that increases in fire danger should be considered in emergency, land use and management planning to mitigate future fire risk. Taking the results of our study into account, these suggestions apply for Hydrological Bavaria as well.

## 5    Conclusion

This study presents the first regional Single Model Initial-Condition Large Ensemble (SMILE) assessment of fire danger increase for Central Europe, more specifically, the study area Hydrological Bavaria (HydBav). The study area is not yet affected by wildfires and high fire danger to date, but will be affected in the future when assuming an RCP8.5 emission scenario and accounting for natural variability. The strongest increases and most hazardous developments are observed North of the river Danube in the summer months July and August for the subregions South German Escarpment and Eastern Mountain Ranges.
Regions south of the Danube (Alps and Alpine Foreland), are less strongly affected by changes in the FWI but increases are still significant. Further, we find that the FWI has a stronger variability for regions with heterogeneous terrain (i.e. the Alps and the Eastern Mountain Ranges) than for regions with less complex terrain (i. e. Alpine Foreland and Southgerman Escarpment). The time of emergence (TOE) is reached in all subregions of the study area before 2050 and the return period of a present 100-year event shifts towards a 10-year event by 2090. We accept all of our three hypotheses, stated in the introduction (chapter 1).
Our results reveal more serious developments than assumed in the original hypotheses.

This study highlights fire danger increases for Central Europe, as an example region with currently moderate to low fire danger conditions. Our findings stress the importance of developing fire suppression measures to mitigate future fire risk also in regions with temperate climate.

*Data availability.*  The datasets used in this study can be found in the following repositories: CRCM5-LE: https://www.climex-project.org/
de/datenzugang and ERA-5 based FWI: DOI: 10.24381/cds.0e89c522(31.01.2023)



*Author contributions.* JM, AB, RL and MIB contributed to the conception of the study. JM conducted the data collection for the FWI, statistical analyses and wrote the first version of the manuscript. AB, RL and MIB monitored and supported the research process and revised the manuscript. RL is founder and head of the ClimEx project. All authors contributed to the article and approved the submitted version.

*Competing interests.* The contact author has declared that neither of the authors has any competing interests.

*Acknowledgements.* This research was conducted within the ClimEx project (https://www.climex-project.org/), funded by the Bavarian State Ministry for the Environment and Consumer Protection (Grant No. 81-0270-024570/2015). CRCM5 was developed by the ESCER Centre of Université du Québec 'a Montréal (UQAM; https:// escer.uqam.ca/) in collaboration with Environment and Climate Change Canada. Computations with the CRCM5 for the ClimEx project were performed on the HPC system SuperMUC and the Linux Cluster of the Leibniz Supercomputing Center LRZ, Bavarian Academy of Sciences and the Humanities (BAdW), funded via GCS by BMBF and StMWFK.



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



**Appendix A**



**Figure A1.** Land cover distribution in Hydrological Bavaria (modified, CLMS (2021))

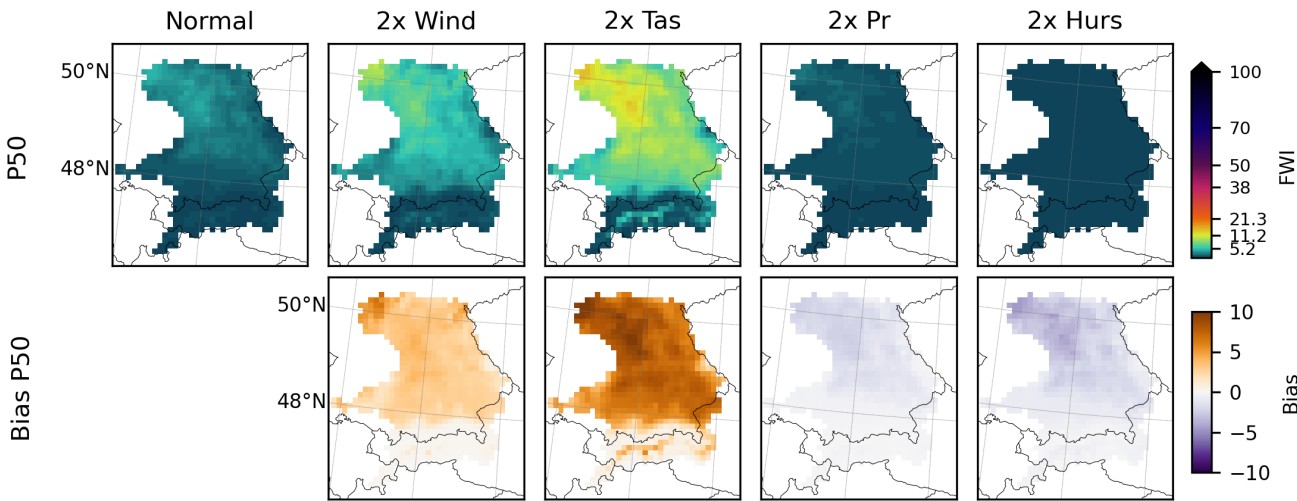

**Figure B1.** FWI median (top, $50_{th}$ percentile) and bias (bottom) of sensitivity runs, where each input variable is increased by a factor two (e. g. 2x wind) and original FWI run (Normal) in the validation period 1981 to 2010. Bias is calculated by subtracting each increased sensitivity run from the original FWI run.

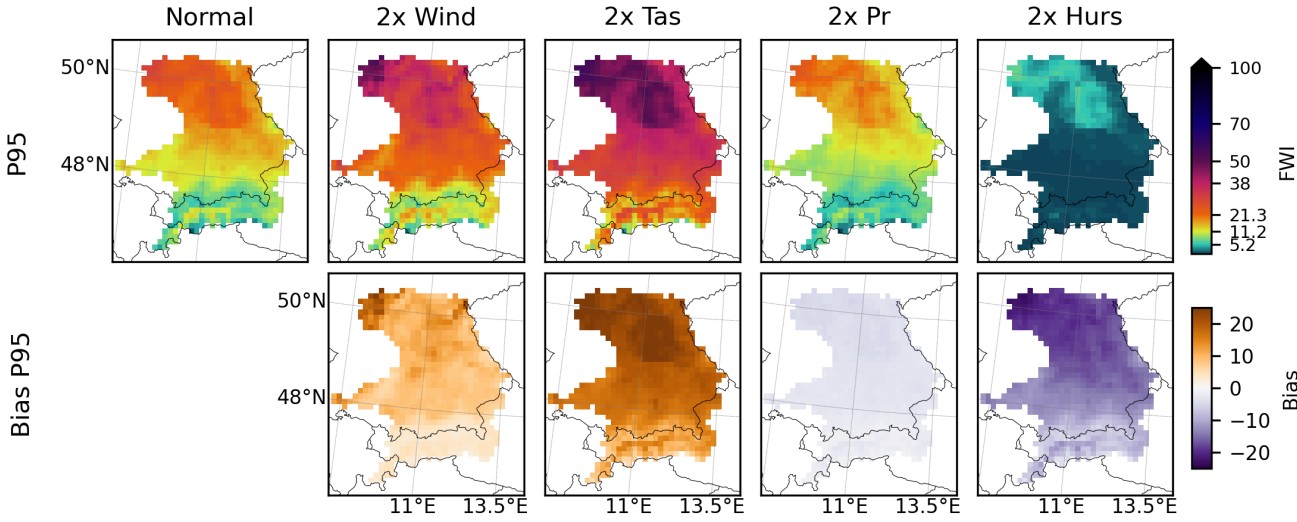

**Figure C1.** FWI extreme (top, $95_{th}$ percentile) and bias (bottom) of sensitivity runs, where each input variable is increased by a factor of two (e. g. 2x wind) and original FWI run (Normal) in the validation period 1981 to 2010. Bias is calculated by subtracting each increased sensitivity run from the original FWI run.