# Peer review of "Climate change impacts on regional fire weather in heterogeneous landscapes of central Europe"

_Natural Hazards and Earth System Sciences, 2023_

## Referee Comment (RC3)

**Review of the paper (NHESS-2023-51) entitled** "Climate change impacts on regional fire weather in heterogeneous landscapes of Central Europe".

**Authors**: Julia Miller, Andrea Böhnisch, Ralf Ludwig, and Manuela I. Brunner

**Recommendation for publication:**

Acceptation after (pending on) minor revisions before publication, with few clarification and addition to be made in the current version of the article.

**General comments**: This paper investigates a regional SMILE (Single Model Initial-Condition Large Ensembles) of the Canadian regional climate model version 5 (CRCM5-LE) over Central Europe (Hydrological Bavaria) under the RCP8.5 scenario from 1980 to 2099, to analyze fire danger trends in a currently not fire-prone area. This evaluation of fire danger (vs current climatic conditions) uses Canadian Fire Weather Index (FWI), and the 3-hourly meteorological data from the large ensemble of available CRCM5-LE simulations. The authors demonstrate that this ensemble (at 0.11°) is a suitable dataset to disentangle climate trends from natural variability in a multivariate fire danger metric. Various results show the increase in the median and extreme percentile of the FWI in the northern parts of the study area (in July and August). The southern parts of the study region are less strongly affected, but time of emergence (TOE) is reached there in the early 2040's. In the northern parts, the climate change trend exceeds natural variability in the late 2040's. In the future, a 100 year (return period) FWI event will occur every 30 years by 2050 and every 10 years by 2099. This study is of a strong interest in order to help the refinement of fire management strategy to reduce the consequences of such forest fires, and to improve the preparation or adaptive capacity knowing the potential changes of this natural hazard under on-going climate change.

The article is well written, and well-articulated in term of scientific findings and presentation of main outcomes. I will suggest to add insights or discuss limitations from the use of one single RCM driven by one specific GCM (i.e. CanESM2) whatever the number of ensemble runs used, as systematic biases from the driving GCM can influence the downscaling simulations and derived products (ex. FWI). For example, as noted in various studies, biased atmospheric circulation features due to coarse-scale resolution (ex. around 2.8° for the CanESM2 model) and/or missing orographic drag, sea surface temperature simulated features, etc. which affect the simulated blocking features (see Pithan et al., 2016; Schiemann et al., 2017; Davini and d'Andrea, 2020) or atmospheric circulation variability responsible for the occurrence of climate extremes over Europe (see Faranda et al., 2023). As revealed in the recent work of Faranda et al. (2023), atmospheric circulation changes modulate extreme events already in the present climate in Europe, and summer heatwaves as well as large regional and seasonal changes in precipitation and surface wind, i.e. hazards or meteorological variables responsible for the occurrence and severity of fire danger (variables used to compute the FWI indices). Also, as shown in Zappa et al. (2014), CanESM2 tends to have one of the largest track density biases for extratropical cyclones among CMIP5-GCMs, as well as in blocking frequency biases over both Norwegian Sea, and central Europe (see their Figure 3). These two features of atmospheric circulation variability play a key role in the occurrence of both anomalies of temperature and precipitation across the study

area. Faranda et al., (2023) and Strommen et al. (2019) strongly argue to use at least three or more ensemble members to deal with the importance of the regional response to anthropogenic forcing (Corti et al., 1999; Palmer, 1999), representing these atmospheric regimes correctly whatever the GCM. Also as noted in Deser et al. (2020), the use (in future works) of large ensembles from different GCMs will give new insights into uncertainties due to internal variability versus model differences. In fact, concerning the so-call internal variability (or natural variability mentioned in the paper that it is partly evaluated from the CRCM5-LE), the current study cannot consider using one single ensemble from one RCM-GCM matrix, structural variability from the differences in (GCM) model formulation, including physics and parameterization, resolution, etc. This structural variability strongly affects the climate change responses at the regional or local scale (uncertainties in the dynamical downscaling simulations depend on the driving GCM; see different studies from the EURO-CORDEX project).

In summary, after having recall the shortcoming and include nuance about the robustness of the climate change signals for the FWI indices across the area, using one single ensemble sample from one combination of RCM-GCM, the paper is sufficiently relevant and scientific documented (sound) to be published after minor revisions. Please see also my specific comments below.

**Specific comments**:

Please be consistent when you use RCP8.5 (without "space" between P and 8.5) in the text.

Abstract:

Please add few words or one sentence considering the need to use larger downscaling ensemble from different GCMs in order to develop more robust climate change signals for all meteorological variables used to compute the FWI indices (further work).

Introduction:

**Line 22:** Please add (Canada) after British Columbia.

**Line 63:** Please nuance this statement, as natural variability of the climate system is not fully represented by one single model initial-condition large ensemble, as a GCM generates a simplification of a complex reality (i.e. the climate system) and includes structural variability and biases that we need to consider in any downscaling exercise (see Strommen et al., 2019; Deser et al., 2020; and recommendations or Plausibility criteria in the new CORDEX-CMIP6 in Sobolowski et al., 2023).

Data and Methods

**Line 96:** As mentioned in Fargeon et al. (2020) and many other studies, bias correction alters the physical consistency of modeled climate and meteorological variables in particular at high frequency (ex. sub-daily values). Quantile mapping makes strong assumptions regarding bias stationarity and can break the co-variation between climatic variables, in particular at high

frequency or meteorological scale (i.e. that is the case here when computing the daily FWI indices). Can the authors provide some insight about these drawbacks or physical consistency among meteorological variables after bias correction and the implication of this in computing FWI indices ?

**Line 100**: "*FWI extremes are significantly better…*": Yes, but these FWI extremes are physically coherent and consistent with meteorological fields ?

**Study area**

**Line 112**: Please can the authors provide some reference from which dataset these (climatological) values come from ? E-Obs, …?

**The Canadian Fire Weather Index**

**Line 125**: Please correct "… asses**s**…".

**Estimating Fire Danger using the CRCM5-LE**

**Line 170**: "…, *this does affect the climate change impacts assessment…*". Yes, but the CRCM5 ensemble seems to underestimate the internannual anomalies of FWI that we see from the reference (ERA5) database. Please can you comment this, as the year 2002 seems to be not in the range of below 75$^{th}$ percentiles of the observed FWI across Europe but rather on more extreme side? This underestimation of interannual FWI anomaly can be due to the debiased method which has an effect on the decreasing year to year variability of each of the ensemble simulations?

**Discussion**

**Line 278:** "… *next few decades…*". This mean 2080s? Please be precise.

**Data basis**

**Line 302**: "…*uncertainties related to emission scenarios and the chosen climate model*". You do not discuss this point (i.e. choice RCM or single RCM-GCM), please provide some insights as suggested in the general comments.

**Spatio-Temporal Trends and Variability**

**Line 359**: "… *on the whole year instead of the summer season only*": Potential avenue will be to use take into account the snow cover season or overwintering conditions, based on cumulative precipitation during the cold season, as used in Canada (see McElhinny et al., 2020).

**Line 373**: "*…or the slight overestimation of the CRCM5-LE*…": Again, this can be due to the lack or limited internannual variability in the debiased CRCM5-LE variables? Please comment slightly on this issue.

**Line 374**: " … *a substantial larger database*…". Yes, but this is a single model (CRCM5) driven by an ensemble of one GCM (CanESM2), as in Fargeon et al. (2020) they use 2 RCMs driven by 3 different GCMs. Please nuance this statement.

**Line 375**: "… *which helps to better represent natural variability*": See my previous remarks, natural variability is more complex that internal variability extracted from one single RCM-GCM matrix, as at least you need to consider more range of boundary conditions, from as many as possible GCMs as those are the main source of uncertainties in particular from the atmospheric circulation over Europe pointed out by Faranda et al. (2023).

Line 377: " … *fire danger are robust*…": From the ensemble runs used (i.e. link to the sample size or RCM-GCM matrix). Please nuance this statement.

Conclusion

**Line 404**: "*We accept all of the three hypotheses*…": Please be more explicit and comment about these, in particular H2 and H3.

References

**Line 531**: The reference Separovic et al. (2013) is not at the right place in the list.

**References**:

Corti S., F. Molteni, T. N. Palmer (1999). Signature of recent climate change in frequencies of natural atmospheric circulation regimes. Nature 398, 799–802

Davini P, D'Andrea F (2020). From CMIP3 to CMIP6: Northern Hemisphere Atmospheric Blocking Simulation in Present and Future Climate. J Clim 33(23):10021-10038.

Deser, C., Lehner, F., Rodgers, K.B. et al. Insights from Earth system model initial-condition large ensembles and future prospects. Nat. Clim. Chang. 10, 277–286 (2020). https://doi.org/10.1038/s41558-020-0731-2

Faranda et al. (2023). Atmospheric circulation compounds anthropogenic warming and impacts of climate extremes in Europe, PNAS, https://www.pnas.org/doi/suppl/10.1073/pnas.2214525120

Fargeon, H., Pimont, F., Martin-StPaul, N., De Caceres, M., Ruffault, J., Barbero, R., and Dupuy, J.-L. (2020). Projections of fire danger under climate change over France: where do the greatest uncertainties lie?, Climatic Change, 160, 479–493. https://doi.org/https://doi.org/10.1007/s10584-019-02629-w

McElhinny, M, Justin F. Beckers, Chelene Hanes, Mike Flannigan, and Piyush Jain, (2020). A high-resolution reanalysis of global fire weather from 1979 to 2018 – overwintering the Drought Code, Earth Syst. Sci. Data, 12, 1823–1833.

Palmer, T. N. (1999). A nonlinear dynamical perspective on climate prediction. Journal of Climate, 12(2), 575–591. https://doi.org/10.1175/1520-0442(1999)012<0575:ANDPOC>2.0.CO;2

Pithan, F., T. G. Shepherd, G. Zappa, b and I. Sandu (2016). Climate model biases in jet streams, blocking and storm tracks resulting from missing orographic drag, Geophys. Res. Lett., 43, 7231–7240, doi:10.1002/2016GL069551.

Pfahl S, Wernli H (2012). Quantifying the relevance of atmospheric blocking for co-located temperature extremes in the Northern Hemisphere on (sub-)daily time scales. Geophys Res Lett 39:L12807.

Schiemann, R., et al. (2017). The resolution sensitivity of Northern Hemisphere blocking in four 25-km atmospheric global circulation models. J. Climate, 30, 337–358, https://doi.org/10.1175/JCLI-D-16-0100.1.

Sobolowski et al. (2023). EURO-CORDEX CMIP6 GCM Selection & Ensemble Design: Best Practices and Recommendations, https://zenodo.org/record/7673400#.ZAWmhezMLP8

Strommen, K., Mavilia, I., Corti, S., Matsueda, M., Davini, P., von Hardenberg, J., et al. (2019). The sensitivity of Euro-Atlantic regimes to model horizontal resolution. Geophysical Research Letters, 46, 7810–7818. https://doi.org/10.1029/2019GL082843

Zappa, G., G. Masato, L. Shaffrey, T. Woollings, and K. Hodges (2014). Linking Northern Hemisphere blocking and storm track biases in the CMIP5 climate models, Geophys. Res. Lett., 41, 135–139, doi:10.1002/2013GL058480.

---

## Author Comment (AC1)

**Reviewer 1:**

The manuscript "Climate change impacts on regional fire weather in heterogeneous landscapes of Central Europe" by Miller et al. presents a study on climate change impacts on the fire danger index FWI in hydrological Bavaria in central Europe, using a single model initial-condition large ensemble (SMILE) data set and the RCP 8.5 emission scenario. Changes in FWI are evaluated in terms of fire danger levels over the whole region, as well as on a sub-regional scale. The study provides new and interesting knowledge of projected changes in FWI in the studied region, and the topic is suitable for the journal. Strengths of the manuscript include the multiple approaches applied to investigate the hypotheses, and the clearly communicating figures. The manuscript has potential but is not at the required level for a scientific paper in its current state. Parts of the analyses are wrong, methods are not clearly presented, the discussion is not clearly presented, and references do not always reflect work supporting the claims. I would recommend all authors to carefully revise the manuscript and correct and clarify where necessary. General and specific comments are provided below.

Thank you very much for taking the time to write this very detailed and constructive review, which we highly appreciate. We are glad that you value our work and are happy to take your comprehensive feedback into account. We think your comments shape the manuscript in a very positive way. We revised the paper carefully according to your feedback. We clarified unclear descriptions in the methods section, revised and updated references, and edited and extended the discussion. In addition, we carefully checked the manuscript for clarity.

*General comments*

1. The method to derive the return period is wrong. Thus, the analysis using return periods must be omitted or corrected and clarified. The applied temporal resolution is not stated, but the method is wrong regardless of the applied resolution. If all daily (or monthly) data in each year is used to extract the 99$^{th}$ percentile value, this value represent the 100 days (or 100 months) return period and not 100 years as stated in the manuscript. See e. g. Camuffo et al. (2020): https://doi.org/10.1007/s11600-020-00452-x. If instead the maximum value in each year was extracted, the period (30 years, i.e. 30 values) is too short to extract the 99$^{th.}$

   Thank you for highlighting the need to clarify the method used for return period calculation. We used a Single Model Initial-Condition Large Ensemble, where each of the 50 realizations represents an equally likely climate within the used climate model. Therefore, we pool over the entire 50-member ensemble to derive the percentiles from the resulting 274 500 data points (183 days per fire season x 30 years of climate period x 50 ensembles) for the present climate period from the data's empirical distribution. The future time periods (stepwise 30-year windows) were pooled over each member separately, resulting in (183 days x 30 years) 5490 data points from which the percentiles were derived from using the empirical distribution function of the ensemble member. Therefore, our sample size is not too short to extract the 99$^{th}$ percentile.

   We added a sentence to the description in chapter 2.5.3 to describe this more accurately: *"We pool over the entire 50-member ensemble using daily FWI values (183 days per fire season x 30 year climate period x 50 members). From this data pool we create an empirical distribution function of which we derive the percentiles representing the 10-, 20-, 50- and 100-year FWI return levels for the present climate period (1980-2009)".*

   We modified the sentence "*from 2010 to 2099, we create centered 30-year windows of our data sample and determine the FWI percentiles corresponding to the different return periods of the*

*present climate period"* to *"from 2010 to 2099, we create centered 30-year windows for each member to determine the empirical distribution function and the FWI percentiles corresponding to the different return periods of the present climate period"*.

We edited the remark on how we map the probabilities of the return periods:

Old: *We then compute the non-exceedance probability of the present percentiles given the future cumulative distribution. From the future non-exceedance probability, we estimate the future return periods using the function T = 1/(1 − p) (1) where T is the return period and p is the non-exceedance probability (Brunner et al., 2021; Coles, 2001)*

New: *We map the non-exceedance probability of the present percentiles given the empirically derived cumulative distribution of each member. From the non-exceedance probability, we estimate the return periods using the function $T = μ/(1 − p)$ where T is the return period, μ is the inter-arrival time (1/183 days in a fire season) and p is the non-exceedance probability (Coles, 2001). We derive p from the rank r with $p = r/n$ where n is the total sample size by using the rv_histogram.cdf function of the Scipy package in Python (Virtanen et al., 2020).*

Lastly, we added a remark that we cropped the dataset to the time period where full 30-year windows are available (see RC-2 comment on L208).

New: *Due to the centered window approach, the first full 30-year window is 1995 and the last full 30-year window is 2084. Therefore, we crop the resulting time series to 1995 to 2084.*

2. Several references do not represent the original work reflecting your statements. Examples are line 46, 126, 157, 181 and 212. Please make sure the original references are used throughout, or in cases where this is not possible, add "e.g." before the references to avoid the reader to believe the reference is the original work.

   Thank you for pointing out the need to focus on original references. We adjusted:

   - Line 46: added "e. g. " to Bakke et al. (2023)
   - Line 126: added "i. e." to Di Giuseppe et al. (2016) and Touma et al. (2021)
   - Line 157: We modified the reference of Vitolo et al. (2019) to a direct quote:
     - Old: (Vitolo et al. 2009)
     - New: as suggested by Vitolo et al. (2009)
   - Line 181: We added Pfeifer et al. (2015) to the reference Böhnisch et al. (2021), because Böhnisch et al. (2021) adjusted the approach from Pfeifer et al. (2015).

     New: *... using the approach of Böhnisch et al. (2021) after Pfeifer et al. (2015).*

   - Line 212: We dropped Brunner et al. (2021).

   We further adjusted all sentences, which were also highlighted in your special comments and which we found during the review process. For those changes, please see the corresponding replies further below and check the new manuscript.

3. The discussion comprises multiple detailed comments on different aspects of the analysis or results, with too general subtitles. Introduced topics (e. g., uncertainties related to the chosen climate model on line 302) and summaries of results are not always followed up. Overall, the current state of the discussion chapter makes it hard for the reader to know what to expect and to follow the line of arguments of the authors. Please revise and clarify the discussion chapter,

avoid mentioning topics without commenting on them in relation to your study, and lift part of the discussion to a more general level.

Thank you for paying attention to the consistency of our discussion section. We agree that certain subtitles do not match the provided discussion points. We revised the discussion section carefully, changed subtitles, critically reflected on the chosen climate model setup and dropped paragraphs which were unrelated to the results. Further RC-3 commented, that uncertainties from the chosen climate model (structural uncertainty) and bias correction should be discussed, which we added to the discussion (see responses to RC-3). We renamed "Data Basis" to 'Uncertainties' in the discussion section. Changes in the other sections of the discussion referring to specific paragraphs of the manuscript are revised in the corresponding comments (RC-1 comments 66. to 86. and RC-3 Discussion)

Old subtitles:
*4.1 Data Basis*
*4.2 FWI and Fire Danger Levels*
*4.3 Spatio-Temporal Trends and Variability*
*4.4 Societal and Ecologic Impacts*

New subtitles:
*4.1. Uncertainties*
*4.2 FWI and Fire Danger Levels*
*4.3 Spatio-Temporal Trends and Variability*
*4.4 Regional Shifts and Implications*

4. The text is in several places informal with the use of unnecessary introductions (e. g. "another aspect which has to be discussed" on line 283 or "needs to be critically reflected upon" on line 318) or subjective words, inconsistent used of concepts, and imprecise descriptions of methods and results. Further, most of the manuscript is written in present tense. Papers are usually written in past tense when presenting analysis and results in abstract, data and methods, results and conclusions. Thus, I would recommend changing to past tense. In general, please carefully revise the whole manuscript and clean and clarify the text.

Thank you for raising stylistic concerns regarding the writing style of the paper. We eliminated subjective words to our best knowledge and iterated over terminologies to ensure consistency. Further, we tried to avoid filling words as much as possible but kept them in places where they are necessary for a fluent reading flow. We changed to past tense in the introduction and data and methods sections. However, we kept the results section and parts of the discussion section in present because our results are here presented for the first time.

5. The manuscript refers to similar analysis in France and UK, but does not refer to results over the same region (HydBav) by others, e.g. from global or regional studies that cover the region. Please include other studies that cover your region (for example https://doi.org/10.1029/2018GL080959 and https://doi.org/10.1007/s10584-016-1661-x). In addition, please comment on potential differences between France, UK and your region, when you refer to results over these regions.

Thank you for highlighting these global and regional studies. We incorporated comparisons from regional studies (de Rigo et al. 2017 and Carnicer et al. 2022) over Europe to better reflect on the spatial differences between regions in Central Europe and how they differ from southern Europe, e. g. the Mediterranean.

*Specific comments:*

1. You state in the title that your study area represents "heterogeneous landscapes". However, in line 110-111 you "assume that the water availability, climatology and landscape of different river systems reflect the fire regime of an area". A more in-depth reflection on the heterogeneity of the regions, and even subregions of your study, in relation to your spatial aggregations and findings would be beneficial to better reflect the title of your study.

   Thank you for highlighting the need to clarify that with "heterogenous landscapes" we refer to heterogeneity across subregions and not within regions. A more detailed reflection of the differences among the subregions is given by the paragraph starting in Line 129 (of the updated manuscript) and underlined by Figure 1.

   Old: *Since fire is closely related to the availability, or rather the absence of water, we assume that the water availability, climatology and landscape of different river systems reflect the fire regime of an area.*

   New: *Since fire is closely related to the availability, or rather the absence of water (in terms of precipitation or soil moisture deficit), we assume that the water availability, climatology, and landscape characteristics of the four different complex landscapes selected in our study are reflected in the subregions specific fire regimes.*

2. «Central Europe» is a concept that refers to a considerably larger region than the study area "Hydrological Bavaria". Please clarify the region to avoid exaggerating your study domain, in particular in the Abstract (e.g., line 7 changing "over Central Europe" to "in a region in central Europe" or similar), the conclusion (e.g. line 395 changing "for Central Europe" to "in central Europe" or similar), and the title ("of Central Europe" to "in central Europe"). Specifically, change to lower case 'c' ("central Europe") throughout.

   Thank you for pointing out the need to be more specific about the region in central Europe we are focusing on. We revised the manuscript according to your suggestion to "a region in central Europe". We agree to write central Europe with small c in central, because our study domain focuses primarily on the mid-latitudes of Europe, rather than the politically defined Central Europe (countries of Germany, Austria, Poland, Czech Republic, Slovakia, Switzerland, Hungary, Slovenia).

   We revised the term in:

   - Line 7 from "*over Central Europe*" to "*over a region in central Europe*"

   - Line 17ff. from "*Our results highlight central Europe's potential for severe fire events from a meteorological perspective and the need for fire management in the near future even in temperate regions*" to "*Our results highlight the potential for severe fire events in multiple regions of central Europe from a meteorological perspective and demonstrate the need for fire management in the near future even in temperate regions.*"

3. Titles: please use NHESS house rules (sentence-style capitalization: https://www.natural-hazards-and-earth-system-sciences.net/submission.html#manuscriptcomposition)

We checked the titles and subtitles and adjusted according to the NHESS house rule "titles and headings follow sentence-style capitalization (i.e. first word and proper nouns only). This applies to table and figure headings as well."

We adjusted heading *Increasing Frequency of Extreme Events* to *Increasing frequency of extreme events.*

4. Line 27 and line 396-397: You state that Central Europe has not been exposed to wildfires before recent years (line 27) or to date (line 396-397). However, central Europe has been exposed to multiple wildfires at least the past three decades. Rephrase to correct statements and provide reference(s) that have fire records underlying your statements.

   We are aware that fires occurred also in the past three decades. Indeed, this sentence does not contain the message we want to communicate. Therefore, we changed the sentence:

   Old: *While the Mediterranean region and the Western US are historically fire prone areas, Central Europe showed exposure to wildfires only in the recent years, e.g. in Treuenbritzen 2022, Brandenburg, Germany (Spiegel, 2022), and Küps 2022, Bavaria, Germany (BR, 2022).*

   New: *While the Mediterranean region and the Western US are historically fire prone areas and have been well studied on a larger regional scale (i. e. Barbero et al., 2015, 2020; Abatzoglou et al., 2021; Ruffault et al., 2020), fire occurrences and risks in the temperate climates of Europe have been studied rather on a national than on a regional level (i. e. Arnell et al., 2021; Fargeon et al., 2020; Bakke et al., 2023).*

5. Line 38-40: The claim that fire indices represent a statistical correlation between fire events and meteorological conditions is wrong. Please correct.

   We changed "*represent the statistical correlation between fire events and meteorological conditions*" to "*are statistical models build on the correlation between fire events and meteorological conditions*".

6. Line 40-41: Please provide reference that "They have been proven to produce reliable ratings of fire danger in short- and long-term weather predictions on a global scale".

   We added Di Giuseppe et al. (2016) as reference.

7. Line 42: please rephrase "do not guarantee", as this is an unclear statement. Fire indices have nothing to do with ignition at all.

   Thank you for your remark, which refers to only one part of the sentence. We rephrased *"do not guarantee"* to *"do not incorporate".*

8. Line 45 and other: Various concepts are used for the fire indices; please be consistent and potentially introduce relevant relations between concepts such as "fire risk", "fire weather", "fire danger", "fire indices", "likelihood of fire" and "probability of fire".

   Thank you for pointing out these inconsistencies in terminology. We changed all occurrences of *"fire danger", "fire risk" and "probability of fire"* to "*fire danger*". "Fire weather" and "fire indices" are kept since they are not directly related to fire danger and are terms on their own. "Likelihood of fire" is not used in our manuscript. It is used as "likelihood of fire events" and cannot be replaced with "fire danger".

9. Line 49, 57 and 59: Central Europe (line 49) and temperate climate (line 58 and 59) refer only to studies of England and France here. As these concepts are used for the HydBav later, please clarify the use and links between geographical regions and climate regions. Are results for England and France directly transferrable to HydBav?

We appreciate your comment regarding the climate comparability of our study region with other studies. HydBav, France and England all belong to the same climate zone according to the Köppen and Geiger classification' (Rubel et al. 2017 & Rubel et al. 2010). Therefore, we think that results for England, France, and HydBav can be compared and that the three regions can be jointly addressed under the umbrella of 'Central Europe and temperate climate zone.

We changed Line 49 from "*In Central Europe, trends related to fire danger are uncertain and not clearly distinguishable from natural variability*" to "*In temperate climate regions, such as central and western Europe, trends related to fire danger are uncertain and not clearly distinguishable from natural variability*" to clarify the statement.

10. Line 49-50: Please provide reference that trends are not distinguishable from natural variability in Central Europe. Further, this sentence relates to the weak trend signal relative to natural variability independent of how models represent the natural variability, and thus the link to the next sentence is wrong or not clear.

We refer to the weak trend signal relative to natural variability. However, in multi-model studies, uncertainty is composed of two factors: (1) natural variability and (2) model uncertainty. Our linkage to the next sentences highlights the need for using modelling approaches representing natural variability, i. e. Single Model Initial-Condition Large Ensembles, as used in our study.

We clarified our statement:

Old: "*In Central Europe, trends related to fire danger are uncertain and not clearly distinguishable from natural variability. Arnell et al. (2021) and Fargeon et al. (2020) have shown for England and France, respectively, that this uncertainty originates from an under-representation of natural variability in climate multimodel ensembles.*"

New: "*In temperate climate regions of central Europe, i.e. northern France and the UK, trends related to fire danger are uncertain and not clearly distinguishable from natural variability when multi-model climate ensembles are used (i. e. Fargeon et al., 2020; Arnell et al., 2021). Arnell et al. (2021) and Fargeon et al. (2020) have shown for England and France, respectively, that this uncertainty originates from the confusion of natural variability with structural uncertainty originating from the different climate models in the ensemble (model uncertainty after Hawkins and Sutton (2009)).*"

11. Line 52-53: Unclear sentence "In France, …" Please rephrase.

We agree this sentence is unclear. We meant to say by "the exceedance of the fire danger signal decreases from South to North", that the fire danger signal in the South of France exceeds earlier the boundaries of inter-annual variability than in the North of France. However, we think this sentence does not add any significant additional value to our introduction and removed it.

12. Line 56: clarify the meaning of "natural variability of changes".

Thank you for highlighting the need for clarification. We do not mean "natural variability of changes", we mean natural variability of changes in future climate, where changes refers to "in future climate". For a better understanding we changed the prepositions in the sentence from "*Both studies highlight the importance of quantifying the natural variability of changes in future fire weather*" to "*Both studies highlight the importance of quantifying the natural variability in changes of future fire weather*".

13. Line 59-63: The claim here is that climate model ensembles using multiple models (but fewer simulations per model compared to SMILE) underrepresent natural variability, whereas SMILE does not. Please clarify how large ensembles using a single model (SMILE) better represent natural variability as compared to large ensembles from different models, and add references that support this claim.

We agree that the term underrepresent is wrong in this context. Multi-Model ensembles mix natural variability with model uncertainty. For this reason, we chose a SMILE framework in our study that allows for a clear isolation of climate change signals from natural variability. We changed the sentences in Lines 59ff.:

Old: "*This limitation, i. e. the under-representation of natural variability in fire danger estimates in regions with currently temperate climate, can be overcome by evaluating climate model simulations derived from a single model initial-condition large ensemble (SMILE). SMILEs represent an ensemble of simulations derived using one single climate model started at different initial conditions. This allows SMILEs to account for the internal variability of the climate system.*"

New: "*This challenge can be addressed by evaluating climate model simulations derived from a single model initial-condition large ensemble (SMILE) which enables a clear isolation of the forced climate change signal from natural variability (Deser et al. 2020). SMILEs represent an ensemble of simulations derived using one single climate model started at different initial conditions. The ensemble spread between the different SMILE members represents the internal variability, from which the forced response of the climate change scenario (i. e. RCP8.5) can be estimated by averaging over the SMILE members (Deser et al. 2020). Therefore, SMILEs are capable to robustly sample extreme events and their probability distribution (Maher et al. 2021).*

14. Line 70: The reference period (1980-2009) is not one of the established reference periods. Please explain the choice of the period. Why not use the almost identical period 1981-2010, which is a widely used reference period?

We chose this time period because the CRCM5-LE model runs until the year 2099. Therefore, the future time period can only be set to the maximum year of 2099, i.e. 2070-2099, which is why we set the present time period to 1980 to 2009. We argue that using $1980 - 2009$ is as good as using $1981 - 2010$ and does not lead to wrong assumptions in our analysis.

15. Line 71: Please change "increases" to "changes", because your analyses were also able to detect if there were any decreases.

You are right, we adjusted "*increases*" to "*changes*".

16. Line 74: Please clarify which TOE you refer to (TOE of what?)

We changed "*the time of emergence (TOE) is reached latest by 2099*" to "*the time of emergence (TOE) of the FWI is reached latest by the year 2099*".

17. Line 82: Please clarify whether your mean two domains in Europe, or two domains of which one is in Europe.

We adjusted from "*over the domains in Europe and Northeast North America*" to "*over the two domains Europe and Northeast North America*".

18. Line 88: clarify "independent" (in which regards?). Fifty members based on the same model are far from independent as such.

Thank you for raising this concern, which is also claimed by RC-3 (general comment). We conclude that our initial data set description is not clearly pointing out the setup of our dataset. Therefore, we revised the first two paragraphs of chapter 2.1:

Old: *To quantify changes and natural variability in fire danger trends for Central Europe, we use the Canadian Regional Climate Model version 5 Large Ensemble (CRCM5-LE) of Leduc et al. (2019). The dataset consists of 50 members at a spatial resolution of 12 km and was generated within the ClimEx project (https://www.climex-project.org/) to assess the hydrological impacts of climate change in Bavaria and Québec. It includes continuous simulations of climate variables from 1950 to 2099 under the RCP8.5 emission scenario over two domains in Europe and Northeast North America (Leduc et al., 2019).*

*The CRCM5-LE is derived from the CanESM2-LE (Fyfe et al., 2017), which was created by applying small random perturbations at two different points in time (i. e. 1850 and 1950) to a 1000-year equilibrium climate simulation under pre-industrial conditions (Leduc et al., 2019). In a first step, small random atmospheric perturbations were added to the equilibrium run to obtain five historical simulation families starting in 1850. In a second step, ten random perturbations were added to each family, resulting in a 50 member ensemble. After a 5-year spin-up phase, the modeled climate of the initialized 50 members can be regarded as independent. This global SMILE was dynamically downscaled using the CRCM5 (Martynov et al., 2013; Šeparovic et al., 2013) to obtain the regional SMILE CRCM5-LE (Leduc et al., 2019). For more details on the ensemble setup, the reader is referred to Leduc et al. (2019) (CRCM5-LE) and Fyfe et al. (2017) (CanESM2-LE)*

New:

*To quantify changes and internal variability in fire danger trends for Central Europe, we use the Canadian Regional Climate Model version 5 Large Ensemble (CRCM5-LE) of Leduc et al. (2019). The CRCM5-LE obtained by nesting the regional climate model CRCM5 (Separoví c et al., 2013; Martynov et al., 2013) into the CanESM2-LE (Fyfe et al. 2017) over two domains (Europe and Northeast America). Thereby, the CanESM2 at an original spatial resolution of 2.88° was dynamically downscaled to 0.11° over these regions. The dynamical downscaling of a regional single-model initial condition large ensemble (SMILE) was carried out within the ClimEx project (https://www.climex-project.org/) to assess the hydrological impacts of climate change in Bavaria and Québec. The dataset includes continuous simulations of climate variables from 1950 to 2099 under the RCP8.5 emission scenario (Leduc et al., 2019).*

*The driving CanESM2-LE (Fyfe et al., 2017) consists of 50 simulations, which were started by adding random perturbations to the initial atmospheric state of January 1st in 1950. These random perturbations were introduced by parameterizing a single aspect of model cloud properties using a different pre-set seed for each of the 50 simulations. This ensures that the climate change realizations are different from each other without changing the model dynamics, physics, or structure (Fyfe et al., 2017). After a 5-year spin-up phase, the modelled climate of the*

*initialized 50 members in the CRCM5-LE can be regarded as independent (Leduc et al., 2019), because the chaotic climate properties cause diverging climate trajectories soley based on the macro- and micro-initialization of the CanESM2 (Wood, 2023). Therefore, the differences among the 50 CRCM5-LE members can be interpreted as natural variability. For more details on the ensemble setup, the reader is referred to Leduc et al. (2019) (CRCM5-LE) and Fyfe et al. (2017) (CanESM2-LE).*

19. Line 91: please clarify "at this time"

We dropped the sentence during editing our manuscript.

20. Line 93: please clarify the link between your study choice and the provided references.

We changed the sentence from "*In this study, we interpret internal variability as natural variability (Böhnisch et al., 2021; Von Trentini et al., 2019; Kay et al., 2015)*" to "*Therefore, the differences among the 50 CRCM5-LE members can be interpreted as natural variability (Böhnisch et al., 2021; Wood, 2023; Mittermeier et al., 2019; Leduc et al., 2019)*".

21. Line 93: Please comment on this assumption (internal variability = natural variability) in the discussion or here. Potential limitations or lack thereof? Comment why this assumption is correct.

We agree that internal variability does not equate to natural variability in any case. We decided to stick to the term internal variability throughout the manuscript, because we used the term natural variability to describe model internal variability. We clarified this section and changed all terms of "natural variability" to "internal variability".

Old: *In this study, we interpret internal variability as natural variability, similar to Böhnisch et al. (2021); Von Trentini et al. (2019) and Kay et al. (2015)"*

New: *Therefore, the differences among the 50 CRCM5-LE members can be interpreted as natural variability (Böhnisch et al., 2021; Wood, 2023; Mittermeier et al., 2019; Leduc et al., 2019), but will be referred to as internal variability throughout this paper (Hawkins and Sutton 2009)*.

22. Line 95: How does smaller (temperature) and equal (precipitation) member spread in your SMILE compared to EURO-CORDEX relate to the previous claim that SMILE overcome the limitation of multi-model ensembles related to under-representation of climate variability. The results of Von Trentini (2019) imply that multi-model ensembles represent a larger variability as compared to CRCM5-LE.

Thank you for this comment. Ensemble spread in a SMILE is originating solely from internal variability, while in multi-model ensembles ensemble spread also includes structural variability (model uncertainty (s. Hawkins and Sutton 2009)). In cases where structural uncertainty is larger than internal variability, a multi-model ensemble such as EURO-CORDEX that also includes structural uncertainty would show larger variability between members than a SMILE that does represent internal variability only. However, the spread of the multi-model ensemble in such a case would come from structural uncertainty rather than from climate variability. If we want to achieve a good representation of climate variability rather than structural uncertainty, SMILEs are needed.

Old: *Their results have shown that the CRCM5-LE shows smaller member spread for temperature and equal member spread for precipitation than EURO-CORDEX (Von Trentini et al., 2019). In cases where structural uncertainty is larger than internal variability, a multi-model ensemble such as EURO-CORDEX that also includes structural uncertainty would show larger variability between members than a SMILE that does represent internal variability only.*

New: *Their results have shown that the CRCM5-LE shows a smaller member spread for temperature and equal member spread for precipitation than EURO-CORDEX (Von Trentini et al., 2019). In cases where model uncertainty is larger than internal variability, a multi-model ensemble such as EURO-CORDEX that also includes structural uncertainty would show larger variability between members than a SMILE that does represent internal variability only.*

23. Line 98: state which observational data were used for the bias correction.

Thank you for pointing out this very important missing information. We added a more specific description of the observation data used for bias correction:

Old: *The CRCM5-LE was bias corrected over the study area for the FWI input variables at a three-hourly resolution using the quantile mapping approach of Mpelasoka and Chiew (2009) (Poschlod et al., 2020).*

New: *The CRCM5-LE was bias corrected using the univariate quantile mapping approach of Mpelasoka and Chiew (2009) (Poschlod et al., 2020) over the study area for the different FWI input variables. […] For the bias correction, the ClimEx project's own meteorological Sub-Daily Climatological REFerence dataset (SDCLIREF) served as an observation reference. It combines hourly and disaggregated daily station data and is described in detail in Brunner et al. (2021). For each quantile bin of each month and sub-daily time step, correction factors were determined by pooling data over all members. The correction factors were applied to each member of the CRCM5-LE separately (Brunner et al., 2021).*

24. Line 100: please rephrase and clarify. Better represented when evaluating against what (wouldn't that be against climate data, which you state is what should be bias-adjusted in the first place)? "climate data" is very general, do you mean data from climate models?

We changed from "*Bias corrected data are commonly used for projections of fire weather indicators like the FWI, because frequencies of FWI extremes are significantly better represented than in non-bias-corrected climate data*" to "*Bias corrected data are commonly used for projections of fire weather indicators like the FWI (e. g. Yang et al., 2015; Cannon, 2018; Kirchmeier-Young et al., 2017; Ruffault et al., 2017; Fargeon et al., 2020) because they have been shown to be more accurate in reflecting fire danger than raw climate data in comparison to observation data (Yang et al., 2015)*".

25. Line 100: Has there been any studies evaluating the data you use against meteorological variables from observations (independent of the bias adjustment) or reanalysis over the region?

Thank you for this valuable remark. Yes, a comparison between observation data and bias corrected and reanalysis data is provided in the supplementary material of Poschlod et al. (2020). Their results show that the biases for temperature are positive and highly variable for the Alps, while they are negative for the other parts of the study area in the summer months June, July and August. For precipitation, the bias correction affects the pre-alpine regions, where the non-bias corrected data show differences > 200 mm from the bias corrected data.

26. Line 103: Please insert the stated rivers in Figure 1 in order to inform a reader, who is not familiar in the region, how the named rivers relate to the study area.

    We updated the figure according to your suggestions by adding the rivers.

27. Line 108: Does 's' refer to 'see'? Please write out. Also, use capital F in figure names.

    We adjusted to capital F in figure names and changed "s." to "see".

28. Line 110: 'water' and 'water availability' are imprecise. Clarify what water you mean and add a supporting reference. If you mean soil moisture or precipitation, these (and thus also the fire regimes) are likely highly heterogeneous within each subregion (in particular in mountainous areas).

    Thank you for pointing out the need for clarification. We changed the sentence to:

    Old: *This subdivision into complex landscapes is adopted from the ClimEx-Project and the study of Willkofer et al. (2020). Since fire is closely related to the availability, or rather the absence of water, we assume that the water availability, climatology and landscape of different river systems is reflected the fire regime of the selected subregions.*

    New: *This subdivision into complex landscapes is adopted from the study of Willkofer et al. (2020) and derived from the Bavarian State Office for the Environment (Landesamt für Umwelt n. d.). Since fire is closely related to the availability, or rather the absence of water (in terms of precipitation deficits), we assume that the water availability, climatology and landscape characteristics of different complex landscapes are reflected in the fire regime of the selected subregions.*

29. Line 109-111: Please clarify and justify your assumption. The subregions are not defined according to the river systems, i.e. river catchments (as seen from Fig. 1). As you state earlier (line 102 and title) and later (line 120-123), hydrology, climatology and landscape are highly variable in the study area, and is likely highly variable in particular in the mountainous areas, within a subregion, with consequences for fire characteristics. What is mean by "an area"?

    We adjusted the unclear terminology regarding river systems (to landscapes) and "an area" (to selected subregions) as described in the previous comment (28.). The subdivision of the study region into four subregions aims to address the trade-off between the number of regions and the amount of inter-subregion variability. While a further subdivision into even more subregions would further increase within subregion homogeneity, it would also make it more difficult to summarize findings. The four regions chosen for the analysis are sufficiently similar in terms of their climate in order to allow for a succinct spatial summary.

30. Line 112-120: Which period and data underlie the numbers presented here? Could a figure (temperature and precipitation spatial patterns) be added (e.g. in appendix) to make the information more intuitive to the reader?

    The numbers presented in the sentence you are referring to are supported by plots of mean temperature and precipitation in Willkofer et al. (2020). As we think that reproducing the content of these figures is of little added value, we instead added a clearer reference to the paper to the text: *Figures illustrating the climatology are provided by Willkofer et al. (2020).*

31. Line 129: As written, 'noon' refers only to wind. Rephrase so it refers to all variables (even 24h precipitation is measured at noon).

    We changed: *(temperature, relative humidity, wind speed at noon and 24-h accumulated precipitation)* to *(temperature, relative humidity, and wind speed - all at noon - and 24-h accumulated precipitation).*

32. Line 132: what is meant by 'bookkeeping'? This concept is linked to financial transactions. Can it be replaced by a more commonly used concept within natural sciences to be more intuitive to the reader?

    We adjusted: *The first three sub-indices represent the fuel moisture codes and can be understood as bookkeeping systems, which increase moisture after rain and reduce moisture for each day of drying.* to *the first three sub-indices represent the fuel moisture codes that contain information about antecedent conditions, e. g. increasing moisture after rain and decreasing moisture for each day of drying.*

33. Figure 2 caption: suggest replacing 'vegetation' with 'organic matter', 'fuel layers' or similar (as used in the main text) for clarity.

    We replaced *"vegetation"* with *"organic matter"*.

34. Line 143: please rephrase or clarify "without memory of past conditions". Because the fuel moisture codes have memory of past conditions, BUI and ISI have too.

    We changed *they are stateless and without memory of past conditions* to *they are stateless and only indirectly linked to past conditions.*

35. Line 156: High-altitude parts of the region will likely have snow in the beginning of the defined fire season. Please state if/how you have accounted for snow in the evaluation here. E.g. see last section in https://www.nwcg.gov/publications/pms437/cffdrs/fire-weather-index-system. If you are neglecting the effect of snow on fire danger, it is worth reflecting on it in the discussion.

    Thank you for your comment. Indeed, we do not take snow cover into account. As you suggest, we added this to the discussion section (s. RC-1 comment 81. and RC-3 comment on Line 359).

36. Line 157: Vitolo et al (2019) does not apply any fire season and should be replaced by reference(s) using or arguing for using April-September.

    We derive the definition of our fire season from Vitolo et al. (2019) who state: "By convention, the dry season in the northern hemisphere is assumed to start on 1st April and ends on 30 September, while in the southern hemisphere it starts on 1st October and ends on 31st March." Therefore, we believe that the reference is appropriate to justify the choice of the fire season definition.

    Old: *The generated dataset is later cropped to the fire season (April 1st to September 30th) of the northern hemisphere (Vitolo et al., 2019).*

    New: *The generated dataset is later cropped to the dry season (April 1st to September 30th) of the northern hemisphere, which is used as the fire season in our study as suggested by Vitolo et al. (2019).*

37. Line 158: "annually calculated FWI values" is unclear (may refer to annual values, which I assume is not the case). Suggest to delete "of annually calculated FWI values" for clarity.

We implemented your suggestion and deleted "of annually calculated FWI values". Further, we changed "*We calculated daily FWIs for each year (January to December) and climate model ensemble member between 1980 and 2099 using the CFFDRS R package (Wang et al., 2017)*" to "*We calculated the FWI on a daily basis for each full year (January to December) between 1980 and 2099 and for each climate model ensemble member using the CFFDRS R package.*" *for clarification.*

38. Figure 3: Please add a legend (ref. NHESS figure composition: https://www.natural-hazards-and-earth-system-sciences.net/submission.html#manuscriptcomposition).

We added a legend to Figure 3.

[Figure]

*Median FWI for the CRCM5-LE mean (thick blue line) and standard deviation (light blue shading) in comparison to the reference dataset of Vitolo et al. (2020) marked pink (X for values, lines for deviation from the CRCM5-LE mean). Top and bottom blue lines mark the 25th and 75th percentile of the CRCM5-LE.*

39. Line 170: As you state yourself, the results of the evaluation you have performed does not affect the climate change impact assessment of your study. Why do you not evaluate your data using measures that can actually reflect your data's ability to assess climate change impacts? For example its ability to represent historical changes.

Thank you for this remark. We agree that this sentence is misleading in its message. On a temporal scale (see Figure 3), we show that the CRCM5-LE captures observation values within a tolerated range between the 25th and 75th percentile well.

We clarified this sentence also in correspondence with RC-3 (comment on Line 170) and added a remark to this paragraph, that the ensemble overestimates interannual variability:

New: *The CRCM5-LE overestimates the internal variability in comparison to the reference dataset (s. Figure 3 following the framework of Suarez-Guiterrez et al. (2021) to evaluate internal variability in SMILEs.*

40. Figure 4: The applied FWI colour scale is almost identical to the FWI colour scale provided in Table 1, but they reflect different FWI intervals. Please change for clarity.

We adjusted the color scale to the colors in Table 1 and updated the figure.

[Figure]

*Median FWI of (a) the CRCM5-LE, (b) reference dataset of Vitolo et al. (2020) and (c) difference (CRCM5-LE − reference dataset) for the present time period (1980−2009). The dataset difference is calculated from resampling (a) to the spatial resolution of (b) using a nearest neighbour approach.*

41. Section 2.5.1: The description is unclear in terms of when the different aggregations were applied (both in space and time), and when the continuous analysis vs the data split are applied. Consider reorganise the section to better fit each part of the analysis. Please also clarify how the 'extreme condition' (90[th] percentile) is computed (is it over the analysed time period, region or models, and in which order is it calculated). Clarifications in this section are necessary for reproducibility.

Thank you for pointing out the need for clarification. We did not perform a trend analysis but compared two time periods and therefore changed the subsection title from "*Trends*" to "*Changes in Fire Danger*". Further, we clarified the description of the pooling procedure, e. g. how we derived the median and extreme percentiles:

Old: *We evaluate the fire danger trends derived from the CRCM5-LE over the time period 1980 to 2099 in the study area with statistical metrics: Median conditions are examined using the 50th percentile (median) of the FWI. Extreme conditions are evaluated via the 90th percentile*

*(extreme). The percentiles are calculated for different aggregation levels, either temporally, summarizing FWI values of a fire season on daily, monthly or annual basis, or spatially for the previously defined subregions. Increasing fire danger is either analyzed continuously from 1980 to 2099 or compared between two climate periods. For the climate period comparison, the dataset is split into two 30-year periods: 1980–2009 as present and 2070–2099 as future.*

New: *We evaluate changes of fire danger derived from the CRCM5-LE over the time period 1980 to 2099 in the study area with statistical metrics: Median conditions are examined using the 50th percentile (median) of the FWI. Extreme conditions are evaluated via the 90$^{th}$ percentile (extreme). The percentiles are calculated for different aggregation levels, either temporally on a monthly scale or spatially for the previously defined subregions. We derive the median (50$^{th}$ percentile) and the extreme (90$^{th}$ percentile) for each ensemble member separately. Changes of fire danger are either compared between two climate periods or analyzed continuously from 1980 to 2099. For the climate period comparison, the dataset is split into two 30-year periods: 1980–2009 as present and 2070–2099 as future. For these periods we derive the median and extreme percentiles for each fire season month for each of the 50 members of the CRCM5-LE.*

42. Line 174: Please provide which trend method was applied.

    We did not perform a trend analysis, instead, we compared FWIs across two time periods (1980-2009 and 2070 - 2099). Therefore, we rephrased "trend" to "changes".

43. Line 177: You state that you are "summarizing FWI values over a fire season on daily, monthly or annual basis". Summarizing would provide very different ranges of FWI on the different temporal scales, and it does not look like they are summarized in e. g. Fig. 5 (looks like average or median over each month). Please correct or clarify.

    Thank you for indicating this misleading description. We clarified the sentence by changing from: "*The percentiles are calculated for different aggregation levels, either temporally, summarizing FWI values of a fire season on daily, monthly or annual basis, or spatially for the previously defined subregions.*" To: "*The percentiles are calculated for different aggregation levels, either temporally on a monthly scale or spatially for the previously defined subregions*".

44. Line 178 and line 182: Suggest replacing 'increasing' with 'changes in' for clarity. The analyses allow for changes in both directions.

    We changed "*increasing"* to "*changing"* in the suggested lines.

45. Line 216: Your results are scenario specific. Please specify the scenario, e. g. "according to RCP 8.5".

    The sentence starts now with "Based on the RCP8.5 emission scenario…".

46. Line 219-222: Please rephrase to clarify your reasoning.

    We agree that the original distinction between weak (one-level) and strong (two-level) fire danger level increases is imprecise and subjective. We rephrased "*weak*" to "*one level*" and "*strong*" to "*two level*". Further, we added a sentence, which explains why we also look at two-level increases. This is mainly because one level increases are found throughout the entire study

area and we want to provide additional information by distinguishing between one level and two level rises in fire danger.

Old: *We distinguish between weaker (one 220 level, thin dots) and stronger (two levels, thick dots) fire danger level rises, because in July and August almost the entire study area shows a robust fire danger level rise of one level for both median and extreme conditions. This helps us to identify regional hotspots*.

New: *We distinguish between one-level (thin dots) and two-level (thick dots) fire danger level rises because the entire study area shows fire danger level rises of one level for both median and extreme conditions in July and August. Highlighting grid cells, which experience a rise of at least two levels, helps us to identify regional hotspots of future increases in fire danger*.

47. Line 223: Why is not the Southgerman Escarpment mentioned here (regions of strongest rises in extreme FWI in July to Sep)?

    We clarified this sentence, which refers to changes of at least two fire danger levels.

    Old: *The Southgerman Escarpment in the northwest of our study region is most affected by changes in the median FWI, while the Alps and the Eastern Mountain Ranges experience the strongest fire danger level rises in the extreme FWI in the months July to September (s. figure 5)*

    New: *We find increases in fire danger of at least two levels for the Southgerman Escarpment in July and August for the median FWI. The other subregions (Alps, Alpine Foreland, Eastern Mountain ranges) are affected by a two level rise only in July, whereas the western parts of the Alpine Foreland and parts of the Eastern Mountain ranges are affected by a two-level rise in September (see Figure 5).*

48. Line 226, 228 and 229 and potentially other places: 'median case' and 'extreme case' are unclear concepts. Do you refer to 'median FWI' and 'extreme FWI'? Please be consistent with concepts or clarify newly introduced ones.

    Thank you for highlighting this. We refer to the median FWI (50th percentile) and extreme FWI (90th percentile) of our results. We adjusted the sections accordingly.

    Old: *The median case points out that high fire danger becomes the average condition in the Alpine Foreland by 2080, in the Southgerman Escarpment by 2060 and in the Eastern Mountain Ranges by 2070 (s. figure 6 [1]). The Alps are exposed to high fire danger only in the extreme case (s. figure 6 [2]) from 2070 onwards. The other subregions are much more strongly affected in the extreme case:*

    New: *The median FWI points out that high fire danger becomes the average condition in the Alpine Foreland by 2080, in the Southgerman Escarpment by 2060 and in the Eastern Mountain Ranges by 2070 (see Figure 6 [1]). The Alps are exposed to high fire danger only in the extreme FWI (see Figure 6 [2]) from 2070 onwards. The other subregions are much more strongly affected in the extreme FWI:*

49. Line 233: Why do you state 'mean conditions (median)' and not 'median conditions' (what is the difference)? Please clarify in the text. Similarly, clarify similar statement in line 349: 'On average (median)'

Thank you for pointing this out. We are looking at median conditions throughout the study and clarified this by replacing "mean" or "average" conditions by "median" conditions. Further, we edited Section 2.5.1. to clarify that we derive the median and extreme percentile on a member basis and use the average / mean to derive the ensemble mean for the median and extreme percentiles, when we speak about ensemble mean.

Old: *The ensemble mean shows hardly any fire danger changes over the 21st century in the median and extreme case for April (s. figure 6). High fire danger becomes the mean condition (median) in the summer months for large parts of the study region (figures 5 and 6).*

New: *The ensemble mean shows hardly any fire danger changes over the 21$^{st}$ century in the median and extreme FWI for April (see Figure 6). High fire danger becomes the median condition in the summer months for large parts of the study region (Figures 5 and 6).*

50. Line 236: Please consider using the phrasing 'mid 21$^{st}$ century' instead of 'middle of the 21$^{st}$ century'.

    Thank you for pointing this out. We replaced "*middle of the 21st century*" by "*mid 21st century*".

51. Line 240-241: Please clarify how your findings indicate that the distribution of the FWI extremes resembles the distribution of the FWI median? Figure 7 clearly shows that the distributions are different both in terms of mean and standard deviation.

    We removed this statement from the text.

52. Line 242: consider replacing 'changes' to 'increases' for clarification.

    We clarified the sentence by following your suggestion.

    Old: *FWI changes in the Alps are weaker than in the other subregions.*

    New: *FWI increases in the Alps are weaker than in the other subregions.*

53. Line 244: please specify what you mean by 'strongly'.

    We exchanged *"strongly"* with *"continuously"*.

54. Line 244-245: Please clarify what you mean and which parts of the results you refer to. Your statement here seem opposite compare to the preceding sentence (median FWI increase strongly vs median FWI shows hardly any changes).

    We clarified that this section soley refers to the results for the Alps.

    Old: *Throughout the 21$^{st}$ century, the median and extreme FWI increase strongly. While the extreme FWI is projected to shift from low to moderate fire danger, the median FWI shows hardly any changes and remains in the no danger level (below five) according to EFFIS (2021).*

    New: *Throughout the 21$^{st}$ century, the median and extreme FWI increase continuously in the Alps. While the extreme FWI is projected to shift from low to moderate fire danger in this subregion, the median FWI shows hardly any changes and remains in the no danger level (below five) (see Table 1).*

55. Line 245: Why state the EFFIS reference here, when every classification of fire danger level in the (also previously mentioned) results is based on it?

    You are correct, this is duplicated information. We changed the reference to Table 1 (see applied changes in comment 54.)

56. Line 247: clarify which average you are referring to.

    We clarified the sentence from "*In the extreme case, the average fire danger is moderate*" to "*In the extreme case, the ensemble mean fire danger is moderate*".

57. Line 247-248: the interval signs in parenthesis are wrong.

    Thank you for finding this tiny but very relevant mistake – we corrected it.

    Old: *(11.2 < moderate > 21.3) & (21.3 < high > 38)*

    New*: (11.2 > moderate < 21.3) & (21.3 > high < 38)*

58. Figure 5: The levels referred to in the result section would be more easily recognisable if a colour scale using discrete colours was used. Discrete colours would provide a more clear message to the reader, in particular when these levels are the main message of these results, and not the small varieties in between. Please consider changing to discrete colours.

    We agree that a discrete color scale is more appropriate here than a continuous color scale. Therefore, we, adjusted the plot accordingly.

[Figure]

*Fire rings show the ensemble mean of the monthly median ([1], 50th percentile) and extreme ([2], 90th percentile) FWI of each subregion (Alps, Alpine Foreland, Southgerman Escarpment and Eastern Mountain Ranges (a-d)) during the fire season (April - September) between 1980 and 2099.*

59. Figure 5 caption: do you mean "at least" two levels (indicated by thick black dot), or are there never more than two levels?

We clarified this to at "at least one/two levels".

Old: *Ensemble mean of the median ([1], 50th percentile) and extreme FWI ([2], 90th percentile) by fire season month (April (a) - September (f)) for the future time period 2070–2099. Dots indicate that 90% of the CRCM5-LE members agree on a fire danger level increase of one (thin black dots) or two (thick black dots) levels compared to the present period (1980–2009).*

New: *Ensemble mean of the median ([1], 50th percentile) and extreme FWI ([2], 90th percentile) by fire season month (April (a) - September (f)) for the future time period 2070–2099. Dots indicate that 90% of the CRCM5-LE members agree on a fire danger level increase of at least one (thin black dots) or at least two (thick black dots) levels compared to the present period (1980–2009).*

60. Line 259-260: please provide numbers or proportions in parenthesis.

We provided the percentage of days per fire season in parenthesis.

Old *While the higher danger levels already occur in the present, very high danger levels additionally occur in the future.*

New: *High fire danger already occurs in the present (1 % in the Alpine Foreland, 3 % in the Eastern Mountain Ranges) and shifts towards fractions of 10% in the Alpine Foreland and almost 20% in the Eastern Mountain Ranges in the future, where very high danger levels additionally occur (1 % in the Alpine Foreland, 3 % in the Eastern Mountain Ranges)*

61. Line 262: Please provide in what ways they are similar. EMR is not described in other terms than relative to Alpine Foreland.

We clarified the section.

Old: *In the Eastern Mountain Ranges, similar results are observed:*

New: *The Eastern Mountain Ranges show similar results as the Alpine Foreland in terms of the number of days with a certain fire danger level. However, the Eastern Mountain Ranges differ slightly from the Alpine Foreland: Higher fire danger levels already occur in the present, very high danger levels additionally occur in the future. In comparison to the Alpine Foreland, moderate fire danger days are less frequent and high fire danger days are more frequent in the Eastern Mountain Ranges.*

62. Figure 8: Please consider adding proportions on the right y-axis, as proportions are used in the text.

We added proportions to the figure.

[Figure]

*Number of days of fire danger levels in the fire season (April – September, 183 days) for the present (1; 1980–2009) and future (2; 2070–2099) climate period. FWI danger classes are derived for the subregions (a) Alps, (b) Alpine Foreland, (c) Southgerman Escarpment and (d) Eastern Mountain Ranges*

63. Figure 8 caption: Please clarify by specifying what is meant by frequency (e.g. "number of days within a fire season")

We exchanged *"frequency"* by the *"number of days"*. For the implementation see the Figure caption in the previous comment (RC-1 comment 62)

64. Line 278: As in line 215, clarify the scenario dependence of your results (in line with your statement in line 306-307). The way it is phrased now imply more certainty about the future than we can state.

We added "*when a RCP8.5 scenario is assumed*" to the sentence to emphasize that our results soley refer to the RCP8.5 scenario.

Old: *Our results demonstrate that fire danger increases dramatically over the next few decades in Central Europe.*

New: *Our results demonstrate that fire danger increases dramatically until the end of the 21st century in central Europe, when a RCP8.5 scenario is assumed.*

65. Line 279: Why move away from the defined classes? How is hazardous defined?

Thank you for recognizing this stylistic inconsistency. We rephrased the sentence:

Old: "*The trend towards hazardous fire danger conditions in the future emerges for all presented metrics in this study, i.e. different temporal, spatial and ensemble aggregation levels.*"

New: "*The increase of days with conditions favoring fire danger levels of high and higher in the future emerges for all presented metrics in this study, i. e. different temporal, spatial and ensemble aggregation levels*".

66. Line 285-286: please state the variable (FWI) that is compared.

We added FWI to the sentence.

Old: *Before starting our analysis, we compared the results from the CRCM5-LE to the dataset of Vitolo et al. (2020) for the present climate period (1980–2009).*

New: *Before starting our analysis, we compared the FWI results from the CRCM5-LE to the FWI dataset of Vitolo et al. (2020) for the present climate period (1980–2009).*

67. Line 289-291: Please state in what relevant ways the formulas have been adjusted (i.e. relevant implications). Is this a more likely reason for the differences as compared to the fundamental differences in how the underlying meteorological data are produced?

Thank you for this valuable remark. This statement originates from Vitolo et al (2019). Vitolo et al. (2019) state in their algorithm validation section "Although the outputs are rather close, they do not match exactly. The reason is that the ECMWF model follows the formulation defined in the reference FWI implementation outlined in Van Wagner (1987) without modifications. Wang et al. (2017) instead, have modified some of the original equations (i.e. EQs 12 and 15) leading to the calculation of DMC and DC. As a consequence, FWI and DSR also slightly differ."

However, Wang et al. (2017) do not explain where and how they adjusted the original formulas. We therefore keep this sentence as it is.

68. Line 293: please rephrase sentence to be more to the point. It is unclear how the tiling patterns referred to in the text 'has to be discussed' and not the ones seen e. g. in September (Fig. 5 [2]f) or at smaller scale in the Alps in July-Sep (Fig 5 [1]def and [2]def).

Thank you for your comment. We agree that this sentence is not clearly brought to the point. We rephrased the whole section to clarify our intentions with mentioning the tiling pattern, though it is not as strongly visible in Figure 5 after adjusting to a discrete color scale.

Old: *Another aspect, which has to be discussed, is the strong tiling pattern visible in figure 5 [2] in the months June and August. This tiling pattern is already visible in the extreme values of the input variables. We provide a sensitivity analysis of the FWI in the Appendix (s. figure C1), where the tiling occurs for temperature and relative humidity in the 95th 295 percentiles as well. The pattern correlates with invariate fields from the geophysical baseline parameterization of the CanESM2, e.g. bedrock depth. Over the areas where the strong tiling occurs, bedrock depth is about 5m. The water storage potential of the ground is especially high in this area compared to its surrounding areas with an average bedrock depth between 1 or 2 meters. Such high storage potential can affect evaporation and leads to a higher cooling in areas with high bedrock depths which results in lower 300 temperatures and higher relative humidity.*

New: *Though the CRCM5 reproduces the response structures much finer than CanESM2 and adds robust high-resolution features (Böhnisch et al., 2020), we find in the northern parts of the study area tiling patterns corresponding to the geophysical baseline parameterization of the CanESM2 (see Figure A3). The tiling occurs in the sensitivity analysis provided in Figure A.3 for temperature*

*and relative humidity in the 95ᵗʰ percentile, when the FWI is calculated with a factor of two for temperature and relative humidity. The pattern correlates with the bedrock depth of the CanESM2, which might affect the water storage potential of the ground. Over the areas where the tiling occurs, bedrock depth is about 5m, which is relatively high in comparison to the surrounding areas with an average bedrock depth between one or two meters. Such high storage potential can affect evaporation and leads to a higher cooling in areas with high bedrock depths which results in lower temperatures and higher relative humidity. The tiling occurs only under very extreme FWI conditions (95th percentile) and might lead to an overestimation of our results in the extreme FWI (90th percentile) for the Southgerman Escarpment.*

69. Line 296: please change 'correlates' with a more appropriate word or provide correlation results.

    We exchanged *'correlate'* with *'correspond'*.

70. Line 302: you mention the uncertainty related to the chosen climate model. Please elaborate on this point in relation to the specific model you applied.

    Thank you for highlighting the missing discussion of the performance of the CRCM5-LE in comparison with other climate models. We edited this section carefully by adding a sentence that explains the difference between the CRCM5-LE and other CORDEX models in terms of precipitation and temperature.

    New: "*In comparison to the CORDEX ensemble, the CRCM5-LE shows drier and warmer climate change signals for temperature and precipitation (Von Trentini et al. 2019). These characteristics of the CRCM5-LE are in line with the results from the validation (see Figure 3) and indicate an overestimation of our results.*"

71. Line 316: please remove "potential of". FWI describes the fire weather, not the potential of fire weather.

    We removed "*potential of*".

72. Line 318-327: Please consider deleting this paragraph, and alternatively reduce the main message to a single sentence in the methods chapter arguing for your use of danger levels.

    We deleted the paragraph and explained the reason for using fire danger levels in the methods section.

    New sentence in chapter 2.4 (Methods): *To facilitate the interpretation of the FWI, we use the seven fire danger classes proposed by the European Forest Fire Information System (EFFIS) (EFFIS, 2021) and assign the FWI to particular fire danger levels. These FWI danger levels and their corresponding color scheme are shown in Table 1.*

73. Line 328-331: Please elaborate briefly on the flammability of the surface in your study region.

    We added a land use map to the appendix and described the flammability in the study region and subregions briefly in the second paragraph of chapter 4.2. We also discussed your previous comment on snow cover here.

    New: *While the FWI addresses fire danger in a meteorological context, it does not account for the flammability of the surface. Land-Use in our study area is complex, but contiguous forests are*

*present in all four subregions, especially the Eastern Mountain Ranges and the Alps. Persistent snow cover from snowfall in the winter season prevents fire occurrences in spring in the Alps (Conedera et al. 2018) and other regions of high elevation, though fire weather conditions might be met. Large parts of the South German Escarpment and Alpine Foreland are used for agricultural purposes, where fires can spread fast under dry conditions (see Figure A1). However, these regions are more densely populated than the other two regions (Eastern Mountain Ranges and the Alps), which allows a faster mitigation of fire incidents. For large-scale FWI analyses, non-burnable areas such as deserts and bare soil are masked out (Vitolo et al. 2020, Touma et al. 2021). In the context of the study area HydBav and the 11-km resolution of the CRCM5-LE, land use is highly variable on a sub-pixel scale and non-burnable areas (e. g. lakes, snow- and ice-covered areas and urban areas) are therefore not masked out (see Figure A1).*

74. Line 333-339: Please reflect/explain results rather than summarise them.

We shortened this section and emphasized the differences between mountainous and non-mountainous terrain.

Old: *We find that the region affected most strongly by FWI increases is the northwest, i. e. the Southgerman Escarpment (s. figures 5, 6 and 8). Noteworthy is, that average changes (median) are smaller in the Alps but increases in the extreme FWI are strongest in the Alps. The trends of the median are similar for the Alpine Foreland and the Eastern Mountain Ranges, but FWI extremes in the Eastern Mountain Ranges increase more strongly than in the Alpine Foreland. We summarize that increases in fire danger extremes are more pronounced than increases in median conditions and therefore variability increases in regions with heterogeneous terrain (Alps and Eastern Mountain Ranges). For less complex terrain (Alpine Foreland and Southgerman Escarpment), the increases in fire danger extremes are less variable. These findings corroborate findings by Wastl et al. (2012), who explained the higher fire danger variability in mountain regions by the higher terrain variability, i.e. rain-shadow effects and katabatic dry winds (foehn).*

New: *Our results (see Figures 5, 6 and 8) show that increases in fire danger extremes are more pronounced than increases in median conditions. In the Alps, this is demonstrated by smaller changes in the median FWI than in the extreme FWI. For less complex terrain (Alpine Foreland and Southgerman Escarpment), the increases in fire danger extremes are less variable. For example, the increases of the median FWI are similar for the Alpine Foreland and the Eastern Mountain Ranges, but extreme FWI (90th percentile) in the Eastern Mountain Ranges increase more strongly than in the Alpine Foreland. This finding indicates that variability of the FWI increases more strongly in mountain regions than in non-mountain regions and corroborates findings by Wastl et al. (2012), who explained the higher fire danger variability in mountain regions by the higher terrain variability, i.e., rain-shadow effects and katabatic dry winds (foehn), by evaluating weather station data.*

75. Line 340: increases in variability (line 337) is not the same as high variability in general (line 340). Please elaborate what you mean by your findings (increasing variability over time in mountainous regions) corroborate the findings by Wastl et al (2012; higher variability in mountainous regions than other regions).

We clarified this sentence in line with specific comment 74.

New: *For example, the increases of the median FWI are similar for the Alpine Foreland and the Eastern Mountain Ranges, but extreme FWI (90th percentile) in the Eastern Mountain Ranges increase more strongly than in the Alpine Foreland. This finding indicates that the variability of*

*the FWI increases more strongly in mountain regions than in non-mountain regions and corroborates findings by Wastl et al. (2012), who explained the higher fire danger variability on in mountain regions by the higher terrain variability, i.e. rain-shadow effects and katabatic dry winds (foehn), by evaluating weather station data.*

76. Line 345: Unclear whether 'extreme FWI conditions' represent the 90[th] percentile or the classes (FWI>50). In case of the former, do you mean elevated conditions compared to former months or compared future to present. In case of the latter, is that not seen directly from the figure and not 'implied' from your findings? Please clarify the meaning of this sentence.

    We changed this sentence from: *This finding indicates that the seasonal variability is higher for the last three months of the fire season and implies that the probability for extreme FWI conditions is elevated during these late summer months.*

    To: *This finding indicates that the variability of the FWI is higher in the last three months (July, August, September) in comparison to the first three months of the fire season (April, May, June). This implies that extreme FWI events are more likely to occur in the second half of the fire season (July, August, and September) than in the first half of the fire season (July, August, and September) than in the first half of the fire season (April, May and June).*

77. Line 348: 'tremendous' is subjective, please clarify. See also 'dramatic' in line 362 and 'strikingly' in line 364-365.

    We changed "*tremendous*" to "*substantial*", "*dramatic*" to "*as high fire danger levels as*" and "*strikingly*" to "*remarkable*".

78. Line 349: by 'seasonal', do you mean 'monthly'? In which ways are they hotspots, in terms of general conditions/increases/other?

    We rephrased this sentence:

    Old: *Especially the months August and July can be identified as seasonal hotspots throughout the study area. On average (median), the fire danger will be high in the Alpine 350 Foreland, Southgerman Escarpment and Eastern Mountain Ranges and moderate in the Alps by the end of the century.*

    New: *Especially the months August and July can be identified as months with the highest fire danger of the season throughout the study area.*

79. Line 358: The use of vegetation in Figure 2 caption implies also litter and organic matter on the ground. In this context, vegetation is necessary for fire development because it comprise the fuel. Is it the same use of vegetation here? I assume vegetation is highly present during winter also, although parts are covered by snow, and deciduous trees lack their green leaves. Please clarify the text.

    We clarified this sentence. We do not refer to vegetation itself, but to the vegetation period. In the winter season, the vegetation is not actively growing, which leads to decreased fuel moisture. We rephrased this section (see Wastl et al. 2012 and Conedera et al. 2018).

    Old: *For the Southern Alps, Wastl et al. (2012) identified the main fire season between December and April because of low precipitation and missing vegetation cover in the winter half year. Therefore, future studies assessing changes in fire danger in the Alps should focus on the whole year instead of the summer season only.*

*New: For the Southern Alps, Wastl et al. (2012) identified the main fire season between December and April because of low precipitation and decreased fuel moisture outside the vegetation period in the months December to April (Conedera et al., 2018). With respect to the increasing altitude of vegetation, increasing length of the vegetation period and decreasing snow cover (Rumpf et al., 2022), future studies assessing changes in fire danger and fire events in the Alps and other temperate climate regions should consider analyzing the whole year instead of the summer months only.*

80. Line 359: 'half year' typically refers to six months. Consider changing to 'period' or similar, as you refer to December-April.

   We exchanged *"winter half year"* to *"months from December to April"*.

81. Line 358-360: would FWI be suitable for the winter season? The reasoning provided here include lack of vegetation, whereas this is not accounted for in FWI. And what about snowfall and snow cover? Further, would you assume the temperature thresholds included in FWI calculation be exceeded in the Alps in winter? Please reflect on the considerations needed for such assessments.

   We added a sentence explaining that the FWI is not suitable for the winter season and suggest to consider using other approaches in cases where the winter season is explicitly considered, e.g., the one proposed by Pezzatti et al. (2020). However, our study only focuses on the months April to September, when snow cover in a 11 km grid scale plays a minor role for forest fire danger, because it occurs only in unvegetated high alpine terrain, which is sampled only by a small fraction of the 11 km grid.

   Old: *For the Southern Alps, Wastl et al. (2012) identified the main fire season between December and April because of low precipitation and missing vegetation cover in the winter half year. Therefore, future studies assessing changes in fire danger in the Alps should focus on the whole year instead of the summer season only.*

   New: *For the Southern Alps, Wastl et al. (2012) […]. However, the FWI can not capture these land cover and vegetation specific changes and therefore other methods should be considered to quantify fire danger outside of the summer period, i. e. Pezzatti et al. (2020).*

82. Line 366: states 'exists currently no fire danger', however you have fire danger everywhere (as fire danger is defined as the estimates from the index, regardless of values). Please clarify.

   We rephrased to *"the fire danger (FWI) is almost zero"*.

83. Line 372-373: you mention overestimation of natural variability. How does this relate to line 59? What about potential underestimation when using SMILE? If a model has a limitation (e.g. in representing natural variability), all realisations from that model suffer from the same limitation. If you or other have validated the ability of SMILE to represent natural variability, please state this in the text and refer to relevant evidence. Applies also for line 375.

   Line 59 refers to multi-model ensembles, which do not overestimate natural variability but do not allow to distinguish between natural variability and model variability. We revised this section also in accordance with specific comment 12:

Old: *Reasons for this delay could be due to the later and shorter reference period (1995–2015), the overestimation of natural variability in the multi-model ensemble (Fargeon et al., 2020) or the slight overestimation of the CRCM5-LE (s. chapter 2.4).*

New: *Reasons for this delay could be the later and shorter reference period (1995–2015), the larger uncertainty range originating from natural variability, model uncertainty in the multi-model ensemble (Fargeon et al., 2020), or the warmer and drier climate change signal of the CRCM5-LE (Von Trentini et al., 2019).*

Further, we edited line 59 in RC-1 comment 13 and elaborated on this in the new discussion chapter uncertainties as suggested by RC3 comment 1.

84. Line 373: 'slight overestimation of the CRCM5-LE'. Clarify, what does it overestimate?

Revised in specific comment 83.

85. Section 4.4: the title and content of the section does not match (impacts [title] vs conditions influencing flammability, emergency in other regions. Further, the content is not coherent. Please revise and clarify the message.

Thank you for your comment. We agree that this section is not consistent in terms of its message. We revised this section in the following way:

1. Adjusted the Title to "Regional Shifts and Implications"
2. Focused on spatial differences between the fire regime in the Mediterranean and Central Europe in the first paragraph:

   Old: *However, for wetter, more productive regions and seasons, i.e. our study area in Central Europe, aridity does not limit fuel availability, which implies higher sensitivity to flammable conditions (e.g., after hot and dry seasons) and points out the importance of considering vegetation and fuel structure changes in further studies (Pausas and Paula, 2012; Turco et al., 2018). Further, Bowman et al. (2020) suggest that declining snow cover in spring and drier fuels in summer will increase burned area in mountain forests, as present in the Alps and Eastern Mountain Ranges in our study area.*

   New*: For wetter, more productive regions, i.e. our study area, aridity does not limit fuel availability. Bowman et al. (2020) suggest that declining snow cover in spring and drier fuels in summer will increase burned area in mountain forests, as present in the Alps and Eastern Mountain Ranges in our study area. This implies higher sensitivity to flammable conditions (e.g., after hot and dry seasons) and an extension of fire events to more northern latitudes and higher elevations.*

3. We generalized the Bavarian specific section to a broader call for mitigation measures in Central Europe

   Old: *For the Mediterranean, Turco et al. (2018) expect changes in meteorological fire weather of such a magnitude, that current fire suppression measures are not sufficient anymore. The guidelines for forest fire defence in the federal state of Bavaria currently only ask the public for cautious behaviour when fire danger is elevated. In case of high or very high fire danger, 390 surveillance flights are carried out in the respective areas*

*(StMLF, 2013). Studies in other regions, i.e. the UK (Arnell et al., 2021) and France (Fargeon et al., 2020), suggest that increases in fire danger should be considered in emergency, land use and management planning to mitigate future fire risk. Taking the results of our study into account, these suggestions apply for Hydrological Bavaria as well.*

New: *Expected changes in fire weather in the Mediterranean are of such a magnitude, that current fire suppression measures are not sufficient anymore (Turco et al., 2018). Studies in other regions, i.e. the UK (Arnell et al., 2021) and France (Fargeon et al., 2020), suggest that increases in fire danger should be considered in emergency, land use and management planning to mitigate future fire danger. Despite the differing climatic conditions and land cover in comparison to France and England, our research findings indicate that forest fire mitigation measures must be proposed for central Europe as well.*

86. France and UK: Several places in the manuscript, results of France and UK is used for guiding and comparing the results of the present study, and to make final recommendations for fire emergency. However, you do not reflect on potential relevant differences between the regions (e.g. hydroclimotology and vegetation). Please consider commenting on such aspects.

   We agree that our manuscript was not taking differences between the study areas of Fargeon et al. (2020) (France) and Arnell et al. (2021) (UK) into account sufficiently. We stressed the regional differences between France, UK and Germany in the last sentence of our Discussion chapter.

   Old: *Taking the results of our study into account, these suggestions apply for Hydrological Bavaria as well.*

   New: *Despite the differing climatic conditions and land cover in comparison to France and England, our research findings indicate that forest fire mitigation measures must be proposed for Central Europe as well.*

   For a broader context of this paragraph the reader is referred to the response of the previous comment (RC-1, comment 85, 3[rd] answer)

87. Line 397 (and line 406): You state that the study area is not affected by high fire danger to date, but high fire danger is present in relatively large areas in current climate (Fig. 5[2]def, where the dots indicate a change from a currently high level).

   We agree with your comment and rephrase the sentence:

   Old: *The study area is not yet affected by wildfires and high fire danger to date, but will be affected in the future when assuming an RCP8.5 emission scenario and accounting for natural variability.*

   New: *To date, the study area is irregularly affected by wildfires and high fire danger occurs only under extreme conditions (90[th] FWI percentile). However, high fire danger will become more frequent in the future when assuming an RCP8.5 emission scenario.*

88. Line 397-398: Please clarify 'by accounting for natural variability'.

   We dropped that phrase (see specific comment 87).

89. Line 398: Please clarify the difference between "strongest increase" and "most hazardous developments"

We rephrased most *"strongest increases"* to *"strongest changes"* and *"hazardous developments"* to *"highest fire danger levels within the study area"* for clarification.

90. Line 400: please clarify in what terms, and in which results the statement "less strongly affected" applies. For example, in fig 5[2], Alps is the only region with dots in April and May, and the two regions you mention increase multiple fir danger levels as seen e.g. in Fig. 6[2] august. As mentioned earlier in the manuscript (line 324-326), increases in classes may provide a better approach to assess increases due to non-linearity, and thus and linear comparison (e.g. Fig 7) may not the best way to conclude the strongest trends.

We revised this section and dropped the imprecise statement about regions which are "less strongly affected".

Old: "*The strongest increases and most hazardous developments are observed North of the river Danube in the summer months July and August for the subregions South German Escarpment and Eastern Mountain Ranges. Regions south of the Danube (Alps and Alpine Foreland), are less strongly affected by changes in the FWI but increases are still significant."*

New: *We find the strongest changes and highest fire danger levels north of the river Danube in the summer months July and August for the subregions South German Escarpment and Eastern Mountain Ranges.*

91. Line 401: the statement that FWI has a stronger variability for Alps and Eastern Mountain Ranges contradicts the findings in Fig. 7, where the standard deviation is smaller for these regions compared to the other subregions. Please clarify.

We appreciate your comment to set this in context with the findings of Figure 7. Figure 7 is derived from highly aggregated data (30-year daily fire season running means) and therefore has a different aggregation level than Figures 5 and 6. However, this does not clarify the findings and we decided to drop this section.

92. Line 404: please consider repeating the hypothesis, and structure the conclusions by these.

We restructured the conclusions to follow the different research questions/hypotheses.

New Paragraph: *Our results provide clear answers to our initially proposed research questions. They demonstrate that fire danger increases significantly throughout the study area. We find the strongest changes and highest fire danger levels north of the river Danube in the summer months July and August for the subregions South German Escarpment and Eastern Mountain Ranges.. Our results also show that the time of emergence(TOE) is reached in all subregions before 2050. Further, we showed that not only the mean but also the lower boundary of the running mean, represented by the CRCM5-LEs standard deviation, exceeds the upper boundaries of the present climate (1980 - 2009) standard deviation before 2099 in all subregions for the 90th FWI percentile. Last, our findings highlight that the return period of present 100-year events shifts towards 10-year events by 2090 and the return periods for 100-, 50- and 20-year events shift to 50-, 20- and 10-year events, respectively, before 2050 throughout the analyzed subregions.*

93. Line 407: please clarify what 'also' refer to.

We dropped "also".

94. Line 410: What about the data of the subregions and land cover (Fig. 1 and A1)?

    We added the sources for the subregions and landcover in the data availability section.

    Old: *The datasets used in this study can be found in the following repositories: CRCM5-LE: https://www.climex-project.org/de/datenzugang and ERA-5 based FWI: DOI: 10.24381/cds.0e89c522(31.01.2023).*

    New: *The datasets used in this study can be found in the following repositories: CRCM5-LE: https://www.climex-project.org/de/datenzugang, ERA-5 based FWI: DOI: 10.24381/cds.0e89c522 (31.01.2023), sub-regional division: https://www.lfu.bayern.de/natur/ naturraeume/index.htm, landcover data from Copernicus Land Monitoring Service: https://land.copernicus.eu/pan-european/corine-land-cover/clc2018.*

95. Figure C1: Why do you use 95$^{th}$ percentile and not 90$^{th}$ percentile as done in the remaining analysis?

    Thank you for this remark. We aimed to show more extreme results in the sensitivity analysis and therefore decided to use the 95th percentile.

96. Why number the Figures A1, B1 and C1 instead of A1, A2 and A3 as is normally done?

    Thank you for pointing this out. We fixed this overleaf template issue and numbered the Figures in the appendix according to your suggestion.

---

## Author Comment (AC2)

**Reviewer 2**

Miller et al. (2023) present a detailed analysis of the climate change impacts on fire weather across a study region in Central Europe that is historically not fire-prone. They accomplish this by using the Single Model Initial-Condition Large Ensemble (SMILE) of a regional climate model to: a) study the temporal and spatial trends in the Fire Weather Index (FWI), a commonly used indicator of fire weather; b) disentangle the contribution of natural variability from climate trends in the median and extreme percentiles of the FWI as inferred from two metrics: time of emergence (TOE) and temporal evolution of the current fire danger return period.

Overall, I found the manuscript to be well-written, and I appreciated the clear presentation of the analysis techniques and results throughout the text. The subject matter is quite important and within the purview of NHESS's scope. However, I think there are several areas where the authors could improve the discussion in the manuscript, either through clarification of confusing statements or by illustrating their argument with an additional figure or two. Once these changes are incorporated, I would be happy to review the manuscript's suitability for publication. Please find my comments listed below.

Thank you very much for editing our manuscript and critically reflecting on our results. We highly appreciate your constructive comments and implemented your feedback.

*Comments:*

- L130: The phrasing of this statement lends me to believe that FWI is calculated using antecedent weather over the previous 52 days. This is, however, not the case based on the documentation for FWI available here: https://cfs.nrcan.gc.ca/publications?id=19927

  Thank you for raising your concerns regarding this statement. We reread the documentation of the FWI (provided in your link) and realized that the time delay of 52-days in the Drought Code refers to the drying rate and not the antecedent weather conditions. We revised the section and dropped the statement about the previous 52-days.

  Old: *The CFFWIS uses meteorological conditions of the atmosphere on the day of interest (temperature, relative humidity, wind speed at noon and 24-h accumulated precipitation) and antecedent weather conditions up to 52 days to estimate fire behavior and fuel moisture (Van Wagner, 1987).*

  New: *The CFFWIS uses meteorological conditions of the atmosphere on the day of interest (temperature, relative humidity, wind speed - all at noon and 24-h accumulated precipitation) and antecedent weather conditions represented in fuel moisture codes to estimate fire behavior and fuel moisture (Van Wagner, 1987).*

- L146: I appreciated the authors quoting the units for FWI. However, these units are conspicuously missing in the relevant tables and figures in the rest of the text (Table 1, Figure 4, 5, 6, 7)

  Thank you for comment, which we highly value. According to Van Wagner (1987), the unit of the FWI is *I* or *HWR* (fire intensity represented by energy output rate (H) per fuel consumed per unit area (W) and rate of spread (R)). We could add this unit to the tables and figures of our manuscript, but we think this will add confusion to readers which are not familiar with the field. However, we updated the labels in the Figures to FWI to indicate that the color bar refers to the FWI and not to one of the FWIs subindices.

- L204: "...and 99th percentiles [of the FWI] in the present climate period" (missing text)

Thank you for pointing this out - we added "of the FWI" in L204.

Old: *We calculate the return periods on the basis of the 90th, 95th, 98th and 99th percentiles of the present climate period (1980–2009) to account for 10-, 20-, 50- and 100-year FWI events in the four subregions.*

New: *We calculate the return periods on the basis of the 90th, 95th, 98th and 99th quantiles of the FWI of the present climate period (1980–2009) to account for 10-, 20-, 50- and 100-year FWI return levels in the four subregions.*

- L208: "We then compute the non-exceedance probability of the present percentiles given the future cumulative distribution" -- Present percentiles and future cumulative distribution of what quantity? The writing here can be improved, in general, to clarify whether the percentiles are with respect to all 30 years of the whole ensemble or of one model within the ensemble.

Thank you for highlighting the missing description. Since this section was highlighted by RC-1 (general comment 1) as well, we edited this section carefully and provided a more detailed description of how we derive the return periods:

New: *From 1980 to 2099, we create centered 30-year windows for each ensemble member to determine its empirical distribution function and the FWI quantiles corresponding to the different return periods of the present climate period. We map the non-exceedance probability of the present percentiles given the empirically derived cumulative distribution of each member. From the non-exceedance probability, we estimate the return periods using the function T = μ/(1 − p) where T is the return period, μ is the inter-arrival time between two events (1/183 days in a fire season) and p is the non-exceedance probability (Coles, 2001). We derive p from the rank r with p = r/n (2) where n is the total sample size by using the rv_histogram.cdf function of the Scipy package in Python (Virtanen et al., 2020). Due to the centred window approach, the first full 30-year window is 1995 and the last full 30-year window is 2084. Therefore, we crop the resulting time series to 1995 to 2084.*

- L240-241: "This finding indicates that the distribution of the FWI extremes resembles the distribution of the FWI median." -- This statement seems unintuitive: wouldn't the distribution of median and extreme FWI (which contains temperature as a predictor) diverge in a warming world? Perhaps this is an artifact of how the TOE is calculated with SMILEs and there is not enough variability, or that 90th percentile isn't extreme enough in the future. It would be great to see a version of Fig. 7 with the 95th and 99th percentile as well.

Thank you for your comment, which we highly appreciate. Indeed, it is counter-intuitive, that the variability (turquoise shading in Fig. 7) appears to be the same for the 50th and 90th percentile. Variability should increase for the 90th percentile and yet, it looks smaller than for the 50th percentile. The reason for this is that we initially did not display the results for the two percentiles on the same scale. When plotted on the same scale (Figure 9 in this response to the reviewer), it becomes apparent that indeed and as expected the variability is substantially larger for the 90th than for the 50th percentile. We adjusted Fig. 7 in the manuscript to a common y-axis between the 50th and 90th percentile (s. below), which clearly shows that the distribution of median and extreme FWI is not the same, as stated previously. We therefore removed this line and updated Fig. 7.

[Figure]

*Trends of the median ([1], 50th percentile) and extreme ([2], 90th percentile) FWI between 1980 and 2099 differentiated by subregion: (a) Alps, (b) Alpine Foreland, (c) Southgerman Escarpment, (d) Eastern Mountain ranges. The ensemble mean trend is derived on a fire season basis and represented by solid pink lines smoothed over a 30-year window. The ensemble's standard deviation is represented by shaded blue areas. Black solid and dashed lines represent the ensemble mean and spread of the present climate period (1980–2009). The TOE, marked with a pink dot and year annotation, is reached when the ensemble mean (pink line) crosses the upper boundary of the ensemble standard deviation in the present climate period (black dashed line)*

- Figure 9: Why is the ensemble mean of the 100-year return period only about ~75-80 years for all 4 subregions?

We created centered 30-year windows (between 1980 and 2099) for each member to determine the FWI percentiles corresponding to the different return periods of the present climate period (all 50 members). The first full 30-year window is 1995 and the last full 30-year window is 2084. Therefore, the ensemble mean of the 100-year return period for the present all member pool is 100 in the year 1995. This is not shown in the Fig. 9 of the preprint. We updated the figure and methods section (s. your comment on L208) accordingly.

[Figure]

*Changes in present return periods (1980–2009) of the 90th, 95th, 98th and 99th FWI percentile throughout the 21st century (1995–2084), distinguished by subregion: (a) Alps, (b) Alpine Foreland, (c) Southgerman Escarpment, (d) Eastern Mountain Ranges. The thick solid line represents the CRCM5-LE mean, while thin lines represent the ensemble members.*

- L293: "...which has to be discussed..." -- improve phrasing.

We rephrased this section in correspondence to RC-1 (comment 68).

Old*: Another aspect, which has to be discussed, is the strong tiling pattern visible in figure 5 [2] in the months June and August. This tiling pattern is already visible in the extreme values of the input variables. We provide a sensitivity analysis of the FWI in the Appendix (s. figure C1), where the tiling occurs for temperature and relative humidity in the 95th 295 percentiles as well. The pattern correlates with invariate fields from the geophysical baseline parameterization of the CanESM2, e.g. bedrock depth. Over the areas where the strong tiling occurs, bedrock depth is about 5m. The water storage potential of the ground is especially high in this area compared to its surrounding areas with an average bedrock depth between 1 or 2 meters. Such high storage potential can affect evaporation and leads to a higher cooling in areas with high bedrock depths which results in lower 300 temperatures and higher relative humidity.*

New: *Though the CRCM5 reproduces the response structures much finer than CanESM2 and adds robust high-resolution features (Böhnisch et al., 2020), we find in the northern parts of the study area tiling patterns corresponding to the geophysical baseline parameterization of the CanESM2 (see Figure A3). The tiling occurs in the sensitivity analysis provided in Figure A.3 for temperature and relative humidity in the 95$^{th}$ percentile, when the FWI is calculated with a factor of two for temperature and relative humidity. The pattern correlates with the bedrock depth of the CanESM2, which might affect the water storage potential of the ground. Over the areas where the tiling occurs, bedrock depth is about 5m, which is relatively high in comparison to the surrounding areas with an average bedrock depth between one or two meters. Such high storage potential can affect evaporation and leads to a higher cooling in areas with high bedrock depths which results in lower temperatures and higher relative humidity. The tiling occurs only under very extreme FWI conditions (95th percentile) and might lead to an overestimation of our results in the extreme FWI (90th percentile) for the Southgerman Escarpment.*

- L293: "...tiling pattern visible in figure 5 [2]..." -- [2] seems to be a typographical error.

  The tiling pattern is not visible in Figure 5 anymore, because we changed to a discrete color scale (s. RC-1 comment 58).

---

## Author Comment (AC3)

**Reviewer 3**

*General comments*

This paper investigates a regional SMILE (Single Model Initial-Condition Large Ensembles) of the Canadian regional climate model version 5 (CRCM5-LE) over Central Europe (Hydrological Bavaria) under the RCP8.5 scenario from 1980 to 2099, to analyze fire danger trends in a currently not fire-prone area. This evaluation of fire danger (vs current climatic conditions) uses Canadian Fire Weather Index (FWI), and the 3-hourly meteorological data from the large ensemble of available CRCM5-LE simulations. The authors demonstrate that this ensemble (at 0.11°) is a suitable dataset to disentangle climate trends from natural variability in a multivariate fire danger metric. Various results show the increase in the median and extreme percentile of the FWI in the northern parts of the study area (in July and August). The southern parts of the study region are less strongly affected, but time of emergence (TOE) is reached there in the early 2040's. In the northern parts, the climate change trend exceeds natural variability in the late 2040's. In the future, a 100 year (return period) FWI event will occur every 30 years by 2050 and every 10 years by 2099. This study is of a strong interest in order to help the refinement of fire management strategy to reduce the consequences of such forest fires, and to improve the preparation or adaptive capacity knowing the potential changes of this natural hazard under ongoing climate change. The article is well written, and well-articulated in term of scientific findings and presentation of main outcomes.

I will suggest to add insights or discuss limitations from the use of one single RCM driven by one specific GCM (i.e. CanESM2) whatever the number of ensemble runs used, as systematic biases from the driving GCM can influence the downscaling simulations and derived products (ex. FWI). For example, as noted in various studies, biased atmospheric circulation features due to coarse-scale resolution (ex. around 2.8° for the CanESM2 model) and/or missing orographic drag, sea surface temperature simulated features, etc. which affect the simulated blocking features (see Pithan et al., 2016; Schiemann et al., 2017; Davini and d'Andrea, 2020) or atmospheric circulation variability responsible for the occurrence of climate extremes over Europe (see Faranda et al., 2023). As revealed in the recent work of Faranda et al. (2023), atmospheric circulation changes modulate extreme events already in the present climate in Europe, and summer heatwaves as well as large regional and seasonal changes in precipitation and surface wind, i.e. hazards or meteorological variables responsible for the occurrence and severity of fire danger (variables used to compute the FWI indices). Also, as shown in Zappa et al. (2014), CanESM2 tends to have one of the largest track density biases for extratropical cyclones among CMIP5-GCMs, as well as in blocking frequency biases over both Norwegian Sea, and central Europe (see their Figure 3). These two features of atmospheric circulation variability play a key role in the occurrence of both anomalies of temperature and precipitation across the study area. Faranda et al., (2023) and Strommen et al. (2019) strongly argue to use at least three or more ensemble members to deal with the importance of the regional response to anthropogenic forcing (Corti et al., 1999; Palmer, 1999), representing these atmospheric regimes correctly whatever the GCM. Also as noted in Deser et al. (2020), the use (in future works) of large ensembles from different GCMs will give new insights into uncertainties due to internal variability versus model differences. In fact, concerning the so-call internal variability (or natural variability mentioned in the paper that it is partly evaluated from the CRCM5-LE), the current study cannot consider using one single ensemble from one RCM-GCM matrix, structural variability from the differences in (GCM) model formulation, including physics and parameterization, resolution, etc. This structural variability strongly affects the climate change responses at the regional or local scale (uncertainties in the dynamical downscaling simulations depend on the driving GCM; see different studies from the EURO-CORDEX project).

In summary, after having recall the shortcoming and include nuance about the robustness of the climate change signals for the FWI indices across the area, using one single ensemble sample from one combination of RCM-GCM, the paper is sufficiently relevant and scientific documented (sound) to be published after minor revisions. Please see also my specific comments below.

We appreciate the careful revision of RC-3 on our manuscript. We clarified the remarked sections specifically and improved the overall description of our modelling set-up, which is a regionally downscaled single-model initial condition large ensemble (SMILE).

We agree with RC-3 that our modelling setup cannot represent model spread because our analysis focuses on a regional SMILE of one GCM-RCM combination only. The reason for this model choice is that we want to overcome biases in atmospheric circulation features due to coarse-scale resolution, e.g. orographic drag, by using a regionally and dynamically downscaled SMILE. Rather than on in-between model spread, this study focuses on the quantification of internal variability, which requires the use of a SMILE (s. Deser et al. 2020). This is also proposed by Faranda et al. (2023). The SMILE used in our study (CRCM5-LE) was compared to the CORDEX-Family by Von Trentini et al. (2019). However, to date only 2 other regional SMILES for the area of interest exist, but they differ in the study domain and spatial resolution (s. Von Trentini 2020). The implications of the regional downscaling of the CanESM2-LE using the CRCM5 was investigated by Böhnisch et al. (2020). The study of Böhnisch et al. (2020) shows that "important large-scale teleconnections present in the driving data propagate properly to the fine-scale dynamics in the RCM". Further, the studies of Mittermeier et al. (2019 and 2021) demonstrate that the CRCM5-LE is capable to quantify large scale pressure patterns which lead to heat-waves and extreme precipitation (i. e. Vb-Cyclones).

Since we find this was not clearly enough emphasized in our initial manuscript, we edited the introduction and discussion section and reflected the strengths and weaknesses of the CRCM5-LE better by adding new paragraphs:

New Paragraph (Introduction): *In this study, we use the CRCM5-LE, a regionally dynamically downscaled high-resolution SMILE (0.11° grid cell size) nested into the CanESM2-LE (Fyfe et al., 2017), to disentangle climate change induced fire danger trends from natural variability over heterogeneous landscapes in Central Europe. Benefits of using a regional SMILE are the better spatial representation of climatic patterns for regional- and local-scale analyses, i.e. NAO (Böhnisch et al., 2020) and pressure patterns leading to extreme precipitation (Mittermeier et al., 2019) or drought and heat events (Mittermeier et al., 2021) in the study area. In comparison to the global CanESM2-LE, the regional CRCM5-LE replicates response structures more precisely and incorporates durable high-resolution geographical characteristics that are prominently apparent in the ensemble mean.*

New Paragraph (Discussion): *While the SMILE-setup used in our study allows us to estimate the internal variability and the forced response of the selected climate change scenario (Suarez-Gutierrez et al., 2021; Deser et al., 2012), it can not account for the structural uncertainty of the climate model (Deser et al., 2020), which can be assessed only in multi-model studies (i. e. Fargeon et al. (2020)). In order to quantify both, internal variability and structural uncertainty, multiple SMILEs as provided by the "Multi-Model Large Ensemble Archive" (MMLEA) (Deser et al., 2020) should be used. However, all SMILES in the MMLEA are based on Global Climate Models (GCMs) with a spatial resolution ranging between 2.8° and 0.9° (Deser et al., 2020). On a regional and local scale, a higher spatial resolution is needed to quantify climate change impacts. For Europe, only two other dynamically downscaled SMILEs from Regional Climate Models (RCMs) exist: The 16-member EC-EARTH-RACMO ensemble at 0.11° (Aalbers et al., 2018) and the 21-member CESM-CCLM ensemble at 0.44° grid cell size (Fischer et*

*al., 2013; Brönnimann et al., 2018). The three models differ in their study domain (EC-EARTH-RACMO) and spatial resolution (CESM-CCLM) from the CRCM5-LE (Wood, 2023; von Trentini et al., 2020).*

*Specific comments:*
Please be consistent when you use RCP8.5 (without "space" between P and 8.5) in the text.

We unified to RCP8.5 and RCP2.6 without a blank space.

Abstract:

Please add few words or one sentence considering the need to use larger downscaling ensemble from different GCMs in order to develop more robust climate change signals for all meteorological variables used to compute the FWI indices (further work).

Thank you for your comment. We agree that this should be nuanced in the abstract and therefore added a new sentence to the abstract.

New: *To date, only a few dynamically downscaled regional SMILEs exist, although they enhance the spatial representation of climatic patterns on a regional or local scale.*

Introduction:

**Line 22:** Please add (Canada) after British Columbia.

We added *(Canada)* after British Columbia.

**Line 63:** Please nuance this statement, as natural variability of the climate system is not fully represented by one single model initial-condition large ensemble, as a GCM generates a simplification of a complex reality (i.e. the climate system) and includes structural variability and biases that we need to consider in any downscaling exercise (see Strommen et al., 2019; Deser et al., 2020; and recommendations or Plausibility criteria in the new CORDEX-CMIP6 in Sobolowski et al., 2023).

We added a statement highlighting that a SMILE – while capturing internal variability - does not allow for the quantification of the structural uncertainty. For this latter purpose, a multi-model large ensemble has to be used.

New: *This limitation, i.e. the confusion of natural variability and model uncertainty for changes in fire danger estimates in regions with currently temperate climate, can be overcome by evaluating climate model simulations derived from a single model initial-condition large ensemble (SMILE). SMILEs represent an ensemble of simulations derived using one single climate model started at different initial conditions. The ensemble spread between the different SMILE members provides a robust estimate of the internal variability, from which the forced response of the climate change scenario (i. e. RCP 8.5) can be estimated by averaging over the SMILE members (Deser et al., 2020). Therefore, SMILEs are capable to robustly sample extreme events and their probability distribution (Maher et al., 2021). While SMILEs allow for the quantification of internal variability, they do not enable a quantification of model uncertain (Deser et al., 2020).*

Data and Methods

**Line 96:** As mentioned in Fargeon et al. (2020) and many other studies, bias correction alters the physical consistency of modelled climate and meteorological variables in particular at high frequency (ex. sub-daily values). Quantile mapping makes strong assumptions regarding bias stationarity and can break the co-variation between climatic variables, in particular at high frequency or meteorological scale (i.e. that is the case here when computing the daily FWI indices). Can the authors provide some

insight about these drawbacks or physical consistency among meteorological variables after bias correction and the implication of this in computing FWI indices?

Thank you for this remark. We agree that univariate bias correction methods may not perfectly represent the covariation between variables. This is especially relevant for multivariate indices like the FWI used in this study. Zscheischler et al. (2019) analyze the effect of univariate bias adjustment for multi-variate hazards and discuss different studies arguing for and against the need of multi-variate bias correction methods. For example, Yang et al. (2015) argue that a univariate bias correction is sufficient, while Cannon et al. (2018) propose the opposite. They find that "we cannot draw the general conclusion that multivariate bias adjustment is not necessary in any case from individual, typically regional, studies" and "it is difficult to pin down under which exact circumstances univariate bias adjustment might fail".

We added a section to the discussion, where we discuss the advantages and disadvantages of univariate / multi-variate bias correction:

New Paragraph:

*Correcting the bias between climate model data and observation data is often an inevitable step in climate impact studies (Piani et al., 2010). The CRCM5-LE was bias adjusted using the quantile mapping approach after Mpelasoka and Chiew (2009), for each of the FWI variables separately (Poschlod et al., 2020). Univariate bias correction methods, as used in our study, can change the co-variation between multiple variables (Zscheischler et al., 2019). Changing the co-variation through bias-adjustment can affect the analyses of fire weather indices, like the FWI, which is why there have been calls for the usage of multi-variate bias correction methods (Cannon, 2018). Despite these concerns, (Yang et al., 2015) showed that bias-adjusting multiple variables separately was sufficient to study fire weather changes in Sweden and Zscheischler et al., (2019) state that the reasons for univariate bias adjustment to fail are hard to specify. Furthermore, multivariate bias correction is non-trivial and while fixing co-variation issues lead to other problems e.g. with temporal or spatial dependencies (i. e. Vrac, 2018). In this regard, we assume that the bias correction applied on the CRCM5-LE is appropriate.*

**Line 100:** "FWI extremes are significantly better…": Yes, but these FWI extremes are physically coherent and consistent with meteorological fields?

Together with your previous comment, we discuss this in the new discussion paragraph. The studies of Yang et al (2015) and Cannon et al. (2018) have shown that bias-corrected climate model output better reflects observed fire danger. We therefore adjusted this section:

Old: *Bias corrected data are commonly used for projections of fire weather indicators like the FWI (e. g. Yang et al. (2015), Kirchmeier-Young et al. (2017), Ruffault et al. (2020), Fargeon et al. (2020)), because frequencies of FWI extremes are significantly better represented than in non-bias-corrected climate data (Yang et al., 2015).*

New: *Bias corrected data are commonly used for projections of fire weather indicators like the FWI (e. g. Yang et al., 2015; Cannon, 2018; Kirchmeier-Young et al., 2017; Ruffault et al., 2017; Fargeon et al., 2020), because frequencies of FWI extremes are significantly better represented than in non-bias-corrected climate model data (Yang et al., 2015).*

Study area

**Line 112:** Please can the authors provide some reference from which dataset these (climatological) values come from ? E-Obs, …?

We added a notation that the data is derived from the meteorological SDCLIREF dataset. This Sub-Daily Climatological REFerence dataset (SDCLIREF) was created within the scope of the project that this study is based on (ClimEx). It combines hourly and disaggregated daily station data and is described in detail by Brunner et al. (2021).

New:

*The ClimEx project's own meteorological Sub-Daily Climatological REFerence dataset (SDCLIREF) served as an observation reference served. It combines hourly and disaggregated daily station data and is described in detail by Brunner et al. 2021.*

*[…]*

*The mean precipitation over the study areas increases from north to south, with annual precipitation sums between 500 and 1100 mm for the South German Escarpment, 1000 mm for the Eastern Mountain Ranges, 1500 and 2500 mm in the Alpine Foreland and 1000 and 2000 mm in the Alps, according to the SDCLIREF observation dataset for the present climate period between 1980 and 2009 (present).*

The Canadian Fire Weather Index

**Line 125:** Please correct "… assess…".

We corrected "assesses" to "assess".

Estimating Fire Danger using the CRCM5-LE

**Line 170:** "…, this does affect the climate change impacts assessment…". Yes, but the CRCM5 ensemble seems to underestimate the internannual anomalies of FWI that we see from the reference (ERA5) database. Please can you comment this, as the year 2002 seems to be not in the range of below 75th percentiles of the observed FWI across Europe but rather on more extreme side? This underestimation of interannual FWI anomaly can be due to the debiased method which has an effect on the decreasing year to year variability of each of the ensemble simulations?

Thank you for pointing this out. We realized that the outlier in the ERA-5 Dataset is referring to the year 2003 and the time periods between our comparison differed (ERA-5: 1981-2010, CRCM5-LE: 1980-2009). We remade the figures by using a unified time period (1980 – 2009). The year 2003, especially the summer months, were affected by an extreme heat wave in Europe, which is reflected by the high FWI value in the ERA-5 dataset. However, the CRCM5-LE data is modeled data and does not contain observed values. Figure 3 shows, that the 2003 heatwave lies below the 75th percentile of our single-model initial-condition large ensemble and is well situated in the ensemble spread. Following the framework of Suarez-Guiterrez et al. (2021) (https://doi.org/10.1007/s00382-021-05821-w), we interpret that the CRCM5-LE overestimates internal variability, because most observation points lie between one standard deviation in the ensemble and all observation points lie between the 25th and 75th percentile of the ensemble. For this reason, we cannot precisely answer how the bias correction should affect an underestimation of interannual FWI anomalies.

We added to this paragraph a remark, that the ensemble overestimates interannual variability:

New: *The CRCM5-LE overestimates the internal variability in comparison to the reference dataset (s. Figure 3 following the framework of Suarez-Guiterrez et al. (2021) to evaluate internal variability in SMILEs.)*

[Figure]

*Median FWI for the CRCM5-LE mean (thick blue line) and standard deviation (light blue shading) in comparison to the reference dataset of Vitolo et al. (2020) marked pink (X for values, lines for deviation from the CRCM5-LE mean). Top and bottom blue lines mark the 25th and 75th percentile of the CRCM5-LE.*

Further we discussed this more closely in the answer to your comment (RC-3) on Line 302.

Discussion

**Line 278:** "… next few decades…". This mean 2080s? Please be precise.

We changed "*next few decades*" to "*until the end of the 21$^{st}$ century*" to be precise that we mean our whole observation until the end of the century.

Data basis

**Line 302:** "…uncertainties related to emission scenarios and the chosen climate model". You do not discuss this point (i.e. choice RCM or single RCM-GCM), please provide some insights as suggested in the general comments.

We changed this section in correspondence to comment 69 of RC1 and added a section that explains the difference between the CRCM5-LE and other CORDEX models in terms of precipitation and temperature.

Old: *Lastly, the SMILE used in this study assesses climate change signals against internal climate variability but does not consider uncertainties related to emission scenarios and the chosen climate model. However, the choice of emissions scenarios also introduces uncertainty. Fire danger increase is projected and analyzed only for the RCP8.5 scenario, which represents the strongest temperature increase scenario. It remains open and subject to policy making if this scenario becomes reality. Arnell et al. (2021) find that reducing emissions to a level consistent with an increase of a global mean*

*temperature of 2°C, i.e. RCP 2.6, reduces fire danger substantially compared to RCP8.5. This finding implies that our change estimates represent an upper boundary of changes in fire danger expected in the future.*

New: *The SMILE used in this study allowed us to identify climate change signals in the FWI and compare them against internal climate variability. However, SMILES do not consider scenario and structural, model specific uncertainty, because only one scenario and one climate model are usually available. In comparison to the CORDEX ensemble, the CRCM5-LE shows drier and warmer climate change signals for temperature and precipitation (Von Trentini et al., 2019). These characteristics of the CRCM5-LE are in line with the results from the validation (see Figure 3) and indicate an overestimation of our results. Fire danger increase is projected and analysed only for the RCP8.5 scenario, which represents the strongest temperature increase scenario. Arnell et al. (2021) find that reducing emissions to a level consistent with an increase of a global mean temperature of 2°C, i.e. RCP2.6, reduces fire danger substantially compared to RCP8.5. Due to the strong warm and dry climate change signal in the CRCM5-LE (Von Trentini et al., 2019), we assume that our change estimates represent an upper bound of changes in fire danger expected in the future.*

Further, we added a paragraph discussing the benefits and downsides of using a regionally downscaled SMILE.

New: *All SMILES in the MMLEA are based on Global Climate Models (GCMs) with a spatial resolution rang- ing between 2.8 and 0.9° (Deser et al., 2020). On a regional and local scale, a higher spatial resolution is needed to quantify climate change impacts. For Europe, only two other dynamically downscaled SMILEs from Regional Climate Models (RCMs) exist: The 16-member EC-EARTH-RACMO ensemble at 0.11° (Aalbers et al., 2018) and the 21-member CESM-CCLM ensemble at 0.44° grid cell size (Fischer et al., 2013; Brönnimann et al., 2018). The three models differ in their study domain (EC-EARTH-RACMO) and spatial resolution (CESM-CCLM) from the CRCM5-LE (Wood, 2023; von Trentini et al., 2020).*

Spatio-Temporal Trends and Variability

**Line 359:** "… on the whole year instead of the summer season only": Potential avenue will be to use take into account the snow cover season or overwintering conditions, based on cumulative precipitation during the cold season, as used in Canada (see McElhinny et al., 2020).

Thank you for this comment, we indeed did not discuss the overwintering option of the Drought Code. We added this to this section, which was already edited in RC-1 comment 80:

Old: *For the Southern Alps, Wastl et al. (2012) identified the main fire season between December and April because of low precipitation and missing vegetation cover in the winter half year. Therefore, future studies assessing changes in fire danger in the Alps should focus on the whole year instead of the summer season only.*

New: *For the Southern Alps, Wastl et al. (2012) identified the main fire season between December and April because of low precipitation and decreased fuel moisture outside the vegetation period in the months December to April (Conedera et al. 2020). With respect to the increasing altitude of vegetation, increasing length of the vegetation period and decreasing snow cover (Rumpf et al, 2022), future studies assessing changes in fire danger and fire events in the Alps and other temperate climate regions should consider analyzing the whole year instead of the summer months only. If the FWI can not be calculated continuously, the long term moisture deficit represented by the Drought Code (DC) should be "overwintered" in further studies. Overwintering in case of the DC means, that the value of the DC*

*in the new fire season is set to the last value of the DC in the previous season (Wang et al., 2017). However, the FWI can not capture these land cover and vegetation specific changes and therefore other methods should be considered to quantify fire danger outside of the summer period (see e.g. Pezzatti et al. 2020.)*

**Line 373:** "…or the slight overestimation of the CRCM5-LE…": Again, this can be due to the lack or limited internannual variability in the debiased CRCM5-LE variables? Please comment slightly on this issue. Line 374: " … a substantial larger database…". Yes, but this is a single model (CRCM5) driven by an ensemble of one GCM (CanESM2), as in Fargeon et al. (2020) they use 2 RCMs driven by 3 different GCMs. Please nuance this statement.

We value your comment regarding the overestimation of natural variability of the CRCM5-LE. This section was already revised in RC-1 (comment 82.). However, we reedited the section to emphasize more strongly that the CRCM5-LE is a regionally downscaled single-model initial-condition large ensemble (SMILE) of a GCM (CanESM2-LE), resulting in 50 climate realizations of a spatial resolution of 0.11°.

Old: *The CRCM5-LE used in our study embodies a substantially larger database than the database used by Fargeon et al. (2020) thanks to its SMILE-setup, which helps to better represent natural variability.*

New: *Due to its SMILE setup, the CRCM5-LE used in our study embodies a substantially larger database (50 members) and is able to quantify natural variability in contrast to the climate multi-model database (5 members) used by Fargeon et al. (2020).*

**Line 375:** "… which helps to better represent natural variability": See my previous remarks, natural variability is more complex that internal variability extracted from one single RCM-GCM matrix, as at least you need to consider more range of boundary conditions, from as many as possible GCMs as those are the main source of uncertainties in particular from the atmospheric circulation over Europe pointed out by Faranda et al. (2023).

Thank you for your comment, which we highly appreciate, because it demonstrates that our study set-up is not clearly described. The CRCM5-LE used in this study (s. Leduc et al. 2019) is a single-model initial condition large ensemble, which is widely used on a global scale (e. g. Deser et al. 2020, Suarez-Guiterrez et al. 2021) to distinguish internal / natural variability from the forced response of a climate model simulation. Faranda et al. (2023) state that they would use "initial condition large ensembles to separate forced signals from internal variability in the context of our analog analysis" in future work. However, our study does not aim to quantify model uncertainty but to study climate variability. We added a section on this to the discussion (s. your comment on the abstract and line 62, RC-3 Abstract, RC-3 Introduction Line 63). We hope the changes in the previous comments clarify the modelling setup in our study.

New Paragraph (Discussion): *While the SMILE-setup used in our study allows us to estimate the internal variability and the forced response of the selected climate change scenario (Suarez-Gutierrez et al., 2021; Deser et al., 2012), it can not account for the structural uncertainty of the climate model (Deser et al., 2020), which can be assessed only in multi-model studies (i. e. Fargeon et al. (2020)). In order to quantify both, internal variability and structural uncertainty, multiple SMILEs as provided by the "Multi-Model Large Ensemble Archive" (MMLEA) (Deser et al., 2020) should be used. However, all SMILES in the MMLEA are based on Global Climate Models (GCMs) with a spatial resolution ranging between 2.8° and 0.9° (Deser et al., 2020). On a regional and local scale, a higher spatial resolution is needed to quantify climate change impacts. For Europe, only two other dynamically downscaled SMILEs from Regional Climate Models (RCMs) exist: The 16-member EC-EARTH-RACMO ensemble at*

*0.11° (Aalbers et al., 2018) and the 21-member CESM-CCLM ensemble at 0.44° grid cell size (Fischer et al., 2013; Brönnimann et al., 2018). The three models differ in their study domain (EC-EARTH-RACMO) and spatial resolution (CESM-CCLM) from the CRCM5-LE (Wood, 2023; von Trentini et al., 2020).*

**Line 377:** " … fire danger are robust…": From the ensemble runs used (i.e. link to the sample size or RCM-GCM matrix). Please nuance this statement.

Thank you for your comment. We changed the sentence to emphazise that we used a regionally downscaled SMILE:

Old: *While Fargeon et al. (2020) point out that fire danger increases are hard to distinguish from natural variability in northern France in multi-model ensembles, we demonstrate using a SMILE that increases in fire danger are robust for Central Europe.*

New: *While Fargeon et al. (2020) point out that fire danger increases are hard to distinguish from natural variability in northern France in multi-model ensembles, we demonstrate that increases in fire danger can robustly be quantified for Central Europe by using a regional SMILE.*

Conclusion

**Line 404:** "We accept all of the three hypotheses…": Please be more explicit and comment about these, in particular H2 and H3.

We appreciate your comment, which was also remarked in a similar way by RC-1 (comment 92). We restructured the conclusions to follow the different research questions/hypotheses.

Old: *The strongest increases and most hazardous developments are observed North of the river Danube in the summer months July and August for the subregions South German Escarpment and Eastern Mountain Ranges. Regions south of the Danube (Alps and Alpine Foreland), are less strongly affected by changes in the FWI but increases are still significant. Further, we find that the FWI has a stronger variability for regions with heterogeneous terrain (i.e. the Alps and the Eastern Mountain Ranges) than for regions with less complex terrain (i. e. Alpine Foreland and Southgerman Escarpment). The time of emergence (TOE) is reached in all subregions of the study area before 2050 and the return period of a present 100- year event shifts towards a 10-year event by 2090. We accept all of our three hypotheses, stated in the introduction (chapter 1). Our results reveal more serious developments than assumed in the original hypotheses.*

New: *Our results provide clear answers to our initially proposed research questions. They demonstrate that fire danger increases significantly throughout the study area. We find the strongest changes and highest fire danger levels north of the river Danube in the summer months July and August for the subregions South German Escarpment and Eastern Mountain Ranges. Our results also show that the time of emergence (TOE) is reached in all subregions before 2050. Further, we showed that not only the mean but also the lower boundary of the running mean, represented by the CRCM5-LEs standard deviation, exceeds the upper boundaries of the present climate (1980 - 2009) standard deviation before 2099 in all subregions for the 90th FWI percentile. Last, our findings highlight that the return period of present 100-year events shifts towards 10-year events by 2090 and the return periods for 100-, 50- and 20-year events shift to 50-, 20- and 10-year events, respectively, before 2050 throughout the analyzed subregions.*

References

Line 531: The reference Separovic et al. (2013) is not at the right place in the list.

We corrected the misplacement of the reference Separovic.

---

## Author Response (AR1)

Dear Editor,

Thank you very much for your and the reviewers' assessment of our manuscript. We revised the manuscript according to the reviewers' comments with a particular focus on improving the discussion of our climate model setup by adding paragraphs about model uncertainties and uncertainties related to the bias correction applied. Further, we emphasized the added value of our study in the introduction by highlighting the benefits of using a regional Single Model Initial Condition Large Ensemble (SMILE). As requested by the reviewers, we improved the interpretability of our visualizations by adding a legend to Figure 3, changing to a discrete color scheme in Figure 4 and adding a second y-axis to Figure 8. We hope that you find the revised version of this manuscript suitable for publication in NHESS. Thank you very much for your re-assessment.

Best regards,

Julia Miller

**Reviewer 1:**

The manuscript "Climate change impacts on regional fire weather in heterogeneous landscapes of Central Europe" by Miller et al. presents a study on climate change impacts on the fire danger index FWI in hydrological Bavaria in central Europe, using a single model initial-condition large ensemble (SMILE) data set and the RCP 8.5 emission scenario. Changes in FWI are evaluated in terms of fire danger levels over the whole region, as well as on a sub-regional scale. The study provides new and interesting knowledge of projected changes in FWI in the studied region, and the topic is suitable for the journal. Strengths of the manuscript include the multiple approaches applied to investigate the hypotheses, and the clearly communicating figures. The manuscript has potential but is not at the required level for a scientific paper in its current state. Parts of the analyses are wrong, methods are not clearly presented, the discussion is not clearly presented, and references do not always reflect work supporting the claims. I would recommend all authors to carefully revise the manuscript and correct and clarify where necessary. General and specific comments are provided below.

Thank you very much for taking the time to write this very detailed and constructive review, which we highly appreciate. We are glad that you value our work and are happy to take your comprehensive feedback into account. We think your comments shape the manuscript in a very positive way and revised the paper carefully according to your feedback. We clarified unclear descriptions in the methods section, revised and updated references, and edited and extended the discussion. In addition, we carefully checked the manuscript for clarity.

*General comments*

1.  The method to derive the return period is wrong. Thus, the analysis using return periods must be omitted or corrected and clarified. The applied temporal resolution is not stated, but the method is wrong regardless of the applied resolution. If all daily (or monthly) data in each year is used to extract the 99th percentile value, this value represent the 100 days (or 100 months) return period and not 100 years as stated in the manuscript. See e. g. Camuffo et al. (2020): https://doi.org/10.1007/s11600-020-00452-x. If instead the maximum value in each year was extracted, the period (30 years, i.e. 30 values) is too short to extract the 99th.

    Thank you for highlighting the need to clarify the method used for return period calculation. We used a Single Model Initial-Condition Large Ensemble, where each of the 50 realizations represents an equally likely climate within the used climate model. Therefore, we pool over the

entire 50-member ensemble to derive the quantiles from the resulting 274 500 data points (183 days per fire season x 30 years of climate period x 50 ensembles) for the present climate period from the data's empirical distribution. The future time periods (stepwise 30-year windows) were pooled over each member separately, resulting in (183 days x 30 years) 5490 data points from which we mapped the current-climate quantiles corresponding to different return periods to future return periods. Therefore, our sample size is large enough to extract the 99$^{th}$ quantile. We carefully edited the paragraph and described the derivation of return periods according to our procedure (see updated 2.5.3. section in the updated manuscript).

*New: For the second analysis, we calculated changes in the return periods of FWI quantiles that correspond to return periods of 10-, 20-, 50- and 100-years under current climate conditions (period 1980–2009) for the four subregions. To do so, we pooled daily FWI values over the entire 50-member ensemble (183 days per fire season x 30 year climate period x 50 members). Using this data pool, we determined the non-exceedance probability p of each FWI value in the present climate period using its rank r and the total sample size n following p = r/n. We derived FWI quantiles in the current climate period for non-exceedance probabilities of p = [0.9, 0.95, 0.98, 0.99] and the corresponding FWI return periods T of 10-, 20-, 50-, and 100 years using T = µ / (1 − p), where µ is the inter-arrival time (1/183 days in a fire season) (Coles, 2001). To analyze changes in return periods over time (from 1980 to 2099), we created centred, rolling 30-year windows for each ensemble member (183 days per fire season x 30 year climate period) and derived the cumulative distribution of the time window using the rv_histogram.cdf function of the Scipy package in Python (Virtanen et al., 2020). We mapped the FWI quantiles representing the 10-, 20-, 50-, and 100 year return periods of the current period (1980 - 2009) to future return periods, by deriving their non-exceedance probability p in the cumulative distribution of the rolling window climate period (future). Next, we placed their future probability p into T = µ / (1 − p) (Coles, 2001) to determine the return period T of the present FWI quantile under future climate conditions. This approach allows us to show how the return period of e.g. the current 100-year FWI will change over time with climate change. Due to the centered window approach, the first full 30-year window is 1995 and the last full 30-year window is 2084. Therefore, we show results between 1995 and 2084.*

2.  Several references do not represent the original work reflecting your statements. Examples are line 46, 126, 157, 181 and 212. Please make sure the original references are used throughout, or in cases where this is not possible, add "e.g." before the references to avoid the reader to believe the reference is the original work.

    Thank you for pointing out the need to focus on original references. We adjusted:

    - Line 46: added "e. g." to Bakke et al. (2023)
    - Line 126: added "e. g." to Di Giuseppe et al. (2016) and Touma et al. (2021)
    - Line 157: We modified the reference of Vitolo et al. (2019) to a direct quote:
        o Old: (Vitolo et al. 2009)
        o New: as suggested by Vitolo et al. (2009)
    - Line 181: We exchanged Böhnisch et al. (2021), with Hawkins (2018), because the fire rings are inspired from the warming stripes.
    - Line 212: We dropped Brunner et al. (2021).

We further adjusted all sentences, which were also highlighted in your special comments. For those changes, please see the corresponding replies further below and refer to the new manuscript.

3. The discussion comprises multiple detailed comments on different aspects of the analysis or results, with too general subtitles. Introduced topics (e. g., uncertainties related to the chosen climate model on line 302) and summaries of results are not always followed up. Overall, the current state of the discussion chapter makes it hard for the reader to know what to expect and to follow the line of arguments of the authors. Please revise and clarify the discussion chapter, avoid mentioning topics without commenting on them in relation to your study, and lift part of the discussion to a more general level.

Thank you for paying attention to the consistency of our discussion section. We agree that certain subtitles do not match the provided discussion points. We revised the discussion section carefully, changed subtitles, critically reflected on the chosen climate model setup and dropped paragraphs which were unrelated to the results. We reflected the ordering of our arguments carefully and adjusted the discussion section accordingly. RC-3 commented, that uncertainties from the chosen climate model (structural uncertainty) and bias correction should be discussed, which we also added to the discussion (see responses to RC-3). Changes in the other sections of the discussion referring to specific paragraphs of the manuscript have been revised in the corresponding comments (RC-1 comments 66. to 86. and RC-3 Discussion). We made very strong changes to this chapter and would like to point the reviewer to the new version of the manuscript for details. .

*New subtitles:*
*4.1. Spatio-temporal trends and variability*
*4.2 Dataset specific uncertainties*
*4.3 Limitations of fire danger metrics*
*4.4 Increasing fire danger and implications*

4. The text is in several places informal with the use of unnecessary introductions (e. g. "another aspect which has to be discussed" on line 283 or "needs to be critically reflected upon" on line 318) or subjective words, inconsistent used of concepts, and imprecise descriptions of methods and results. Further, most of the manuscript is written in present tense. Papers are usually written in past tense when presenting analysis and results in abstract, data and methods, results and conclusions. Thus, I would recommend changing to past tense. In general, please carefully revise the whole manuscript and clean and clarify the text.

Thank you for raising stylistic concerns regarding the writing style of the paper. We eliminated subjective words to our best knowledge and iterated over terminologies to ensure consistency. Further, we tried to avoid filling words as much as possible but kept them in places where they are necessary for a fluent reading flow. We changed to past tense in the introduction and data and methods sections. However, we kept the results section and parts of the discussion section in present tense because our results are here presented for the first time.

5. The manuscript refers to similar analysis in France and UK, but does not refer to results over the same region (HydBav) by others, e.g. from global or regional studies that cover the region. Please include other studies that cover your region (for example https://doi.org/10.1029/2018GL080959 and https://doi.org/10.1007/s10584-016-1661-x). In addition, please comment on potential differences between France, UK and your region, when you refer to results over these regions.

Thank you for highlighting these global and regional studies. We incorporated comparisons from regional studies (de Rigo et al. 2017 and Carnicer et al. 2022) over Europe to better reflect on the spatial differences between regions in Central Europe and how they differ from southern Europe, e. g. the Mediterranean.

*Specific comments:*

1. You state in the title that your study area represents "heterogeneous landscapes". However, in line 110-111 you "assume that the water availability, climatology and landscape of different river systems reflect the fire regime of an area". A more in-depth reflection on the heterogeneity of the regions, and even subregions of your study, in relation to your spatial aggregations and findings would be beneficial to better reflect the title of your study.

   Thank you for highlighting the need to clarify that with "heterogenous landscapes" we refer to heterogeneity across subregions and not within regions. A more detailed reflection of the differences among the subregions is given by the paragraph starting in Line 129 (of the updated manuscript) and underlined by Figure 1.

   *New: Since fire is closely related to the availability, or rather the absence of water (in terms of precipitation or soil moisture deficit), we assumed that the water availability, climatology and landscape characteristics of the four different complex landscapes resulted in subregion-specific fire regimes.*

2. «Central Europe» is a concept that refers to a considerably larger region than the study area "Hydrological Bavaria". Please clarify the region to avoid exaggerating your study domain, in particular in the Abstract (e.g., line 7 changing "over Central Europe" to "in a region in central Europe" or similar), the conclusion (e.g. line 395 changing "for Central Europe" to "in central Europe" or similar), and the title ("of Central Europe" to "in central Europe"). Specifically, change to lower case 'c' ("central Europe") throughout.

   Thank you for pointing out the need to be more specific about the region in central Europe we are focusing on. We revised the manuscript according to your suggestion to "a region in central Europe". We adopted central Europe with small c in central, because our study domain focuses primarily on the mid-latitudes of Europe, rather than the politically defined Central Europe (countries of Germany, Austria, Poland, Czech Republic, Slovakia, Switzerland, Hungary, Slovenia).

   We revised the term in:

   - Line 5 from "*over Central Europe*" to "*over a region in central Europe*"

   - Line 22ff. from "*Our results highlight central Europe's potential for severe fire events from a meteorological perspective and the need for fire management in the near future even in temperate regions*" to "*Our results highlight the potential for severe future fire events in central Europe, which is currently little fire-prone, and demonstrate the need for fire management even in regions with a temperate climate.*"

3. Titles: please use NHESS house rules (sentence-style capitalization: https://www.natural-hazards-and-earth-system-sciences.net/submission.html#manuscriptcomposition)

We checked the titles and subtitles and adjusted them according to the NHESS house rule "titles and headings follow sentence-style capitalization (i.e. first word and proper nouns only). This applies to table and figure headings as well."

We adjusted the heading *Increasing Frequency of Extreme Events* to *Increasing frequency of extreme events* and kept the style of lower-case titles throughout the manuscript.

4. Line 27 and line 396-397: You state that Central Europe has not been exposed to wildfires before recent years (line 27) or to date (line 396-397). However, central Europe has been exposed to multiple wildfires at least the past three decades. Rephrase to correct statements and provide reference(s) that have fire records underlying your statements.

We are aware that fires occurred also in the past three decades. Indeed, this sentence does not contain the message we want to communicate. Therefore, we changed the sentence:

*New: While the Mediterranean region and Western Canada have been historically fire prone and well studied on a larger regional scale (e.g. Abatzoglou et al., 2021; Barbero et al., 2020; Ruffault et al., 2020; Barbero et al., 2015), fire occurrence in the temperate climate regions of Europe has received less attention and is rather studied on a national than on a regional level (e.g. Bakke et al., 2023; Arnell et al., 2021; Fargeon et al., 2020).*

5. Line 38-40: The claim that fire indices represent a statistical correlation between fire events and meteorological conditions is wrong. Please correct.

We changed "*represent the statistical correlation between fire events and meteorological conditions*" to "*These indices are statistical models that were built on the correlation between fire events and meteorological conditions*".

6. Line 40-41: Please provide reference that "They have been proven to produce reliable ratings of fire danger in short- and long-term weather predictions on a global scale".

We added Di Giuseppe et al. (2016) as a reference to support this statement.

7. Line 42: please rephrase "do not guarantee", as this is an unclear statement. Fire indices have nothing to do with ignition at all.

Thank you for your remark. We rephrased *"do not guarantee"* to *"it does not imply an actual fire ignition"*.

8. Line 45 and other: Various concepts are used for the fire indices; please be consistent and potentially introduce relevant relations between concepts such as "fire risk", "fire weather", "fire danger", "fire indices", "likelihood of fire" and "probability of fire".

Thank you for pointing out these inconsistencies in terminology. We changed all occurrences of *"fire danger", "fire risk" and "probability of fire"* to "*fire danger*". "Fire weather" and "fire indices" are kept since they are not directly related to fire danger and are terms on their own. "Likelihood of fire" is not used in our manuscript. It is used as "likelihood of fire events" and cannot be replaced with "fire danger".

9. Line 49, 57 and 59: Central Europe (line 49) and temperate climate (line 58 and 59) refer only to studies of England and France here. As these concepts are used for the HydBav later, please

clarify the use and links between geographical regions and climate regions. Are results for England and France directly transferrable to HydBav?

We appreciate your comment regarding the climate comparability of our study region with other studies. HydBav, France and England all belong to the same climate zone according to the Köppen and Geiger classification' (Rubel et al. 2017 & Rubel et al. 2010). Therefore, we think that results for England, France, and HydBav can be compared and that the three regions can be jointly studied' under the umbrella of 'Central Europe and temperate climate zone.

We changed Line 49 from "*In Central Europe, trends related to fire danger are uncertain and not clearly distinguishable from natural variability*" to "*In contrast to the Mediterranean, temperate climate regions, such as central and western Europe, show uncertain trends in fire danger because these trends are not clearly distinguishable from internal variability when multi-model climate ensembles are used (Arnell et al., 2021; Fargeon et al.,2020)*" to clarify the statement.

10. Line 49-50: Please provide reference that trends are not distinguishable from natural variability in Central Europe. Further, this sentence relates to the weak trend signal relative to natural variability independent of how models represent the natural variability, and thus the link to the next sentence is wrong or not clear.

    We refer to the weak trend signal relative to natural variability. However, in multi-model studies, uncertainty is composed of two factors: (1) natural variability and (2) model uncertainty. Our linkage to the next sentences highlights the need for using modelling approaches representing natural variability, i. e. Single Model Initial-Condition Large Ensembles, as used in our study. We rephrased the sentence to clarify that these statements originate from Frageon et al. (2020) and Arnell et al. (2021).

    We clarified our statement:

    **New:** *This uncertainty originates from the confusion of internal variability with structural uncertainty related to the different climate models in the ensemble (Arnell et al., 2021; Fargeon et al., 2020). Separating the forced signal in FWI changes from internal variability using multi-model ensembles only is challenging, in particular in temperate climate regions with a low signal to noise ratio (Arnell et al., 2021; Fargeon et al., 2020; De Rigo et al., 2017).*

11. Line 52-53: Unclear sentence "In France, …" Please rephrase.

    While revising the manuscript, we came to the conclusion that this sentence did not add any significant additional value to our introduction and removed it.

12. Line 56: clarify the meaning of "natural variability of changes".

    This sentence was removed as well.

13. Line 59-63: The claim here is that climate model ensembles using multiple models (but fewer simulations per model compared to SMILE) underrepresent natural variability, whereas SMILE does not. Please clarify how large ensembles using a single model (SMILE) better represent natural variability as compared to large ensembles from different models, and add references that support this claim.

We agree that the term underrepresent is wrong in this context. Multi-Model ensembles mix natural variability with model uncertainty. For this reason, we chose a SMILE framework in our study that allows for a clear isolation of climate change signals from natural variability. We changed the sentences in Lines 59ff.:

*New: This challenge can be addressed by evaluating climate model simulations derived from a single model initial-condition large ensemble (SMILE) which enables a clear isolation of the forced climate change signal from internal variability (Deser et al., 2012). SMILEs represent an ensemble of simulations derived using one single climate model started at different initial conditions. The ensemble spread between the different SMILE members provides a robust estimate of the internal variability, from which the forced response of the climate change scenario can be estimated by averaging over the SMILE members for a specific variable, e. g. temperature (Deser et al., 2020). While single SMILEs allow for the quantification of internal variability, they do not enable a quantification of model uncertainty (Deser et al., 2020). Most of the available SMILEs rely on global circulation or earth system models with a coarse spatial resolution and are unsuitable to assess changes in fire weather over regions with complex terrain such as central Europe including the Alps.*

14. Line 70: The reference period (1980-2009) is not one of the established reference periods. Please explain the choice of the period. Why not use the almost identical period 1981-2010, which is a widely used reference period?

    We chose this time period because the CRCM5-LE model runs until the year 2099. Therefore, the future time period can only be set to the maximum year of 2099, i.e. 2070-2099, which is why we set the present time period to 1980 to 2009. We argue that using 1980 – 2009 is as good as using 1981 – 2010 and does not lead to wrong assumptions in our analysis.

15. Line 71: Please change "increases" to "changes", because your analyses were also able to detect if there were any decreases.

    You are right, we adjusted *"increases"* to *"changes"*.

16. Line 74: Please clarify which TOE you refer to (TOE of what?)

    We changed "*the time of emergence (TOE) is reached latest by 2099*" to "*when does the FWI reach its time of emergence (TOE)?*".

17. Line 82: Please clarify whether your mean two domains in Europe, or two domains of which one is in Europe.

    We adjusted from "*over the domains in Europe and Northeast North America*" to "*over two domains (i. e., Europe and Northeast America).*".

18. Line 88: clarify "independent" (in which regards?). Fifty members based on the same model are far from independent as such.

    Thank you for bringing up this point, which was also made by RC-3 (general comment). Our initial data set description did not clearly explain the setup of our dataset. For clarification, we revised the text as follows:

*New: These random perturbations were introduced by parameterizing a single aspect of the model's cloud properties using a different pre-set seed for each of the 50 simulations. This ensured that the climate change realizations were different from each other without changing the model dynamics, physics, or structure (Fyfe et al., 2017). After a 5-year spin-up phase, the modeled climate of the initialized 50 members in the CRCM5-LE were considered independent (Leduc et al., 2019), because the chaotic climate properties caused diverging climate trajectories solely based on the macro- and micro-initialization of the CanESM2-LE (Wood, 2023).*

19. Line 91: please clarify "at this time"

   We dropped the sentence while editing our manuscript.

20. Line 93: please clarify the link between your study choice and the provided references.

   We changed the sentence from "*In this study, we interpret internal variability as natural variability (Böhnisch et al., 2021; Von Trentini et al., 2019; Kay et al., 2015)*" to "*Therefore, the differences among the 50 CRCM5-LE members can be interpreted as natural variability (Böhnisch et al., 2021; Wood, 2023; Mittermeier et al., 2019; Leduc et al., 2019), and are referred to as internal variability throughout this paper (Hawkins and Sutton, 2009).*"

21. Line 93: Please comment on this assumption (internal variability = natural variability) in the discussion or here. Potential limitations or lack thereof? Comment why this assumption is correct.

   We agree that internal variability does not equal natural variability in every case. We decided to stick to the term internal variability throughout the manuscript, because we used the term natural variability to describe model internal variability. We clarified this section and changed the term "natural variability" to "internal variability". Further, we added a paragraph to the discussion (Chapter 4.2. Lines 317ff.) to reflect on this aspect more critically.

22. Line 95: How does smaller (temperature) and equal (precipitation) member spread in your SMILE compared to EURO-CORDEX relate to the previous claim that SMILE overcome the limitation of multi-model ensembles related to under-representation of climate variability. The results of Von Trentini (2019) imply that multi-model ensembles represent a larger variability as compared to CRCM5-LE.

   Thank you for this comment. Ensemble spread in a SMILE is originating solely from internal variability, while in multi-model ensembles, ensemble spread also includes structural variability (model uncertainty (see Hawkins and Sutton 2009)). In cases where structural uncertainty is larger than internal variability, a multi-model ensemble such as EURO-CORDEX that also includes structural uncertainty would show larger variability between members than a SMILE that does represent internal variability only. However, the spread of the multi-model ensemble in such a case would come from structural uncertainty rather than from climate variability. If we want to achieve a good representation of climate variability rather than structural uncertainty, SMILEs are needed.

23. Line 98: state which observational data were used for the bias correction.

   Thank you for pointing out this very important missing information. We added a more specific description of the observation data used for bias correction:

*New: Further, the CRCM5-LE was bias corrected using the univariate quantile mapping approach of Mpelasoka and Chiew (2009) (Poschlod et al., 2020) for all the FWI input variables. {…} For the bias correction, the meteorological Sub-Daily Climatological REFerence dataset (SDCLIREF), which combines hourly and disaggregated daily station data (Brunner et al., 2021), served as an observation reference.*

24. Line 100: please rephrase and clarify. Better represented when evaluating against what (wouldn't that be against climate data, which you state is what should be bias-adjusted in the first place)? "climate data" is very general, do you mean data from climate models?

We clarified the sentence as follows:

*New: Bias-corrected data have been commonly used for projections of fire weather indicators such as the FWI (e. g. Yang et al., 2015; Cannon, 2018; Kirchmeier-Young et al., 2017; Ruffault et al., 2017; Fargeon et al., 2020), as they have been shown to reflect fire danger more accurately than raw climate data when compared to observational data (Yang et al., 2015).*

25. Line 100: Has there been any studies evaluating the data you use against meteorological variables from observations (independent of the bias adjustment) or reanalysis over the region?

Thank you for this valuable remark. Yes, a comparison between observation data and bias corrected and reanalysis data is provided in the supplementary material of Poschlod et al. (2020). Their results show that the biases for temperature are positive and highly variable for the Alps, while they are negative for the other parts of the study area in the summer months June, July and August. For precipitation, the bias correction affects the pre-alpine regions, where the non-bias corrected data show differences > 200 mm from the bias corrected data.

26. Line 103: Please insert the stated rivers in Figure 1 in order to inform a reader, who is not familiar in the region, how the named rivers relate to the study area.

We updated the figure according to your suggestions by adding the main rivers and their names.

[Figure]

*Figure 1 - Subregions of Hydrological Bavaria by landscapes and land cover (modified, CLMS (2021).*

27. Line 108: Does 's' refer to 'see'? Please write out. Also, use capital F in figure names.

We adjusted to capital F in figure names and changed "s." to "see".

28. Line 110: 'water' and 'water availability' are imprecise. Clarify what water you mean and add a supporting reference. If you mean soil moisture or precipitation, these (and thus also the fire regimes) are likely highly heterogeneous within each subregion (in particular in mountainous areas).

Thank you for pointing out the need for clarification. We changed the sentence to:

*New*: *Since fire is closely related to the availability, or rather the absence of water (in terms of precipitation or soil moisture deficit), we assumed that the water availability, climatology and landscape characteristics of the four different complex landscapes resulted in subregion-specific fire regimes.*

29. Line 109-111: Please clarify and justify your assumption. The subregions are not defined according to the river systems, i.e. river catchments (as seen from Fig. 1). As you state earlier (line 102 and title) and later (line 120-123), hydrology, climatology and landscape are highly variable in the study area, and is likely highly variable in particular in the mountainous areas, within a subregion, with consequences for fire characteristics. What is mean by "an area"?

We adjusted the unclear terminology regarding river systems (to landscapes) and "an area" (to selected subregions) as described in the previous comment (28.). The subdivision of the study region into four subregions aims to address the trade-off between the number of regions and the amount of inter-subregion variability. While a further subdivision into even more subregions would further increase within subregion homogeneity, it would also make it more difficult to summarize findings. The four regions chosen for the analysis are sufficiently similar in terms of their climate in order to allow for a succinct spatial summary.

30. Line 112-120: Which period and data underlie the numbers presented here? Could a figure (temperature and precipitation spatial patterns) be added (e.g. in appendix) to make the information more intuitive to the reader?

The numbers presented in this sentence are supported by plots of mean temperature and precipitation in Willkofer et al. (2020), which we listed as a reference.

31. Line 129: As written, 'noon' refers only to wind. Rephrase so it refers to all variables (even 24h precipitation is measured at noon).

We changed: *(temperature, relative humidity, wind speed at noon and 24-h accumulated precipitation)* to *(temperature, relative humidity, and wind speed - all at noon - and 24-h accumulated precipitation).*

32. Line 132: what is meant by 'bookkeeping'? This concept is linked to financial transactions. Can it be replaced by a more commonly used concept within natural sciences to be more intuitive to the reader?

We adjusted: "*The first three sub-indices represent the fuel moisture codes and can be understood as bookkeeping systems, which increase moisture after rain and reduce moisture for*

*each day of drying"* to *"the first three sub-indices represent the fuel moisture codes and contain information about antecedent conditions"*.

33. Figure 2 caption: suggest replacing 'vegetation' with 'organic matter', 'fuel layers' or similar (as used in the main text) for clarity.

   We replaced *"vegetation"* with *"organic matter"*.

34. Line 143: please rephrase or clarify "without memory of past conditions". Because the fuel moisture codes have memory of past conditions, BUI and ISI have too.

   We changed *they are stateless and without memory of past conditions* to *they are stateless and only indirectly linked to past conditions.*

35. Line 156: High-altitude parts of the region will likely have snow in the beginning of the defined fire season. Please state if/how you have accounted for snow in the evaluation here. E.g. see last section in https://www.nwcg.gov/publications/pms437/cffdrs/fire-weather-index-system. If you are neglecting the effect of snow on fire danger, it is worth reflecting on it in the discussion.

   Thank you for your comment. Indeed, we do not explicitly take snow cover into account. As you suggest, we added this specification to the discussion section (seeRC-1 comment 81. and RC-3 comment on Line 359).

36. Line 157: Vitolo et al (2019) does not apply any fire season and should be replaced by reference(s) using or arguing for using April-September.

   We derived the definition of our fire season from Vitolo et al. (2019) who stated: "By convention, the dry season in the northern hemisphere is assumed to start on 1st April and ends on 30 September, while in the southern hemisphere it starts on 1st October and ends on 31st March." Therefore, we believe that the reference is appropriate to justify the choice of the fire season definition.

   *New: The generated dataset was later cropped to the dry season (April 1st to September 30th) of the northern hemisphere, which was used as the fire season in our study as suggested by Vitolo et al. (2019).*

37. Line 158: "annually calculated FWI values" is unclear (may refer to annual values, which I assume is not the case). Suggest to delete "of annually calculated FWI values" for clarity.

   We implemented your suggestion and deleted "of annually calculated FWI values". Further, we changed *"We calculated daily FWIs for each year (January to December) and climate model ensemble member between 1980 and 2099 using the CFFDRS R package (Wang et al., 2017)"* to *"We calculated daily FWIs on a daily basis for each full year (January 1st to December 31st) and climate model ensemble member between 1980 and 2099 using the CFFDRS R package (Wang et al., 2017)."* for clarification.

38. Figure 3: Please add a legend (ref. NHESS figure composition: https://www.natural-hazards-and-earth-system-sciences.net/submission.html#manuscriptcomposition).

   We added a legend to Figure 3.

[Figure]

*Figure 3 - Median FWI for the CRCM5-LE mean (thick blue line) and standard deviation (light blue shading) in comparison to the reference dataset of Vitolo et al. (2020) marked pink (X for values, lines for deviation from the CRCM5-LE mean). Top and bottom blue lines mark the 25th and 75th percentile of the CRCM5-LE.*

39. Line 170: As you state yourself, the results of the evaluation you have performed does not affect the climate change impact assessment of your study. Why do you not evaluate your data using measures that can actually reflect your data's ability to assess climate change impacts? For example its ability to represent historical changes.

Thank you for this remark. We agree that this sentence is misleading in its message. On a temporal scale (see Figure 3), we show that the CRCM5-LE captures observation values within a tolerated range between the 25th and 75th percentile well.

We clarified this sentence also in correspondence with RC-3 (comment on Line 170) and added a remark to this paragraph, that the ensemble overestimates interannual variability:

***New:*** *In the northern and especially northwestern parts (i.e. Southgerman Escarpment) of the study area, the CRCM5-LE overestimates FWI values in comparison to the reference dataset by an order of two to four.*

40. Figure 4: The applied FWI colour scale is almost identical to the FWI colour scale provided in Table 1, but they reflect different FWI intervals. Please change for clarity.

We adjusted the color scale to the colors in Table 1 and updated the figure.

[Figure]

*Figure 4 - Median FWI of (a) the CRCM5-LE, (b) reference dataset of Vitolo et al. (2020) and (c) difference (CRCM5-LE – reference dataset) for the present time period (1980–2009). The dataset difference is calculated from resampling (a) to the spatial resolution of (b) using a nearest neighbour approach.*

41. Section 2.5.1: The description is unclear in terms of when the different aggregations were applied (both in space and time), and when the continuous analysis vs the data split are applied. Consider reorganise the section to better fit each part of the analysis. Please also clarify how the 'extreme condition' (90th percentile) is computed (is it over the analysed time period, region or models, and in which order is it calculated). Clarifications in this section are necessary for reproducibility.

Thank you for pointing out the need for clarification. We did not perform a trend analysis but compared two time periods and therefore changed the subsection title from "*Trends*" to "*Changes in fire danger*". Further, we clarified the description of the pooling procedure, e. g. how we derived the median and extreme percentiles:

*New: We evaluated changes of fire danger derived from the CRCM5-LE over the time period 1980 to 2099 in the study area with statistical metrics: Median and extreme conditions were examined using the 50th and 90th quantiles of the FWI, respectively. The quantiles were calculated for different aggregation levels, either temporally on a monthly scale or spatially for the previously defined subregions. We derived the median and extreme for each ensemble member separately. Changes of fire danger were either compared between two climate periods or analyzed continuously from 1980 to 2099. For the climate period comparison, the dataset was split into two 30-year periods: 1980–2009 and 2070–2099 representing current and future climate conditions, respectively. For both periods, we derived the median and extreme quantiles for each fire season month for each of the 50 members of the CRCM5-LE.*

42. Line 174: Please provide which trend method was applied.

We did not perform a trend analysis, instead, we compared FWIs across two time periods (1980-2009 and 2070 - 2099). Therefore, we rephrased "trend" to "changes".

43. Line 177: You state that you are "summarizing FWI values over a fire season on daily, monthly or annual basis". Summarizing would provide very different ranges of FWI on the different temporal

scales, and it does not look like they are summarized in e. g. Fig. 5 (looks like average or median over each month). Please correct or clarify.

Thank you for indicating this misleading description. We clarified the sentence by changing from: "*The percentiles are calculated for different aggregation levels, either temporally, summarizing FWI values of a fire season on daily, monthly or annual basis, or spatially for the previously defined subregions*." To: "*The percentiles are calculated for different aggregation levels, either temporally on a monthly scale or spatially for the previously defined subregions*".

44. Line 178 and line 182: Suggest replacing 'increasing' with 'changes in' for clarity. The analyses allow for changes in both directions.

    We changed "*increasing"* to "*changing"* in the suggested lines.

45. Line 216: Your results are scenario specific. Please specify the scenario, e. g. "according to RCP 8.5".

    The sentence starts now with "*under the RCP8.5 emission scenario…*".

46. Line 219-222: Please rephrase to clarify your reasoning.

    We agree that the original distinction between weak (one-level) and strong (two-level) fire danger level increases is imprecise and subjective. We rephrased "*weak*" to "*one level*" and "*strong*" to "*two levels*". Further, we added a sentence, which explains why we also look at two-level increases. This is mainly because one level increases are found throughout the entire study area and we want to provide additional information by distinguishing between one level and two level rises in fire danger.

    *New: Significant increases in fire danger levels (thin dots in Figure 5) first occur in June and remain present throughout the study area until September for both the median and extreme FWI. Highlighting grid cells, which experience a rise of at least two levels, helps us to identify regional hotspots of future increases in fire danger. We find increases in fire danger of at least two levels for the Southgerman Escarpment in July and August for the median FWI and in August for the extreme FWI.*

47. Line 223: Why is not the Southgerman Escarpment mentioned here (regions of strongest rises in extreme FWI in July to Sep)?

    We clarified this sentence, which refers to changes of at least two fire danger levels.

    *New: We find increases in fire danger of at least two levels for the Southgerman Escarpment in July and August for the median FWI and in August for the extreme FWI. The other subregions (Alps, Alpine Foreland, Eastern Mountain Ranges) are affected by a two-level rise in danger-level of the extreme FWI in August. Additionally, the western parts of the Southgerman Escarpment and parts of the Eastern Mountain Ranges are affected by a two-level danger-level increase in September for the extreme FWI (see Figure 5).*

48. Line 226, 228 and 229 and potentially other places: 'median case' and 'extreme case' are unclear concepts. Do you refer to 'median FWI' and 'extreme FWI'? Please be consistent with concepts or clarify newly introduced ones.

Thank you for highlighting this. We refer to the median FWI (50th percentile) and extreme FWI (90th percentile) of our results. We adjusted the sections accordingly.

*New: The median FWI points out that high fire danger becomes the average condition in the Alpine Foreland by 2080, in the Southgerman Escarpment by 2060 and in the Eastern Mountain Ranges by 2070 (see Figure 6 [1]). The Alps are exposed to high fire danger only when looking at the extreme FWI (see Figure 6 [2]) from 2070 onwards. The other subregions are more strongly affected by changes in the extreme FWI than the Alps.*

49. Line 233: Why do you state 'mean conditions (median)' and not 'median conditions' (what is the difference)? Please clarify in the text. Similarly, clarify similar statement in line 349: 'On average (median)'

Thank you for pointing this out. We are looking at median conditions throughout the study and clarified this by replacing "mean" or "average" conditions by "median" conditions. Further, we edited Section 2.5.1. to clarify that we derive the median and extreme percentile on a member basis and use the average / mean to derive the ensemble mean for the median and extreme quantiles, when we speak about ensemble mean.

50. Line 236: Please consider using the phrasing 'mid 21st century' instead of 'middle of the 21st century'.

Thank you for pointing this out. We replaced "*middle of the 21st century*" by "*mid-21st century*".

51. Line 240-241: Please clarify how your findings indicate that the distribution of the FWI extremes resembles the distribution of the FWI median? Figure 7 clearly shows that the distributions are different both in terms of mean and standard deviation.

We removed this statement from the text.

52. Line 242: consider replacing 'changes' to 'increases' for clarification.

We clarified the sentence by following your suggestion.

*New: FWI increases in the Alps are weaker than in the other subregions.*

53. Line 244: please specify what you mean by 'strongly'.

We exchanged *"strongly"* with *"continuously"*.

54. Line 244-245: Please clarify what you mean and which parts of the results you refer to. Your statement here seem opposite compare to the preceding sentence (median FWI increase strongly vs median FWI shows hardly any changes).

We clarified that this section solely refers to the results for the Alps.

*New: Throughout the 21st century, the median and extreme FWI will increase continuously in the Alps. While the extreme FWI is projected to shift from low to moderate fire danger in this subregion, the median FWI shows hardly any changes and remains in the no danger level zone even at the end of the century (see Table 1).*

55. Line 245: Why state the EFFIS reference here, when every classification of fire danger level in the (also previously mentioned) results is based on it?

    You are correct, this is redundant information. We changed the reference to Table 1 (see applied changes in comment 54.)

56. Line 247: clarify which average you are referring to.

    We clarified the sentence from "*In the extreme case, the average fire danger is moderate*" to *"for the extreme FWI, the ensemble mean fire danger is moderate"*.

57. Line 247-248: the interval signs in parenthesis are wrong.

    Thank you for finding this tiny but very relevant mistake – we corrected it.

    **Old:** *(11.2 < moderate > 21.3) & (21.3 < high > 38)*

    ***New:*** *(11.2 < moderate < 21.3) & (21.3 < high < 38)*

58. Figure 5: The levels referred to in the result section would be more easily recognisable if a colour scale using discrete colours was used. Discrete colours would provide a more clear message to the reader, in particular when these levels are the main message of these results, and not the small varieties in between. Please consider changing to discrete colours.

    We agree that a discrete color scale is more appropriate here than a continuous color scale. Therefore, we, adjusted the plot accordingly.

[Figure]

*Figure 5 - Ensemble mean of the median ([1], 50th quantile) and extreme FWI ([2], 90th quantile) by fire season month (April (a) – September (f)) for the future time period 2070–2099. Dots indicate that 90% of the CRCM5-LE members agree on a fire danger level increase of at least one level (thin black dots) or at least two (thick black dots) levels compared to the present period (1980–2009).*

59. Figure 5 caption: do you mean "at least" two levels (indicated by thick black dot), or are there never more than two levels?

We clarified this to at "at least one/two levels".

***New:*** *Ensemble mean of the median ([1], 50th quantile) and extreme FWI ([2], 90th quantile) by fire season month (April (a) – September (f)) for the future time period 2070–2099. Dots indicate*

*that 90% of the CRCM5-LE members agree on a fire danger level increase of at least one level (thin black dots) or at least two (thick black dots) levels compared to the present period (1980–2009).*

60. Line 259-260: please provide numbers or proportions in parenthesis.

We provided the percentage of days per fire season in parenthesis in the text (see lines 273ff.) and Figure (see RC-1 comment 61).

*New: In the future (2070–2099), the percentage of days with fire danger (>= low) shifts from 10% to 33% in the Alps, from 25% to 50% in the Alpine Foreland and Southgerman Escarpment, and from 33% to 60% in the Eastern Mountain Ranges (see Figure 8). In the Alps, no and low fire danger days currently account for 182 out of 183 days (99 %) of the fire season.{…}*

61. Line 262: Please provide in what ways they are similar. EMR is not described in other terms than relative to Alpine Foreland.

We clarified the section.

*New: In the Eastern Mountain ranges, the frequency of low (20%) and moderate (15 %) fire danger days in the future is similar to the Southgerman Escarpment and the Alpine Foreland.*

62. Figure 8: Please consider adding proportions on the right y-axis, as proportions are used in the text.

We added proportions to the figure.

[Figure]

*Figure 8 - Number of days of fire danger levels in the fire season (April – September, 183 days) for the present (1; 1980–2009) and future (2; 2070–2099) climate period. FWI danger classes are derived for the subregions (a) Alps, (b) Alpine Foreland, (c) Southgerman Escarpment and (d) Eastern Mountain Ranges*

63. Figure 8 caption: Please clarify by specifying what is meant by frequency (e.g. "number of days within a fire season")

We replaced *"frequency"* by the *"number of days"*. For the implementation see the Figure caption in the previous comment (RC-1 comment 62).

64. Line 278: As in line 215, clarify the scenario dependence of your results (in line with your statement in line 306-307). The way it is phrased now imply more certainty about the future than we can state.

We added "*when a RCP8.5 scenario is assumed*" to the sentence to emphasize that our results soley refer to the RCP8.5 scenario.

*New: Our results demonstrate that fire danger in central Europe increases strongly until the end of the 21st century, if the RCP8.5 scenario is assumed.*

65. Line 279: Why move away from the defined classes? How is hazardous defined?

Thank you for recognizing this stylistic inconsistency. We rephrased the sentence:

*New: The future increase of the number of days with conditions favoring high or higher levels of fire danger emerges for all metrics assessed in this study, i.e. different quantiles and aggregation levels of the ensemble and in space and time.*

66. Line 285-286: please state the variable (FWI) that is compared.

This sentence has been removed during the editing process.

67. Line 289-291: Please state in what relevant ways the formulas have been adjusted (i.e. relevant implications). Is this a more likely reason for the differences as compared to the fundamental differences in how the underlying meteorological data are produced?

Thank you for this valuable remark. This statement originates from Vitolo et al (2019). Vitolo et al. (2019) state in their algorithm validation section "Although the outputs are rather close, they do not match exactly. The reason is that the ECMWF model follows the formulation defined in the reference FWI implementation outlined in Van Wagner (1987) without modifications. Wang et al. (2017) instead, have modified some of the original equations (i.e. EQs 12 and 15) leading to the calculation of DMC and DC. As a consequence, FWI and DSR also slightly differ."

However, Wang et al. (2017) do not explain where and how they adjusted the original formulas. We therefore keep this sentence as it is.

We moved this paragraph to chapter 4.3 about limitations of fire danger metrics.

*New: Our validation set-up demonstrates that the algorithm used to compute the FWI generates comparable results to the reference dataset (see Figure 3) even though our analysis used the CFFDRS R package to calculate the FWI (Wang et al., 2017), whereas the reference dataset was generated with the Global ECMWF Fire Forecast (GEFF) model (Di Giuseppe et al., 2016). These models differ slightly in their results because the GEFF model applies the original FWI formulas from van Wagner and Pickett (1985) and the CFFDRS R-package uses adjusted formulas for DC and DMC (Wang et al., 2017).*

68. Line 293: please rephrase sentence to be more to the point. It is unclear how the tiling patterns referred to in the text 'has to be discussed' and not the ones seen e. g. in September (Fig. 5 [2]f) or at smaller scale in the Alps in July-Sep (Fig 5 [1]def and [2]def).

Thank you for your comment. We agree that this sentence is not clearly brought to the point. We rephrased the whole section to clarify our intentions with mentioning the tiling pattern, which is visible in Figure 5, even after adjusting to a discrete color scale. We mention the tiling pattern now in chapter 4.2 about dataset specific uncertainties. We added a Figure illustrating bedrock depth in the CanESM2 to the Appendix (Figure A4).

*New: The CRCM5 represents FWI at much finer spatial resolution than CanESM2 and therefore adds robust high-resolution features (Böhnisch et al., 2020). However, we find tiling patterns on the border between the Southgerman Escarpment and the Alpine Foreland (see Figure 5), which correspond to the geophysical baseline parameterization of the CanESM2 (see Figure A4). In comparison to the CORDEX multi-model ensemble, the CRCM5-LE shows drier and warmer climate change signals for temperature and precipitation (von Trentini et al., 2019). These characteristics of the CRCM5-LE are in line with the results from the validation (see Figure 3) and suggest that our results represent an upper limit of the expected changes in future fire danger.*

[Figure]

*Figure A4 – Bedrock depth in the CanESM2 and boundaries of Hydrological Bavaria (black).*

69. Line 296: please change 'correlates' with a more appropriate word or provide correlation results.

We replaced *'correlate'* by *'correspond'*.

70. Line 302: you mention the uncertainty related to the chosen climate model. Please elaborate on this point in relation to the specific model you applied.

Thank you for highlighting the missing discussion of the performance of the CRCM5-LE in comparison with other climate models. We edited this section carefully by adding a sentence that explains the difference between the CRCM5-LE and other CORDEX models in terms of precipitation and temperature.

*New: In comparison to the CORDEX multi-model ensemble, the CRCM5-LE shows drier and warmer climate change signals for temperature and precipitation (von Trentini et al., 2019). These characteristics of the CRCM5-LE are in line with the results from the validation (see Figure 3) and suggest that our results represent an upper limit of the expected changes in future fire danger.*

71. Line 316: please remove "potential of". FWI describes the fire weather, not the potential of fire weather.

    We removed "*potential of*".

72. Line 318-327: Please consider deleting this paragraph, and alternatively reduce the main message to a single sentence in the methods chapter arguing for your use of danger levels.

    We deleted the paragraph and explained the reason for using fire danger levels in the methods section.

    New sentence in chapter 2.4 (Methods): *To facilitate the interpretation of the FWI, we used the seven fire danger classes proposed by the European Forest Fire Information System (EFFIS; EFFIS, 2021) and assigned the FWI to particular fire danger levels. These FWI danger levels and their corresponding color scheme are shown in Table 1.*

73. Line 328-331: Please elaborate briefly on the flammability of the surface in your study region.

    We added a land use map to the appendix and described the flammability in the study region and subregions briefly in the second paragraph of chapter 4.2. We also discussed your previous comment on snow cover here.

    *New: While the FWI addresses fire danger in a meteorological context, it does not account for the flammability of the surface. Land-use in our study area is complex, but contiguous forests are present in all four subregions, especially the Eastern Mountain Ranges and the Alps. Persistent snow cover in winter prevents fire occurrences in spring in the Alps (Conedera et al., 2018) and other regions of high elevation, even though fire weather conditions might be met. Large parts of the South German Escarpment and Alpine Foreland are used for agricultural purposes, where fires can spread fast under dry conditions (see A1). However, these regions are more densely populated than the other two regions (Eastern Mountain Ranges and the Alps), which enables a faster suppression of fire incidents. For large-scale FWI analyses, non-burnable areas such as deserts and bare soil were masked out (Touma et al., 2021; Vitolo et al., 2020). In the context of the study area HydBav and the 11-km resolution of the CRCM5-LE, land use was highly variable on a sub-pixel scale and non-burnable areas (e. g. lakes, snow and ice covered areas and urban areas) were therefore not masked out (see Figure A1).*

74. Line 333-339: Please reflect/explain results rather than summarise them.

    We shortened this section and emphasized the differences between mountainous and non-mountainous terrain. Further, we moved this section to the top of the discussion section.

*New: Our results demonstrate that fire danger in Central Europe increases strongly until the end of the 21st century, if the RCP8.5 scenario is assumed. The future increase of the number of days with conditions favoring high or higher levels of fire danger emerges for all metrics assessed in this study, i.e. different quantiles and aggregation levels of the ensemble and in space and time. Within the ensemble spread, increases in fire danger extremes (90th quantile) are more pronounced than increases in median (50th quantile) conditions according to all assessed metrics (see Figures 5, 6 and 8). In space, we find that the variability of the FWI increases more strongly in mountain regions than in non-mountain regions, which is demonstrated by smaller changes in the median FWI than in the extreme FWI in the Alps and smaller differences in the increases in median and extreme fire danger for less complex terrain (Alpine Foreland and Southgerman Escarpment) (see Figure 5).{…} In time, extreme fire weather (90th quantile) is more likely to occur in the second half than in the first half of the fire season because the differences between the median and extreme FWI quantiles are smaller in April, May and June, than in July, August and September (see Figure 5).*

75. Line 340: increases in variability (line 337) is not the same as high variability in general (line 340). Please elaborate what you mean by your findings (increasing variability over time in mountainous regions) corroborate the findings by Wastl et al (2012; higher variability in mountainous regions than other regions).

We clarified this sentence in line with specific comment 74.

New: *In space, we find that the variability of the FWI increases more strongly in mountain regions than in non-mountain regions, which is demonstrated by smaller changes in the median FWI than in the extreme FWI in the Alps and smaller differences in the increases in median and extreme fire danger for less complex terrain (Alpine Foreland and Southgerman Escarpment) (see Figure 5). This corroborates findings by Wastl et al. (2012), who explained the higher fire danger variability in mountain regions by the higher terrain variability, i.e. rain-shadow effects and katabatic dry winds (Foehn).*

76. Line 345: Unclear whether 'extreme FWI conditions' represent the 90th percentile or the classes (FWI>50). In case of the former, do you mean elevated conditions compared to former months or compared future to present. In case of the latter, is that not seen directly from the figure and not 'implied' from your findings? Please clarify the meaning of this sentence.

Thank you for pointing this out. We agree that our phrasing here was imprecise and misleading and rephrased the sentence.

*New: In time, extreme fire weather (90th quantile) is more likely to occur in the second half than in the first half of the fire season because the differences between the median and extreme FWI quantiles are smaller in April, May and June, than in July, August and September (see Figure 6).*

77. Line 348: 'tremendous' is subjective, please clarify. See also 'dramatic' in line 362 and 'strikingly' in line 364-365.

We changed "*tremendous*" to "*substantial*", "*dramatic*" to "*as high fire danger levels as*" and "*strikingly*" to "*remarkable*".

78. Line 349: by 'seasonal', do you mean 'monthly'? In which ways are they hotspots, in terms of general conditions/increases/other?

We removed this sentence.

79. Line 358: The use of vegetation in Figure 2 caption implies also litter and organic matter on the ground. In this context, vegetation is necessary for fire development because it comprise the fuel. Is it the same use of vegetation here? I assume vegetation is highly present during winter also, although parts are covered by snow, and deciduous trees lack their green leaves. Please clarify the text.

We clarified this sentence. We do not refer to vegetation itself, but to the vegetation period. In the winter season, the vegetation is not actively growing, which leads to decreased fuel moisture. We rephrased this section (see Wastl et al. 2012 and Conedera et al. 2018).

*New: Over the course of the 21$^{st}$ century, the fire season will prolong, as fire danger levels are still elevated in September from 2030 onwards (see Figure 6). This suggests that the fire season might extend to at least October towards the end of the century. For the Southern Alps, Wastl et al. (2012) identified the main fire season between December and April because of low precipitation and decreased fuel moisture outside of the vegetation period (Conedera et al., 2018). Future studies assessing changes in fire danger and fire events in temperate climate regions should therefore consider the whole year instead of the vegetation season only.*

80. Line 359: 'half year' typically refers to six months. Consider changing to 'period' or similar, as you refer to December-April.

We exchanged *"winter half year"* to *"months from December to April"*.

81. Line 358-360: would FWI be suitable for the winter season? The reasoning provided here include lack of vegetation, whereas this is not accounted for in FWI. And what about snowfall and snow cover? Further, would you assume the temperature thresholds included in FWI calculation be exceeded in the Alps in winter? Please reflect on the considerations needed for such assessments.

We agree that the FWI is not suitable for the winter season and suggest considering using other approaches in cases where the winter season is explicitly considered, e.g., the one proposed by Pezzatti et al. (2020). However, our study only focuses on the months April to September, when snow cover in a 11 km grid scale plays a minor role for forest fire danger, because it occurs only in unvegetated high alpine terrain, which is sampled only by a small fraction of the 11 km grid.

82. Line 366: states 'exists currently no fire danger', however you have fire danger everywhere (as fire danger is defined as the estimates from the index, regardless of values). Please clarify.

We removed this sentence.

83. Line 372-373: you mention overestimation of natural variability. How does this relate to line 59? What about potential underestimation when using SMILE? If a model has a limitation (e.g. in representing natural variability), all realisations from that model suffer from the same limitation. If you or other have validated the ability of SMILE to represent natural variability, please state this in the text and refer to relevant evidence. Applies also for line 375.

Line 59 refers to multi-model ensembles, which do not overestimate natural variability but do not allow to distinguish between natural variability and model uncertainty. We revised this section also in accordance with specific comment 13 of RC-1 and comment 1 of RC-3:

*New: Reasons for this delay in TOE in France could be the later and shorter reference period (1995–2015) used by Fargeon et al. (2020), the larger uncertainty range originating from natural variability and model uncertainty in the multi-model ensemble as compared to the SMILE (Deser et al., 2012), the warmer and drier climate change signal of the CRCM5-LE (von Trentini et al., 2019), or differences in the climate of the study regions.*

84. Line 373: 'slight overestimation of the CRCM5-LE'. Clarify, what does it overestimate?

    Revised in specific RC-1 comment 83.

85. Section 4.4: the title and content of the section does not match (impacts [title] vs conditions influencing flammability, emergency in other regions. Further, the content is not coherent. Please revise and clarify the message.

    Thank you for your comment. We agree that this section is not consistent in terms of its message. We revised this section in the following way:

    1. Adjusted the Title to "*Increasing fire danger and implications*"
    2. Emphasized the regional and seasonal hotspots of our findings:

       *We identified the Southgerman Escarpment as a hotspot for dangerous FWI conditions within Hydrological Bavaria (see Figures 5 and 6). However, the other subregions are subject to substantial changes in fire danger as well, especially in August and July. On average (median), the fire danger will be high in the Alpine Foreland, Southgerman Escarpment and Eastern Mountain Ranges and moderate in the Alps by the end of the century. In the Alps, the median FWI does not reach as high fire danger levels as the one in the other subregions because of their elevation-dependent colder climate. Nevertheless, this region is very sensitive to climate change induced fire weather changes as demonstrated by its early TOE (see Figure 7) and its significant danger level changes in the months of July and August (see Figure 5).*

    3. Set this in context with climate change impacts:

       *New: Over the course of the 21$^{st}$ century, the fire season will prolong, as fire danger levels are still elevated in September from 2030 onwards (see Figure 6). This suggests that the fire season might extend to at least October towards the end of the century. {...} Prolonged droughts and exacerbating heat events might limit fuel availability and therefore, fire activity in more arid regions, such as the Mediterranean, in the future (Bowman et al., 2020; Pausas and Paula, 2012). For wetter, more productive regions, like our study area, aridity does not limit fuel availability. Bowman et al. (2020) suggested that a declining snow cover in spring and drier fuels in summer will increase burned area in mountain forests, as present in the Alps and Eastern Mountain Ranges. This implies a higher sensitivity to flammable conditions (e.g., after hot and dry seasons) and an extension of fire events to more northern latitudes and higher elevations.*

    4. We generalized the Bavarian specific section to a broader call for mitigation measures in Central Europe

       *New: Expected changes in fire weather in the Mediterranean are of such a magnitude that current fire suppression measures might not be sufficient anymore (Turco et al., 2018). Studies for other regions, e.g. the UK (Arnell et al., 2021) and France (Fargeon*

*et al., 2020), suggested that increases in fire danger should be considered in emergency, land use and management planning to mitigate future fire danger. Our research findings indicate that forest fire mitigation measures must be proposed for central Europe and its mountain regions as well.*

86. France and UK: Several places in the manuscript, results of France and UK is used for guiding and comparing the results of the present study, and to make final recommendations for fire emergency. However, you do not reflect on potentially relevant differences between the regions (e.g. hydroclimotology and vegetation). Please consider commenting on such aspects.

We agree that our manuscript did not sufficiently consider the differences between the study areas of Fargeon et al. (2020) (France) and Arnell et al. (2021) (UK). We disconnected the sentences to avoid misunderstanding and emphasized that these studies refer to other regions than HydBav.

*New: Studies for other regions, e.g. the UK (Arnell et al., 2021) and France (Fargeon et al., 2020), suggested that increases in fire danger should be considered in emergency, land use and management planning to mitigate future fire danger. Our research findings indicate that forest fire mitigation measures must be proposed for central Europe and its mountain regions as well.*

87. Line 397 (and line 406): You state that the study area is not affected by high fire danger to date, but high fire danger is present in relatively large areas in current climate (Fig. 5[2]def, where the dots indicate a change from a currently high level).

We agree with your comment and rephrase the sentence:

*New: To date, the study area is irregularly affected by wildfires and high fire danger occurs only under very rare conditions (90$^{th}$ FWI quantile). However, high fire danger will become more frequent in the future when assuming an RCP8.5 emission scenario.*

88. Line 397-398: Please clarify 'by accounting for natural variability'.

We dropped that phrase (see specific comment 87).

89. Line 398: Please clarify the difference between "strongest increase" and "most hazardous developments".

We rephrased most *"strongest increases"* to *"strongest changes"* and *"hazardous developments"* to *"highest fire danger levels north of the river Danube"* for clarification:

*New: We find the strongest changes and highest fire danger levels north of the river Danube in the summer months of July and August for the subregions South German Escarpment and Eastern Mountain Ranges.*

90. Line 400: please clarify in what terms, and in which results the statement "less strongly affected" applies. For example, in fig 5[2], Alps is the only region with dots in April and May, and the two regions you mention increase multiple fir danger levels as seen e.g. in Fig. 6[2] august. As mentioned earlier in the manuscript (line 324-326), increases in classes may provide a better approach to assess increases due to non-linearity, and thus and linear comparison (e.g. Fig 7) may not the best way to conclude the strongest trends.

We revised this section and dropped the imprecise statement about regions which are "less strongly affected".

*New: We find the strongest changes and highest fire danger levels north of the river Danube in the summer months of July and August for the subregions South German Escarpment and Eastern Mountain Ranges.*

91. Line 401: the statement that FWI has a stronger variability for Alps and Eastern Mountain Ranges contradicts the findings in Fig. 7, where the standard deviation is smaller for these regions compared to the other subregions. Please clarify.

We appreciate your comment to set this in context with the findings of Figure 7. Figure 7 is derived from highly aggregated data (30-year daily fire season running means) and therefore has a different aggregation level than Figures 5 and 6. However, since this does not clarify the findings and we decided to drop this section.

92. Line 404: please consider repeating the hypothesis, and structure the conclusions by these.

We restructured the conclusions to follow the different research questions/hypotheses.

*New: Our results demonstrate that fire danger increases substantially throughout the study area during this century. We find the strongest changes and highest fire danger levels north of the river Danube in the summer months of July and August for the subregions South German Escarpment and Eastern Mountain Ranges. Our results also show that the time of emergence (TOE) is reached in all subregions before 2050. Moreover, they show that not only the mean, but also the lowest range of the running mean, indicated by the CRCM5-Les standard deviation, exceeds the upper limits of the current climate standard deviation (1980-2009) in all subregions before 2099 for the $90^{th}$ FWI quantile. Last, our findings demonstrate that the return periods of present-climate 100-year FWI events shift towards 10-year events by 2090 and the return periods of present-climate 100-, 50- and 20-year events shift to 50-, 20- and 10-year events, respectively, before 2050 for all subregions.*

93. Line 407: please clarify what 'also' refer to.

We dropped "*also*".

94. Line 410: What about the data of the subregions and land cover (Fig. 1 and A1)?

We added the sources for the subregions and landcover data to the data availability section.

*New: The datasets used in this study can be found in the following repositories: CRCM5-LE: https://www.climex-project.org/de/datenzugang, ERA-5 based FWI: DOI: 10.24381/cds.0e89c522 (31.01.2023), sub-regional division: https://www.lfu.bayern.de/natur/naturraeume/index.htm, landcover data from Copernicus Land Monitoring Service:* https://land.copernicus.eu/pan-european/corine-land-cover/clc2018*.*

95. Figure C1: Why do you use $95^{th}$ percentile and not $90^{th}$ percentile as done in the remaining analysis?

Thank you for this remark. We aimed to show more extreme results in the sensitivity analysis and therefore decided to use the $95^{th}$ quantile.

96. Why number the Figures A1, B1 and C1 instead of A1, A2 and A3 as is normally done?

Thank you for pointing this out. We fixed this overleaf template issue and numbered the Figures in the appendix according to your suggestion.

**Reviewer 2**

Miller et al. (2023) present a detailed analysis of the climate change impacts on fire weather across a study region in Central Europe that is historically not fire-prone. They accomplish this by using the Single Model Initial-Condition Large Ensemble (SMILE) of a regional climate model to: a) study the temporal and spatial trends in the Fire Weather Index (FWI), a commonly used indicator of fire weather; b) disentangle the contribution of natural variability from climate trends in the median and extreme percentiles of the FWI as inferred from two metrics: time of emergence (TOE) and temporal evolution of the current fire danger return period.

Overall, I found the manuscript to be well-written, and I appreciated the clear presentation of the analysis techniques and results throughout the text. The subject matter is quite important and within the purview of NHESS's scope. However, I think there are several areas where the authors could improve the discussion in the manuscript, either through clarification of confusing statements or by illustrating their argument with an additional figure or two. Once these changes are incorporated, I would be happy to review the manuscript's suitability for publication. Please find my comments listed below.

Thank you very much for editing our manuscript and critically reflecting on our results. We highly appreciate your constructive comments and implemented your feedback.

*Comments:*

- L130: The phrasing of this statement lends me to believe that FWI is calculated using antecedent weather over the previous 52 days. This is, however, not the case based on the documentation for FWI available here: https://cfs.nrcan.gc.ca/publications?id=19927

  Thank you for raising your concerns regarding this statement. We re-read the documentation of the FWI (provided in your link) and realized that the time delay of 52-days in the Drought Code refers to the drying rate and not the antecedent weather conditions. We revised the section and dropped the statement about the previous 52-days.

  *New: The CFFWIS uses meteorological conditions of the atmosphere on the day of interest (temperature, relative humidity, wind speed – all at noon – and 24-h accumulated precipitation) and antecedent weather conditions represented by fuel moisture codes to estimate fire behaviour and fuel moisture (van Wagner, 1987).*

- L146: I appreciated the authors quoting the units for FWI. However, these units are conspicuously missing in the relevant tables and figures in the rest of the text (Table 1, Figure 4, 5, 6, 7)

  Thank you for comment, which we highly value. According to Van Wagner (1987), the unit of the FWI is *I* or *HWR* (fire intensity represented by energy output rate (H) per fuel consumed per unit area (W) and rate of spread I). We could add this unit to the tables and figures of our manuscript, but we think this will add confusion to readers which are not familiar with the field. However, we updated the labels in the Figures to FWI to indicate that the color bar refers to the FWI and not to one of the FWIs subindices.

- L20": "...and 99th percentiles [of the FWI] in the present climate per"od" (missing text)

  Thank you for pointing this out – we took this remark into account when rephrasing the sentence.

*New: We derived FWI quantiles in the current climate period for non-exceedance probabilities of p = [0.9,0.95,0.98,0.99] and the corresponding FWI return periods T of 10-, 20-, 50-, and 100 years {...}.*

- L20": "We then compute the non-exceedance probability of the present percentiles given the future cumulative distribut"o– -- Present percentiles and future cumulative distribution of what quantity? The writing here can be improved, in general, to clarify whether the percentiles are with respect to all 30 years of the whole ensemble or of one model within the ensemble.

  Thank you for highlighting the missing description. Since this section was highlighted by RC-1 (see general comment 1) as well, we edited this section carefully and provided a more detailed description of how we derived the return periods:

  *New: For the second analysis, we calculated changes in the return periods of FWI quantiles that correspond to return periods of 10-, 20-, 50- and 100-years under current climate conditions (period 1980–2009) for the four subregions. To do so, we pooled daily FWI values over the entire 50-member ensemble (183 days per fire season x 30 year climate period x 50 members). Using this data pool, we determined the non-exceedance probability p of each FWI value in the present climate period using its rank r and the total sample size n following p = r/n. We derived FWI quantills in the current climate period for non-exceedance probabilities of p = [0.9, 0.95, 0.98, 0.99] and the corresponding FWI return periods T of 10-, 20-, 50-, and 100 years using T = μ / (1 − p), where μ is the inter-arrival time (1/183 days in a fire season) (Coles, 2001). To analyze changes in return periods over time (from 1980 to 2099), we created centred, rolling 30-year windows for each ensemble member (183 days per fire season x 30 year climate period) and derived the cumulative distribution of the time window using the rv_histogram.cdf function of the Scipy package in Python (Virtanen et al., 2020). We mapped the FWI quantiles representing the 10-, 20-, 50-, and 100 year return periods of the current period (19–0 - 2009) to future return periods, by deriving their non-exceedance probability p in the cumulative distribution of the rolling window climate period (future). Next, we placed their futu–e probability p into T = μ / (1 − p) (Coles, 2001) to determine the return period T of the presen– FWI quantile under future climate conditions. This approach allows us to show how the return– period of e.g. the current 100-year FWI will change over time with climate change. Due to the centered window approach, the first full 30-year window is 1995 and the last full 30-year window is 2084. Therefore, we show results between 1995 and 2084.*

- L240-24": "This finding indicates that the distribution of the FWI extremes resembles the distribution of the FWI medi"n– -- This statement seems unintuitive: wou'dn't the distribution of median and extreme FWI (which contains temperature as a predictor) diverge in a warming world? Perhaps this is an artifact of how the TOE is calculated with SMILEs and there is not enough variability, or that 90th percentile 'sn't extreme enough in the future. It would be great to see a version of Fig. 7 with the 95th and 99th percentile as well.

  Thank you for your comment, which we highly appreciate. Indeed, it is counter-intuitive, that the variability (turquoise shading in Fig. 7) appears to be the same for the [5]0th and [9]0th percentile. Variability should increase for the [9]0th percentile and yet, it looks smaller than for the [5]0th percentile. The reason for this is that we initially did not display the results for the two percentiles on the same scale. When plotted on the same scale (updated Figure 9 in this response to the reviewer), it becomes apparent that indeed and as expected, the variability is substantially larger for the 90[th] than for the 50[th] percentile. We adjusted Fig. 7 in the

manuscript to a common y-axis between the [5]0th and [9]0th percentile (see below), which clearly shows that the distribution of median and extreme FWI is not the same, as stated previously. We therefore removed this line and updated Fig. 7.

[Figure]

*Figure 7 - Increases of the median ([1], 50th quantile) and extreme ([2], 90th quantile) FWI between 1980 and 2099 differentiated by subregion: (a) Alps, (b) Alpine Foreland, (c) Southgerman Escarpment, (d) Eastern Mountain Ranges. The ensemble mean increase is derived on a fire season basis and represented by solid pink lines smoothed over a 30-year window. The ensemble᾽s standard deviation is represented by shaded blue areas. Black solid and dashed lines represent the ensemble mean and spread of the present climate period (1980‑2009). The TOE, marked with a pink dot and year annotation, is reached when the ensemble mean (pink line) crosses the upper boundary of the ensemble standard deviation in the present climate period (black dashed line)*

- Figure 9: Why is the ensemble mean of the 100-year return period only about ~75-80 years for all 4 subregions?

We created centered 30-year windows (between 1980 and 2099) for each member to determine the FWI percentiles corresponding to the different return periods of the present climate period (all 50 members). The first full 30-year window is 1995 and the last full 30-year window is 2084. Therefore, the ensemble mean of the 100-year return period for the present all member pool is 100 in the year 1995. This is not shown in the Fig. 9 of the preprint. We updated the figure and methods section (see your comment on L208) accordingly.

[Figure]

*Figure 9 - Future changes (1995‒2084) in the return periods of FWI-quantiles corresponding to return periods of 10, 20, 50 and 100 years under current climate conditions (1980‒2009) for the four subregions: (a) Alps, (b) Alpine Foreland, (c) Southgerman Escarpment, (d) Eastern Mountain Ranges. The thick solid line represents the CRCM5-LE mean, while thin lines represent the 50 ensemble members.*

- L293: "...which has to be discussed..." -- improve phrasing.

  We rephrased this section in correspondence with RC-1 (comment 68).

  ***New:*** *The CRCM5 represents FWI at much finer spatial resolution than CanESM2 and therefore adds robust high-resolution features (Böhnisch et al., 2020). However, we find tiling patterns on the border between the Southgerman Escarpment and the Alpine Foreland (see Figure 5 [2]).*

- L293: "...tiling pattern visible in figure 5 [2]..." -- [2] seems to be a typographical error.
  We removed this typographical error.

**Reviewer 3**

*General comments*

This paper investigates a regional SMILE (Single Model Initial-Condition Large Ensembles) of the Canadian regional climate model version 5 (CRCM5-LE) over Central Europe (Hydrological Bavaria) under the RCP8.5 scenario from 1980 to 2099, to analyze fire danger trends in a currently not fire-prone area. This evaluation of fire danger (vs current climatic conditions) uses Canadian Fire Weather Index (FWI), and the 3-hourly meteorological data from the large ensemble of available CRCM5-LE simulations. The authors demonstrate that this ensemble (at 0.11°) is a suitable dataset to disentangle climate trends from natural variability in a multivariate fire danger metric. Various results show the increase in the median and extreme percentile of the FWI in the northern parts of the study area (in July and August). The southern parts of the study region are less strongly affected, but time of emergence (TOE) is reached there in the early 2040's. In the northern parts, the climate change trend exceeds natural variability in the late 2040's. In the future, a 100 year (return period) FWI event will occur every 30 years by 2050 and every 10 years by 2099. This study is of a strong interest in order to help the refinement of fire management strategy to reduce the consequences of such forest fires, and to improve the preparation or adaptive capacity knowing the potential changes of this natural hazard under ongoing climate change. The article is well written, and well-articulated in term of scientific findings and presentation of main outcomes.

I will suggest to add insights or discuss limitations from the use of one single RCM driven by one specific GCM (i.e. CanESM2) whatever the number of ensemble runs used, as systematic biases from the driving GCM can influence the downscaling simulations and derived products (ex. FWI). For example, as noted in various studies, biased atmospheric circulation features due to coarse-scale resolution (ex. around 2.8° for the CanESM2 model) and/or missing orographic drag, sea surface temperature simulated features, etc. which affect the simulated blocking features (see Pithan et al., 2016; Schiemann et al., 2017; Davini and d'Andrea, 2020) or atmospheric circulation variability responsible for the occurrence of climate extremes over Europe (see Faranda et al., 2023). As revealed in the recent work of Faranda et al. (2023), atmospheric circulation changes modulate extreme events already in the present climate in Europe, and summer heatwaves as well as large regional and seasonal changes in precipitation and surface wind, i.e. hazards or meteorological variables responsible for the occurrence and severity of fire danger (variables used to compute the FWI indices). Also, as shown in Zappa et al. (2014), CanESM2 tends to have one of the largest track density biases for extratropical cyclones among CMIP5-GCMs, as well as in blocking frequency biases over both Norwegian Sea, and central Europe (see their Figure 3). These two features of atmospheric circulation variability play a key role in the occurrence of both anomalies of temperature and precipitation across the study area. Faranda et al., (2023) and Strommen et al. (2019) strongly argue to use at least three or more ensemble members to deal with the importance of the regional response to anthropogenic forcing (Corti et al., 1999; Palmer, 1999), representing these atmospheric regimes correctly whatever the GCM. Also as noted in Deser et al. (2020), the use (in future works) of large ensembles from different GCMs will give new insights into uncertainties due to internal variability versus model differences. In fact, concerning the so-call internal variability (or natural variability mentioned in the paper that it is partly evaluated from the CRCM5-LE), the current study cannot consider using one single ensemble from one RCM-GCM matrix, structural variability from the differences in (GCM) model formulation, including physics and parameterization, resolution, etc. This structural variability strongly affects the climate change responses at the regional or local scale (uncertainties in the dynamical downscaling simulations depend on the driving GCM; see different studies from the EURO-CORDEX project).

In summary, after having recall the shortcoming and include nuance about the robustness of the climate change signals for the FWI indices across the area, using one single ensemble sample from one combination of RCM-GCM, the paper is sufficiently relevant and scientific documented (sound) to be published after minor revisions. Please see also my specific comments below.

We appreciate the careful revision of RC-3 on our manuscript. We clarified the highlighted sections specifically and improved the overall description of our modelling set-up, which is a regionally downscaled single-model initial condition large ensemble (SMILE).

We agree with RC-3 that our modelling setup cannot represent model spread because our analysis focuses on a regional SMILE of one GCM-RCM combination only. The reason for this model choice is that we want to overcome biases in atmospheric circulation features due to coarse-scale resolution, e.g. orographic drag, by using a regionally and dynamically downscaled SMILE. Rather than on in-between model spread, this study focuses on the quantification of internal variability, which requires the use of a SMILE (see Deser et al. 2020). This is also proposed by Faranda et al. (2023). The SMILE used in our study (CRCM5-LE) was compared to the CORDEX-Family by Von Trentini et al. (2019). However, to date, only 2 other regional SMILES for the area of interest exist, but they differ in the study domain and spatial resolution (see Von Trentini 2020). The implications of the regional downscaling of the CanESM2-LE using the CRCM5 was investigated by Böhnisch et al. (2020). The study of Böhnisch et al. (2020) shows that "important large-scale teleconnections present in the driving data propagate properly to the fine-scale dynamics in the RCM". Further, the studies of Mittermeier et al. (2019 and 2021) demonstrate that the CRCM5-LE is capable to quantify large scale pressure patterns which lead to heat-waves and extreme precipitation (i. e. Vb-Cyclones).

Since we find that the reason for using a SMILE was not emphasized clearly enough in our initial manuscript, we edited the introduction and discussion section and reflected the strengths and weaknesses of the CRCM5-LE better by adding new paragraphs:

*New Paragraph (Introduction): In this study, we therefore use the CRCM5-LE, a dynamically downscaled, regional, high-resolution SMILE (0.11° grid cell size) nested into the CanESM2-LE (Fyfe et al., 2017), to disentangle climate change induced fire danger trends from internal variability over heterogeneous landscapes in central Europe. Benefits of using a regional instead of a global SMILE are the better spatial representation of climatic patterns, such as pressure patterns leading to extreme precipitation (Mittermeier et al., 2019) or heat waves (Bohnisch et al., 2023), and the seasonality of these extremes (Felsche et al., 2023; Bohnisch et al., 2021; Wood and Ludwig, 2020).*

*New Paragraph (Discussion): Though SMILEs can account for internal variability, they are not designed to evaluate the structural uncertainty of the climate models (Deser et al., 2020). Structural or model uncertainty can only be assessed in multi-model studies (i.e. Fargeon et al., 2020). In order to quantify both – internal variability and structural uncertainty – it would be necessary to use multiple 320 SMILEs as provided by the "Multi-Model Large Ensemble Archive" (MMLEA; Deser et al., 2020). However, all SMILEs in the MMLEA are based on Global Climate Models (GCMs) with a spatial resolution ranging between 2.8° and 0.9° (Deser et al., 2020). On a regional and local scale, a higher spatial resolution is needed to quantify climate change impacts on forest fires. For Europe, only two other dynamically downscaled SMILEs from Regional Climate Models (RCMs) exist besides the CRCM5-LE: The 16-member EC-EARTH-RACMO ensemble at 0.11° (Aalbers et al., 2018) and the 21-member CESM-CCLM ensemble at 0.44° grid cell size (Brönnimann et al., 2018; Fischer et al., 2013). The models differ in their study domain (EC-EARTH-RACMO) and spatial resolution (CESM-CCLM) from the CRCM5-LE used here (Wood, 2023; von Trentini et al., 2020).*

*Specific comments:*
Please be consistent when you use RCP8.5 (without "space" between P and 8.5) in the text.

We unified to RCP8.5 and RCP2.6 without a blank space.

Abstract:

Please add few words or one sentence considering the need to use larger downscaling ensemble from different GCMs in order to develop more robust climate change signals for all meteorological variables used to compute the FWI indices (further work).

Thank you for your comment. We agree that this should be nuanced in the abstract and therefore added a new sentence to the abstract.

*New: So far, the SMILE framework has only been applied for fire danger estimation on a global scale. To date, only a few dynamically downscaled regional SMILEs exist, although they enhance the spatial representation of climatic patterns on a regional or local scale.*

Introduction:

**Line 22:** Please add (Canada) after British Columbia.

We added *(Canada)* after British Columbia.

**Line 63:** Please nuance this statement, as natural variability of the climate system is not fully represented by one single model initial-condition large ensemble, as a GCM generates a simplification of a complex reality (i.e. the climate system) and includes structural variability and biases that we need to consider in any downscaling exercise (see Strommen et al., 2019; Deser et al., 2020; and recommendations or Plausibility criteria in the new CORDEX-CMIP6 in Sobolowski et al., 2023).

We added a statement highlighting that a SMILE – while capturing internal variability - does not allow for the quantification of the structural uncertainty. For this latter purpose, a multi-model large ensemble has to be used.

*New: This challenge can be addressed by evaluating climate model simulations derived from a single model initial-condition large ensemble (SMILE) which enables a clear isolation of the forced climate change signal from internal variability (Deser et al., 2012). SMILEs represent an ensemble of simulations derived using one single climate model started at different initial conditions. The ensemble spread between the different SMILE members provides a robust estimate of the internal variability, from which the forced response of the climate change scenario can be estimated by averaging over the SMILE members for a specific variable, e. g. temperature (Deser et al., 2020). While single SMILEs allow for the quantification of internal variability, they do not enable a quantification of model uncertainty (Deser et al., 2020).*

Data and Methods

**Line 96:** As mentioned in Fargeon et al. (2020) and many other studies, bias correction alters the physical consistency of modelled climate and meteorological variables in particular at high frequency (ex. sub-daily values). Quantile mapping makes strong assumptions regarding bias stationarity and can

break the co-variation between climatic variables, in particular at high frequency or meteorological scale (i.e. that is the case here when computing the daily FWI indices). Can the authors provide some insight about these drawbacks or physical consistency among meteorological variables after bias correction and the implication of this in computing FWI indices?

Thank you for this remark. We agree that univariate bias correction methods may not perfectly represent the covariation between variables. This is especially relevant for multivariate indices like the FWI used in this study. Zscheischler et al. (2019) analyze the effect of univariate bias adjustment for multi-variate hazards and discuss different studies arguing for and against the need of multi-variate bias correction methods. For example, Yang et al. (2015) argue that a univariate bias correction is sufficient, while Cannon et al. (2018) propose the opposite. They find that "we cannot draw the general conclusion that multivariate bias adjustment is not necessary in any case from individual, typically regional, studies" and "it is difficult to pin down under which exact circumstances univariate bias adjustment might fail".

We added a section to the discussion, where we discuss the advantages and disadvantages of univariate / multi-variate bias correction:

*New Paragraph: Correcting the bias between climate model data and observation data is often an inevitable step in climate impact studies (Piani et al., 2010). The CRCM5-LE was bias adjusted using univariate quantile mapping (Poschlod et al., 2020; Mpelasoka and Chiew, 2009). Such univariate methods can change the co-variation between multiple variables (Zscheischler et al., 2019) with potential impacts on the analysis of complex indices like the FWI. Therefore, there have been calls for the use of multi-variate bias correction methods (Cannon, 2018). However, Yang et al. (2015) showed that univariate bias correction was sufficient to study fire weather changes in Sweden. Furthermore, multivariate bias correction is a non-trivial task and fixing co-variation issues between variables might lead to other problems, e.g. with the representation of temporal or spatial dependencies (Vrac, 2018). In this regard, we assume that the univariate bias correction applied on the CRCM5-LE is appropriate for our analysis.*

**Line 100:** "FWI extremes are significantly better…": Yes, but these FWI extremes are physically coherent and consistent with meteorological fields?

Together with your previous comment, we discuss this in the new discussion paragraph. The studies of Yang et al (2015) and Cannon et al. (2018) have shown that bias-corrected climate model output better reflects observed fire danger. We therefore adjusted this section:

*New: Bias-corrected data have been commonly used for projections of fire weather indicators such as the FWI (e. g. Yang et al., 2015; Cannon, 2018; Kirchmeier-Young et al., 2017; Ruffault et al., 2017; Fargeon et al., 2020), as they have been shown to reflect fire danger more accurately than raw climate data when compared to observational data (Yang et al., 2015).*

Study area

**Line 112:** Please can the authors provide some reference from which dataset these (climatological) values come from ? E-Obs, …?

We added a notate that the data were derived from the meteorological SDCLIREF dataset. This Sub-Daily Climatological REFerence dataset (SDCLIREF) was created within the scope of the project that this study is based on (ClimEx). It combines hourly and disaggregated daily station data and is described in detail by Brunner et al. (2021).

*New: For the bias correction, the meteorological Sub-Daily Climatological REFerence dataset (SDCLIREF), which combines hourly and disaggregated daily station data (Brunner et al., 2021), served as an observation reference.*

*{…}*

*Mean precipitation over the study area increases from North to South, with annual precipitation sums ranging from 500 to 1100 mm in the South German Escarpment, around 1000 mm in the Eastern Mountain Ranges, 1500 and 2500 mm in the Alpine Foreland, and 1000 and 2000 mm in the Alps according to the SDCLIREF observation dataset for the present climate period between 1980 and 2009.*

The Canadian Fire Weather Index

**Line 125:** Please correct "… assess…".

We corrected "assesses" to "assess".

Estimating Fire Danger using the CRCM5-LE

**Line 170:** "…, this does affect the climate change impacts assessment…". Yes, but the CRCM5 ensemble seems to underestimate the interannual anomalies of FWI that we see from the reference (ERA5) database. Please can you comment this, as the year 2002 seems to be not in the range of below 75th percentiles of the observed FWI across Europe but rather on more extreme side? This underestimation of interannual FWI anomaly can be due to the debiased method which has an effect on the decreasing year to year variability of each of the ensemble simulations?

Thank you for pointing this out. We realized that the outlier in the ERA-5 Dataset is referring to the year 2003 and the time periods between our comparison differed (ERA-5: 1981-2010, CRCM5-LE: 1980-2009). We remade the figures by using a unified time period (1980 – 2009). The year 2003, especially the summer months, were affected by an extreme heat wave in Europe, which is reflected by the high FWI value in the ERA-5 dataset. However, the CRCM5-LE data are modeled data and do not contain observed values. Figure 3 shows, that the 2003 heatwave lies below the 75$^{th}$ percentile of our single-model initial-condition large ensemble and is well situated in the ensemble spread. Following the framework of Suarez-Guiterrez et al. (2021) (https://doi.org/10.1007/s00382-021-05821-w), we interpret that the CRCM5-LE overestimates internal variability, because most observation points lie between one standard deviation in the ensemble and all observation points lie between the 25$^{th}$ and 75$^{th}$ percentile of the ensemble. For this reason, we cannot precisely answer the question of how the bias correction should affect an underestimation of interannual FWI anomalies. Further we discussed this point more closely in the answer to your comment (RC-3) on Line 302.

We added a remark to this paragraph which states, that the ensemble overestimates interannual variability:

*New: A majority of the reference data points is located within one standard deviation of the CRCM5-LE (see Figure 3). The remaining data points are located between the 25th and 75th quantile of the ensemble (blue lines). Overall, the ensemble slightly overestimates the reference FWI dataset with an average deviation of +0.76. However, it includes the reference dataset values within its 25th and 75th quantiles*

[Figure]

*Median FWI for the CRCM5-LE mean (thick blue line) and standard deviation (light blue shading) in comparison to the reference dataset of Vitolo et al. (2020) marked pink (X for values, lines for deviation from the CRCM5-LE mean). Top and bottom blue lines mark the 25th and 75th percentile of the CRCM5-LE.*

Discussion

**Line 278:** "… next few decades…". This mean 2080s? Please be precise.

We changed "*next few decades*" to "*until the end of the 21$^{st}$ century*" to be precise that we mean our whole observation until the end of the century.

Data basis

**Line 302:** "…uncertainties related to emission scenarios and the chosen climate model". You do not discuss this point (i.e. choice RCM or single RCM-GCM), please provide some insights as suggested in the general comments.

We changed this section in correspondence to comment 70 of RC1 and added a section that explains the difference between the CRCM5-LE and other CORDEX models in terms of precipitation and temperature.

*New: In comparison to the CORDEX multi-model ensemble, the CRCM5-LE shows drier and warmer climate change signals for temperature and precipitation (von Trentini et al., 2019). These characteristics of the CRCM5-LE are in line with the results from the validation (see Figure 3) and suggest that our results represent an upper limit of the expected changes in future fire danger.*

Further, we added a paragraph discussing the benefits and downsides of using a regionally downscaled SMILE in chapter 4.2 (lines 317ff.).

*New: Though SMILEs can account for internal variability, they are not designed to evaluate the structural uncertainty of the climate models (Deser et al., 2020). Structural or model uncertainty can only be assessed in multi-model studies (i.e. Fargeon et al., 2020). In order to quantify both – internal variability and structural uncertainty – it would be necessary to use multiple SMILEs as provided by the "Multi-Model Large Ensemble Archive" (MMLEA; Deser et al., 2020). However, all SMILEs in the MMLEA are based on Global Climate Models (GCMs) with a spatial resolution ranging between 2.8° and 0.9° (Deser et al., 2020). On a regional and local scale, a higher spatial resolution is needed to quantify climate change impacts on forest fires.*

Spatio-Temporal Trends and Variability

**Line 359:** "… on the whole year instead of the summer season only": Potential avenue will be to take into account the snow cover season or overwintering conditions, based on cumulative precipitation during the cold season, as used in Canada (see McElhinny et al., 2020).

Thank you for this comment, we indeed did not discuss the overwintering option of the Drought Code. We commented on this already in RC-1 comment 81 as follows: *We agree that the FWI is not suitable for the winter season and suggest considering using other approaches in cases where the winter season is explicitly considered, e.g., the one proposed by Pezzatti et al. (2020). However, our study only focuses on the months April to September, when snow cover at a 11 km grid scale plays a minor role for forest fire danger, because it occurs only in unvegetated high alpine terrain, which is sampled only by a small fraction of the 11 km grid.*

*New: For the Southern Alps, Wastl et al. (2012) identified the main fire season between December and April because of low precipitation and decreased fuel moisture outside of the vegetation period (Conedera et al., 2018). Future studies assessing changes in fire danger and fire events in temperate climate regions should therefore consider the whole year instead of the vegetation season only.*

**Line 373:** "…or the slight overestimation of the CRCM5-LE…": Again, this can be due to the lack or limited internannual variability in the debiased CRCM5-LE variables? Please comment slightly on this issue. Line 374: " … a substantial larger database…". Yes, but this is a single model (CRCM5) driven by an ensemble of one GCM (CanESM2), as in Fargeon et al. (2020) they use 2 RCMs driven by 3 different GCMs. Please nuance this statement.

We value your comment regarding the overestimation of natural variability of the CRCM5-LE. We edited the section to emphasize more strongly that the CRCM5-LE is a regionally downscaled single-model initial-condition large ensemble (SMILE) of a GCM (CanESM2-LE), resulting in 50 climate realizations of a spatial resolution of 0.11° (see new paragraph lines 317ff.)..

*New: Structural or model uncertainty can only be assessed in multi-model studies (i.e. Fargeon et al., 2020). In order to quantify both – internal variability and structural uncertainty – it would be necessary to use multiple SMILEs as provided by the "Multi-Model Large Ensemble Archive" (MMLEA; Deser et al., 2020). However, all SMILEs in the MMLEA are based on Global Climate Models (GCMs) with a spatial resolution ranging between 2.8° and 0.9° (Deser et al., 2020). On a regional and local scale, a higher spatial resolution is needed to quantify climate change impacts on forest fires. For Europe, only two other dynamically downscaled SMILEs from Regional Climate Models (RCMs) exist besides the CRCM5-LE.*

**Line 375:** "… which helps to better represent natural variability": See my previous remarks, natural variability is more complex that internal variability extracted from one single RCM-GCM matrix, as at least you need to consider more range of boundary conditions, from as many as possible GCMs as those are the main source of uncertainties in particular from the atmospheric circulation over Europe pointed out by Faranda et al. (2023).

Thank you for your comment, which we highly appreciate, because it demonstrates that our study set-up is not clearly described. The CRCM5-LE used in this study (s. Leduc et al. 2019) is a single-model initial condition large ensemble, which is widely used on a global scale (e. g. Deser et al. 2020, Suarez-Guiterrez et al. 2021) to distinguish internal / natural variability from the forced response of a climate model simulation. Faranda et al. (2023) state that they would use "initial condition large ensembles to separate forced signals from internal variability in the context of our analog analysis" in future work. However, our study does not aim to quantify model uncertainty but to study climate variability. We added a section on this to the discussion (s. your comment on the abstract and line 62, RC-3 Abstract, RC-3 Introduction Line 63). We hope the changes in the previous comments clarify the modelling setup in our study.

*New: Though SMILEs can account for internal variability, they are not designed to evaluate the structural uncertainty of the climate models (Deser et al., 2020). Structural or model uncertainty can only be assessed in multi-model studies (i.e. Fargeon et al., 2020). In order to quantify both – internal variability and structural uncertainty – it would be necessary to use multiple SMILEs as provided by the "Multi-Model Large Ensemble Archive" (MMLEA; Deser et al., 2020).*

**Line 377:** " … fire danger are robust…": From the ensemble runs used (i.e. link to the sample size or RCM-GCM matrix). Please nuance this statement.

Thank you for your comment. We changed the sentence to emphasize that we used a regionally downscaled SMILE:

*New: While Fargeon et al. (2020) point out that fire danger increases are hard to distinguish from internal variability in northern France when using a multi-model ensemble, we demonstrate that increases in fire danger can robustly be quantified for central Europe when using a regional SMILE.*

Conclusion

**Line 404:** "We accept all of the three hypotheses…": Please be more explicit and comment about these, in particular H2 and H3.

We appreciate your comment, which was also remarked in a similar way by RC-1 (comment 92). We restructured the conclusions to follow the different research questions/hypotheses.

*New:* *Our results demonstrate that fire danger increases substantially throughout the study area during this century. We find the strongest changes and highest fire danger levels north of the river Danube in the summer months of July and August for the subregions South German Escarpment and Eastern Mountain Ranges. Our results also show that the time of emergence (TOE) is reached in all subregions before 2050. Moreover, they show that not only the mean, but also the lowest range of the running mean, indicated by the CRCM5-LEs standard deviation, exceeds the upper limits of the current climate standard deviation (1980-2009) in all subregions before 2099 for the 90th FWI quantile. Last, our findings demonstrate that the return periods of present-climate 100-year FWI events shift towards 10-year events by 2090 and the return periods of present-climate 100-, 50- and 20-year events shift to 50-, 20- and 10-year events, respectively, before 2050 for all subregions.*

References

Line 531: The reference Separovic et al. (2013) is not at the right place in the list.

We corrected the misplacement of the reference Separovic and checked the bibliography for typographical errors.

---

## Author Response (AR2)

Dear Editor,

Thank you very much for your feedback and the reviewers' final remarks on our manuscript. We implemented the changes suggested and cross-checked the manuscript again. Further we provide a DOI to the FWI dataset derived in our study (see Data availability section). We just requested the dataset to be published on the server of Envidat. It might therefore take until Monday until it is publicly available. However, the DOI of the dataset is set. We are looking forward to our last round of iteration.

Best regards,

Julia Miller

**Reviewer 1**

The manuscript has been substantially improved, and the authors have thoroughly followed up the reviewer comments.

I recommend publication after minor clarifications:

Line 20: Please clarify (i.e. write out) what is meant by 'they'.

Old: However, they reach their time of emergence (TOE) in the early 2040's because of very low current fire danger.

New: However, these regions reach their time of emergence (TOE) in the early 2040's because of very low current fire danger.

Fig. 1: Please clarify what 'landscapes' refers to and what 'land cover' refers to, to avoid confusion between these two overlapping terms.

Thank you for pointing this out. The figure caption was imprecise.

Old: Subregions of Hydrological Bavaria by landscapes and land cover (modified, CLMS (2021))

New: Complex landscapes (subregions) of Hydrological Bavaria (Landesamt für Umwelt, 2023) by elevation, (European Environment Agency, 2016).

Line 173-175/Fig 3: Please state spatial domain of this part of the analysis (e.g. "our study region").

Old: To ensure that the CRCM5-LE sampled the FWI in a meaningful way, we compared the CRCM5-LE FWI median for the current period (1980–2009) with the median of the ERA-5-based FWI dataset from Vitolo et al. (2020), hereafter referred to as the "reference dataset" (REF).

New: To ensure that the CRCM5-LE sampled the FWI in a meaningful way, we compared the CRCM5-LE FWI median for the current period (1980–2009) with the median of the ERA-5-based FWI dataset from Vitolo et al. (2020), hereafter referred to as the "reference dataset" (REF) over the study domain.

Line 178: Repetition of previous sentence (line 176)?

We deleted the sentence.

Line 304-306: The logic of the argument is not clear. Maybe it is the term 'Extreme fire weather', which can refer to both the 90th quantile and the coloured classes in Fig. 5, which give rise to the confusion. Please consider clarifying the argument.

We clarified the statement by adding $90^{th}$ quantile in brackets next to "extreme fire weather" and emphasizing that we use FWI quantiles.

Old: In time, extreme fire weather (90th quantile) is more likely to occur in the second half than in the first half of the fire season because the differences between the median and extreme FWI quantiles are smaller in April, May and June, than in July, August and September (see Figure 5).

New: In time, extreme fire weather **(90th quantile)** is more likely to occur in the second half than in the first half of the fire season because the differences between the median and extreme FWI **quantiles** are smaller in April, May and June, than in July, August and September (see Figure 6).

Line 367: Please consider changing to "One average, the median fire danger ..." or similar, for clarity.

Old: On average (median), the fire danger will be high …

New: On average, the median fire danger will be high …

Line 388: I assume "forest fire mitigation measures" already exist for the region, and that the point of this sentence is that changes in fire danger should be considered in these mitigation measures (more in line with sentences line 387 and line 404). If this is correct, please clarify the sentence.

We softened the statement according to your suggestion.

Old: Our research findings indicate that forest fire mitigation measures must be proposed for central Europe and its mountain regions as well.

New: Our findings indicate that forest fire mitigation measures must be adapted for central Europe and its mountain regions as well.

**Reviewer 2**

Dear Authors,

I have now carefully gone through your response to my comments, the revised manuscript, and new information added based on reviewers' comments. I believe the manuscript is significantly improved, and I'm satisfied with the framing of the methods and results. Hence, I enthusiastically recommend that the paper be accepted for publication in NHESS.

Before acceptance, it'd be great if the authors could correct one small typographical error: replace "MAI" with "MAY" in Figure 6 to ensure consistency with other English month names. :)

We changed the description of MAI to MAY

[Figure]

*Figure 6. Fire rings show the FWI of the ensemble mean of the monthly median ([1], 50th quantile) and extreme ([2], 90th quantile) of each subregion (Alps, Alpine Foreland, Southgerman Escarpment and Eastern Mountain Ranges (a-d)) during the fire season (April - September) between 1980 and 2099.*